Resource

# Single-cell analysis of human MAIT cell transcriptional, functional and clonal diversity

Lucy C. Garner [1][✉], Ali Amini [1], Michael E. B. FitzPatrick [1], Martin J. Lett [2], Gabriel F. Hess [3], Magdalena Filipowicz Sinnreich[2,4], Nicholas M. Provine [1,7] & Paul Klenerman [1,5,6,7][✉]

Mucosal-associated invariant T (MAIT) cells are innate-like T cells that recognize microbial metabolites through a semi-invariant T cell receptor (TCR). Major questions remain regarding the extent of human MAIT cell functional and clonal diversity. To address these, we analyzed the single-cell transcriptome and TCR repertoire of blood and liver MAIT cells and developed functional RNA-sequencing, a method to integrate function and TCR clonotype at single-cell resolution. MAIT cell clonal diversity was comparable to conventional memory T cells, with private TCR repertoires shared across matched tissues. Baseline functional diversity was low and largely related to tissue site. MAIT cells showed stimulus-specific transcriptional responses in vitro, with cells positioned along gradients of activation. Clonal identity influenced resting and activated transcriptional profiles but intriguingly was not associated with the capacity to produce IL-17. Overall, MAIT cells show phenotypic and functional diversity according to tissue localization, stimulation environment and clonotype.

Mucosal-associated invariant T (MAIT) cells are innate-like T cells, abundant in human blood and tissues, particularly the liver and mucosa[1]. MAIT cells express semi-invariant Vα7.2-Jα33/12/20 (*TRAV1-2-TRAJ33/12/20*) T cell receptors (TCRs) specific for microbial riboflavin metabolites presented by MR1 (ref. 1). They can also be activated independent of their TCR by cytokines such as IL-12 and IL-18 (ref. 2). Upon activation, MAIT cells secrete type 1/17 cytokines and exhibit cytotoxic activity[1].

A major outstanding question is whether human MAIT cells comprise transcriptionally and functionally distinct subsets. Alterations in frequency, phenotype and function occur in numerous human diseases, and mouse models indicate protective and pathogenic roles[3]. Understanding the characteristics of the MAIT cell population in health could aid the development of therapeutics targeting specific subsets or functions in disease.

In human blood, MAIT cells are relatively homogeneous, exhibiting a predominantly CD8[+] effector-memory phenotype, and characteristic expression of surface molecules (for example, CD161) and transcription factors (for example, PLZF and RORγt)[1]. However, there is some variability, for example, between CD8[+], CD4[−]CD8[−] (DN) and CD4[+] cells[4–6], and in the expression of innate immune receptors[6,7]. Despite universal RORγt expression, <5% of human MAIT cells produce IL-17 ex vivo[8,9]. This could reflect a committed type 17 subset, as in mice[10–12]. Alternatively, all human MAIT cells may have the capacity to produce IL-17 under appropriate conditions. Conclusive data addressing these competing hypotheses are lacking.

[1]Translational Gastroenterology Unit, Nuffield Department of Medicine, University of Oxford, Oxford, UK. [2]Department of Biomedicine, Liver Immunology, University Hospital Basel and University of Basel, Basel, Switzerland. [3]Division of Visceral Surgery, Clarunis University Center for Gastrointestinal and Liver Diseases, Basel, Switzerland. [4]Gastroenterology and Hepatology, University Department of Medicine, Cantonal Hospital Baselland, Liestal, Switzerland. [5]Peter Medawar Building for Pathogen Research, University of Oxford, Oxford, UK. [6]NIHR Oxford Biomedical Research Centre, John Radcliffe Hospital, Oxford, UK. [7]These authors contributed equally: Nicholas M. Provine, Paul Klenerman. ✉e-mail: lucy.garner@ndm.ox.ac.uk; paul.klenerman@ndm.ox.ac.uk

MAIT cell function is altered by tissue localization and stimulation. Compared with blood, gut and liver MAIT cells display an activated, tissue-resident transcriptome[12–14]. Genital tract[15] and oral mucosal[16] cells show type 17 skewing. MAIT cells exhibit distinct transcriptional responses to TCR and cytokine stimulation[17,18] and produce increased IL-17 upon sustained stimulation[9]. Whether functional diversity indicates the presence of multiple subsets or environment-driven plasticity remains unknown.

In addition, questions remain regarding MAIT cell TCR repertoires, including variability across tissues and donors, and the relationship between TCR usage and function. Studies variably demonstrate similar TCR repertoires across tissues[19,20] or differences in *TRAJ*/*TRBV* usage[16,21,22]. Diverse chain usage could have functional implications. For example, MAIT cell clonal distribution changes during human *Salmonella* Paratyphi A infection−cells transduced with TCRβ chains from expanded and contracted clonotypes show greater and lesser responses to TCR stimulation, respectively[23]. In vitro studies show differential activation potential dependent on clonotype[24] or TCRβ usage[7,25]. Thus, a relationship between TCR architecture and function is suggested but has not been studied systematically.

Overall, human MAIT cells show variation in phenotype, function and TCR repertoire. However, it is unknown whether they comprise multiple functionally distinct subsets, and how phenotype and function relate to TCR usage. To investigate this, we analyzed the single-cell transcriptome and TCR repertoire of human MAIT cells from matched blood and liver, as well as blood cells at rest and following TCR, cytokine or dual TCR+cytokine, stimulation. Our findings revealed a largely homogenous transcriptional program at rest, with variation linked to tissue localization. Activation triggered a plastic, stimulus-specific, effector program. The TCR repertoire was surprisingly diverse, and clonal identity influenced the transcriptome of resting and activated MAIT cells. Following dual stimulation, we identified an IL-17-expressing cluster. IL-17$^+$ cells expressed other effector molecules, such as *IFNG*, and showed similar TCR usage to IL-17$^-$ cells, suggesting they reflect an activation state rather than a bona fide MAIT17 lineage.

## Results

### MAIT cells show tissue-specific transcription and regulation

To investigate heterogeneity using an unbiased genome-wide approach, we performed single-cell RNA-sequencing (scRNA-seq) and single-cell TCR-sequencing (scTCR-seq) of sorted MAIT cells (CD3$^+$MR1/5-OP-RU$^+$) from matched human blood and liver (Supplementary Table 1 and Supplementary Fig. 1). Findings from an initial experiment (Exp 1; $n$ = 3 blood, 4 liver) were validated in a second experiment (Exp 2; $n$ = 8 blood, 3 liver). Conventional memory T (T$_{mem}$; CD3$^+$MR1/5-OP-RU$^-$CCR7$^-$) cells were analyzed in some donors (Exp 1: $n$ = 2 blood, 2 liver; Exp 2: all samples). Exp 2 included 130 oligo-conjugated antibodies for single-cell surface protein analysis.

After filtering, Exp 1 and 2 comprised 89,456 cells. MAIT and T$_{mem}$ cells, and blood and liver cells, were transcriptionally distinct (Fig. 1a,b). Blood T cells from liver donors (who underwent surgery for removal of benign or malignant lesions) were comparable to T cells from healthy donors (Fig. 1b), suggesting our data are not reflective of disease. CD4$^+$ T$_{mem}$ cells localized to distinct clusters, while rare CD4$^+$ MAIT cells were distributed throughout the UMAP (Extended Data Fig. 1a,b). MAIT and T$_{mem}$ cells differentially expressed 532 and 558 genes, and 37 and 33 proteins, in blood and liver, respectively (Supplementary Table 2a–d). Differences in gene and protein expression were highly correlated between tissues (Fig. 1c and Extended Data Fig. 1c–g). We defined core MAIT cell signatures of 167 genes (including *KLRB1* and *SLC4A10*) and 11 proteins (including Vα7.2 TCRα and CD161; Supplementary Table 2e).

MAIT cells from six matched blood-liver pairs (35,407 cells) comprised 11 clusters (Fig. 1d). Clusters largely contained cells from one tissue, but multiple experiments and donors (Fig. 1e,f and Extended Data Fig. 1h,i). Blood and liver cells differentially expressed 566 genes

(most upregulated in the liver) and 24 proteins (Supplementary Table 3a,b). Liver-enriched genes encoded tissue-residency markers (for example, *ITGAE*), TCR-induced transcription factors (for example, *EGR1*), effector cytokines (for example, *IFNG*) and chemokines/ chemokine receptors (for example, *CXCR6*). Some genes showed uniformly higher expression in the liver compared with the blood (for example, *CD69*; Fig. 1g); others were enriched in specific clusters (for example, *CCL3*; Fig. 1h). Interestingly, 84/167 core MAIT cell genes were upregulated in the liver (for example, *RORA* and *IL23R*). Liver-upregulated proteins included CD69 and CD244 (2B4), markers of tissue residency and cell activation (Fig. 1i and Extended Data Fig. 1j,k). T$_{mem}$ cells showed similar tissue imprinting (Supplementary Table 3c,d). We defined core liver signatures of 300 genes and eight proteins (Supplementary Table 3e). Core liver proteins included canonical markers of tissue-resident memory T (T$_{RM}$) cells[26] (for example, CD69 and CD103) and ICAM1, required for MAIT1 retention in mouse liver[12]. Using gene set enrichment analysis, we demonstrated enrichment of human and mouse T$_{RM}$ cell gene signatures[27,28] in liver MAIT (Fig. 1j) and T$_{mem}$ (Extended Data Fig. 1l) cells. Other liver-enriched pathways related to cell activation, cell adhesion and inflammation (Fig. 1k and Extended Data Fig. 1m).

SCENIC[29] was used to discover tissue-specific MAIT cell transcription factor regulons−modules of genes predicted to be regulated by a given transcription factor. Due to batch effects, cells from Exp 1 and Exp 2 were analyzed separately. Compared with blood, liver MAIT cells showed increased activity of AP-1 (for example, FOS and JUN) and NF-κB (for example, NFKB1 and NFKB2) transcription factors, and the TCR-induced transcription factor EGR1 (Fig. 1l, Extended Data Fig. 1n and Supplementary Table 4a,b). RUNX3 (regulates CD8$^+$ T$_{RM}$ cell differentiation[28]) activity was also increased. AP-1- and NF-κB-regulated genes were enriched for pathways associated with T cell activation, inflammation and cytokine production (Extended Data Fig. 1o−q).

In summary, MAIT cells in blood and liver are transcriptionally distinct. MAIT and T$_{mem}$ cells show similar adaptation to the liver environment and MAIT cell signature genes are consistent across tissues. We identify AP-1 and NF-κB transcription factors as central regulators of MAIT cell liver-specific gene expression.

### MAIT cells have a limited TCRα but diverse TCRβ repertoire

We next investigated whether TCR repertoires were tissue-specific. Previous analyses were limited by scale or depth[20,21,30]. Our dataset of >30,000 paired TCRs from 12 donors and matched tissues provided a unique opportunity to examine TCR repertoire characteristics and diversity.

Broad characteristics of the TCR repertoire were comparable in blood and liver. *TRAJ33*, *TRAJ12* and *TRAJ20* were used by 87%, 6% and 6% of TCRs, respectively (Extended Data Fig. 2a). The CDR3α region was highly restricted in length and sequence and included the canonical Tyr95α residue[31,32] (Extended Data Fig. 2b,c). CDR3α sequence, and the number of N-nucleotides and P-nucleotides, varied with *TRAJ* gene usage (Extended Data Fig. 2c−e). *TRBV* expression was diverse but biased toward *TRBV6-1*, *TRBV6-4* and *TRBV20-1* (Extended Data Fig. 2f). CDR3β length and sequence were highly variable (Extended Data Fig. 2g,h).

Studies of small numbers of TCR sequences suggest *TRAJ* usage could influence TCRαβ pairing[21,30]. Our data revealed increased pairing of *TRAJ12* and *TRAJ20* TCRα chains with *TRBV6-4* TCRβ chains compared with *TRAJ33*, while *TRAJ33* TCRα chains more frequently paired with *TRBV20-1* TCRβ chains (Fig. 2a).

TCR clonotypes were defined as cells with identical TCR gene segment usage and CDR3 nucleotide sequences (*TRAV1-2* TCRα required for MAIT cells). MAIT cells were oligoclonal, with oligoclonality comparable across donors and tissues (Fig. 2b) and with T$_{mem}$ cells (Fig. 2c). MAIT cell clonotypes defined using only the TCRα chain (TCRα clonotypes) were more oligoclonal than those defined using

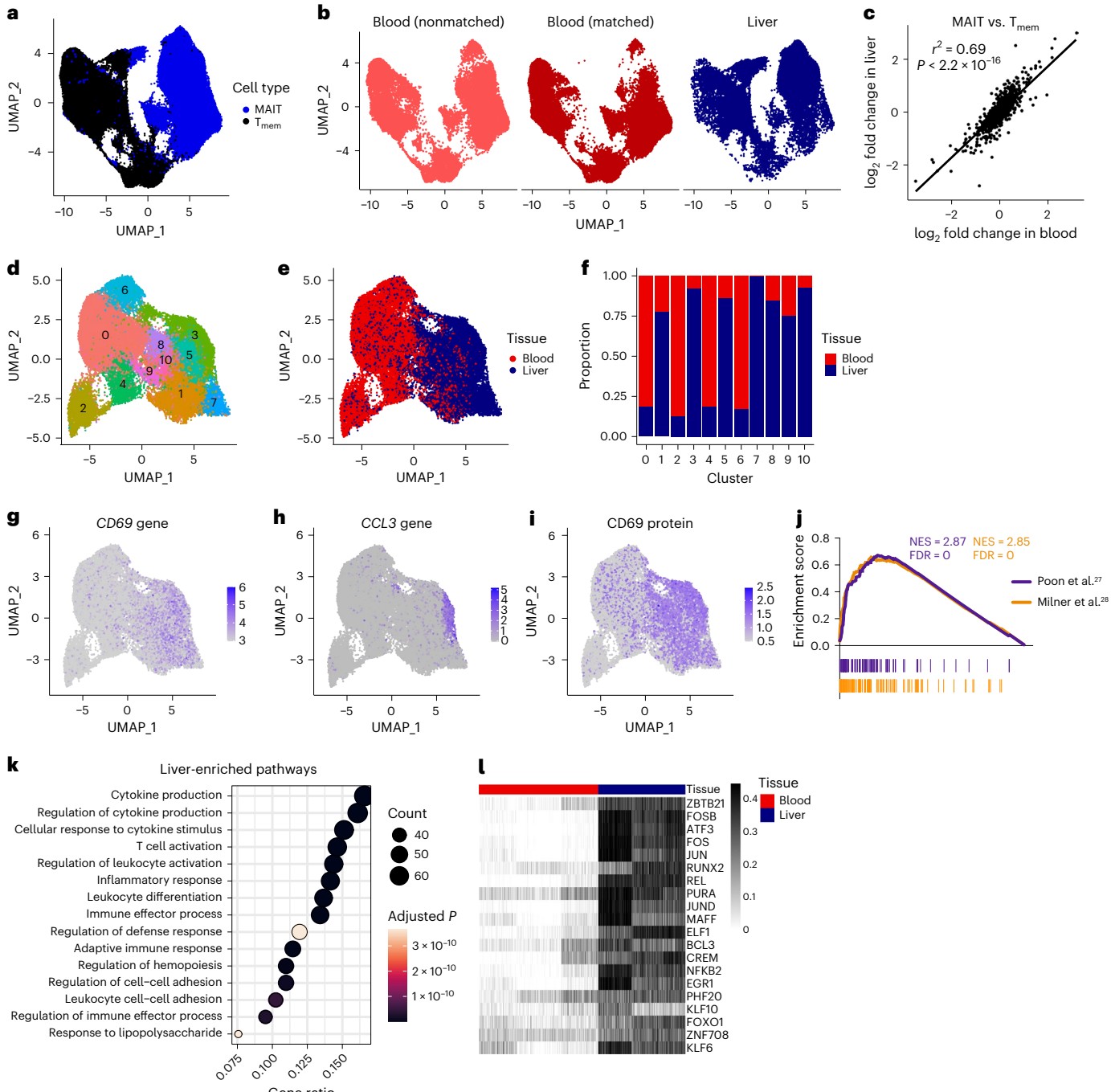

**Fig. 1 | Liver MAIT cells exhibit an activated, tissue-resident transcriptional and regulatory profile. a**, UMAP of blood and liver MAIT cells and conventional memory T ($T_{mem}$) cells colored by cell type. $n = 89,456$ cells from 12 donors. **b**, UMAP split by sample type, namely blood from healthy donors (nonmatched), blood from liver donors (matched) and liver. **c**, Pearson's correlation between the $\log_2$ fold change in gene expression between MAIT and $T_{mem}$ cells in the blood, and MAIT and $T_{mem}$ cells in the liver. **d,e**, UMAP of matched blood and liver MAIT cells ($n = 35,407$ cells from six donors) colored by the 11 identified clusters (**d**) or by tissue (**e**). **f**, Proportion of cells in each cluster from the blood and liver.

**g–i**, UMAPs colored by expression of *CD69* (**g**) and *CCL3* (**h**) genes or CD69 protein (**i**). **j**, Gene set enrichment analysis of liver compared with blood MAIT cells using published human and mouse tissue-resident memory T cell gene signatures. NES, normalized enrichment score. **k**, Over-representation analysis on the genes significantly upregulated in liver MAIT cells compared with blood MAIT cells. Top 15 gene ontology terms and associated Benjamini−Hochberg adjusted *P* values are shown. **l**, Heatmap showing activity (row-scaled AUCell scores) of the 20 most differentially active regulons (largest difference in average AUCell score) between matched blood and liver MAIT cells in Exp 1. $n = 3$ donors.

only the TCRβ chain (TCRβ clonotypes; Fig. 2d). At the population level, MAIT cell TCRα chain pairing was promiscuous, with ~30% of TCRα chains paired with >1 unique TCRβ−generating multiple clones with identical TCRα chains (Fig. 2e). Conversely, most TCRβ chains paired

with a single TCRα (Extended Data Fig. 2i). $T_{mem}$ cell TCRαβ pairings were essentially unique (Fig. 2f and Extended Data Fig. 2j).

Overall, the MAIT cell TCRα repertoire is highly restricted, while the TCRβ repertoire is considerably more diverse. TCR repertoire

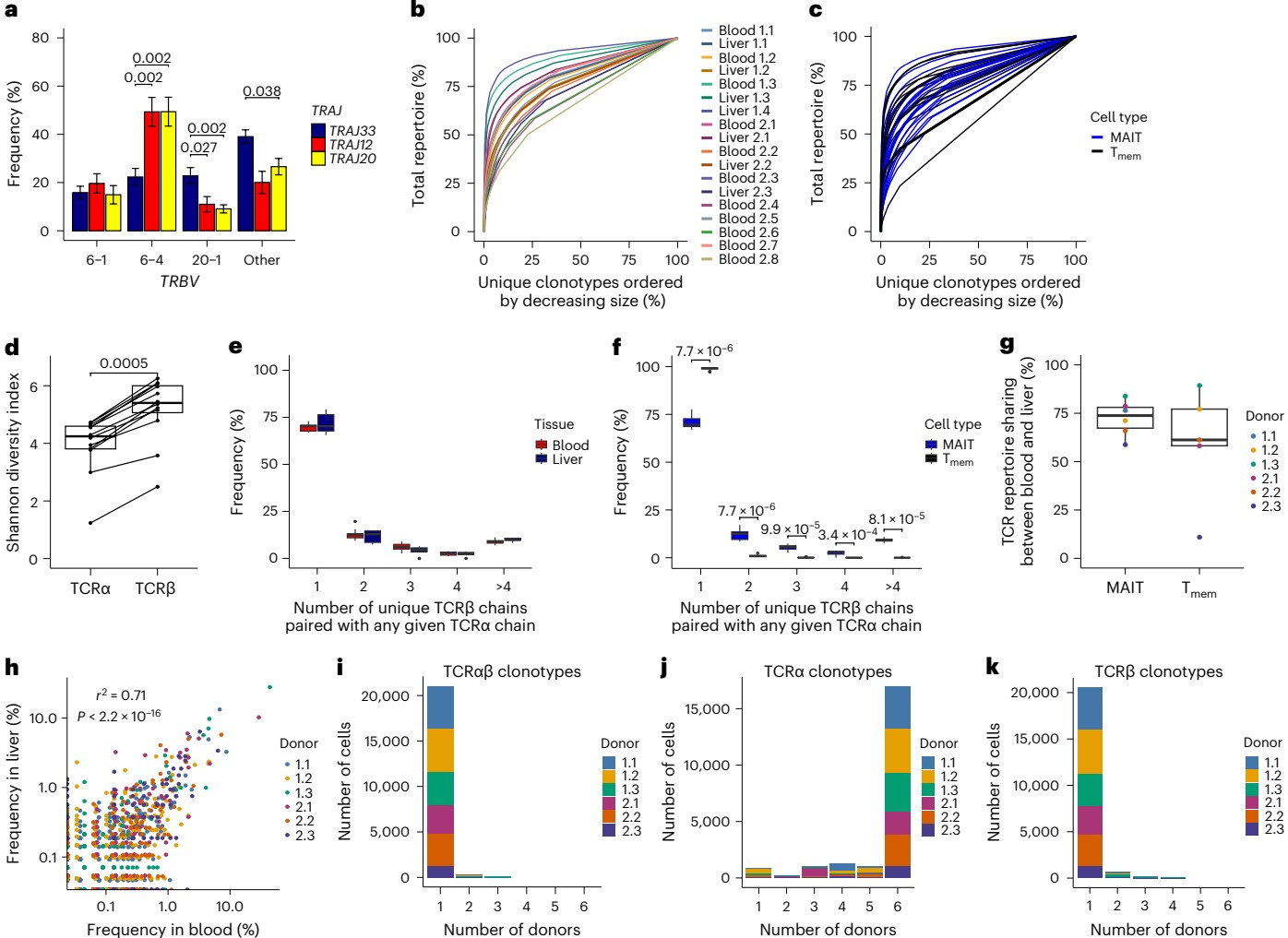

**Fig. 2 | MAIT cells have a restricted TCRα but diverse TCRβ chain, resulting in private TCRαβ repertoires. a**, Percentage of *TRAJ33*, *TRAJ12* and *TRAJ20* TCRα chains paired with *TRBV6-1*, *TRBV6-4*, *TRBV20-1* or other TCRβ chains. Mean ± s.e.m. is shown. **b**, Line plot colored by sample demonstrating the clonality of the TCRαβ repertoire. **c**, Line plot comparing MAIT (blue) and T_mem (black) cell TCRαβ clonality. **d**, Shannon diversity index for TCRα clonotypes and TCRβ clonotypes for each donor. **e,f**, TCR chain pairing at the population level. Number of unique TCRβ chains paired with any given TCRα chain in blood and liver MAIT cells (**e**) or MAIT and T_mem cells (**f**; blood and liver cells combined). **g**, Percentage of MAIT and T_mem cells belonging to a TCRαβ clonotype shared between matched blood and liver. **h**, Pearson's correlation between clonotype frequency in matched blood and liver. **i–k**, Number of cells from each donor belonging to TCRαβ (**i**), TCRα (**j**) or TCRβ (**k**) functional clonotypes found in 1, 2, 3, 4, 5 or 6 (all) donors. Plots show TCR data for MAIT cells (or MAIT and T_mem cells in **c**, **f** and **g**) from all donors (**a–f**; *n* = 12) or matched blood-liver donors (**g–k**; *n* = 6). In **d–g**, boxes span the 25th–75th percentiles, the midline denotes the median and whiskers extend to ±1.5 × IQR. Points in **e** and **f** indicate outliers. Two-sided Wilcoxon signed-rank test (**a** and **d**) and two-sided Wilcoxon rank-sum test (**e–g**) for all pairwise comparisons. Benjamini–Hochberg adjusted *P* values are shown (nonsignificant results omitted).

characteristics were similar in blood and liver, and we identified a preference for *TRAJ12/20-TRBV6-4* and *TRAJ33-TRBV20-1* pairings.

**TCR repertoires are donor-specific but shared across tissues**
As TCR usage was similar across tissues, we hypothesized that blood and liver MAIT cells might show clonal overlap. On average, 72% of MAIT cells belonged to a TCR clonotype present in matched blood and liver (Fig. 2g) and clonotype frequency correlated between tissues (Fig. 2h). Clonal sharing was similarly high for T_mem cells (Fig. 2g). The extent of MAIT cell TCR repertoire overlap between tissues correlated with TCR capture rate, suggesting our data underestimates blood-liver sharing (Extended Data Fig. 2k).

We next examined repertoire overlap between donors, with clonotypes defined using CDR3 amino acid sequences (functional clonotypes). Despite their semi-invariant TCR, 98% of MAIT cells belonged

to a donor-specific clonotype, with no clonotypes shared between all six donors (Fig. 2i). T_mem cell functional clonotypes showed no overlap between donors (Extended Data Fig. 2l).

Given the restricted MAIT cell TCRα repertoire, we reasoned that functional clonotypes defined using the TCRα chain only (functional TCRα clonotypes) may show high overlap between donors. Supporting this hypothesis, the six donors shared 27 functional TCRα clonotypes comprising 79% (16,973/21,421) of MAIT cells (Fig. 2j). In contrast, 1.1% (196/18,564) and <0.1% of T_mem cells belonged to a functional TCRα clonotype found in two and three donors, respectively (Extended Data Fig. 2m). Functional TCRβ clonotypes were largely donor-specific for MAIT and T_mem cells (Fig. 2k and Extended Data Fig. 2n).

Thus, distinct from T_mem cells, the MAIT cell TCRα repertoire is public. In contrast, the TCRβ chain is markedly more private and is what governs the uniqueness of individual MAIT cell TCR repertoires.

## Within-tissue transcriptional heterogeneity is limited

We next explored within-tissue heterogeneity.

Blood MAIT cells comprised nine clusters (Fig. 3a and Extended Data Fig. 3a). Transcriptional diversity between clusters was low, with few genes displaying cluster-specific expression (Fig. 3b and Supplementary Table 5a). Apart from the three clusters discussed below, cluster markers were not indicative of specific functions or known T cell differentiation states. Using mouse MAIT1 and MAIT17 (refs. 33,34) or human Th1 and Th17 (ref. 35) gene signatures, we were unable to identify human MAIT1 and MAIT17 subsets (Fig. 3c and Extended Data Fig. 3b). *CCL4* was upregulated in cluster 6 and interferon-stimulated genes in cluster 8 (Fig. 3b and Extended Data Fig. 3c), perhaps indicating some degree of basal cell activation. Cells in cluster 2 appeared primed for cytotoxicity with increased expression of granulysin and granzymes (Fig. 3d and Extended Data Fig. 3d). This cluster did not simply indicate cell activation, as *GZMB* and *GZMH* (lowly expressed in resting MAIT cells[36]) were only expressed by a small percentage of cells.

Liver MAIT cells comprised ten clusters (Fig. 3e and Extended Data Fig. 3e) with modest transcriptional differences (Extended Data Fig. 3f and Supplementary Table 5b). Clusters did not correspond to known differentiation states such as MAIT1 and MAIT17 (Extended Data Fig. 3g). As in blood, there was a *GNLY*-expressing cluster (cluster 6; Fig. 3f), while remaining clusters expressed different activation- or stress-induced molecules (Fig. 3g). Most genes showed a gradient of expression across clusters.

Despite reported phenotypic, functional and/or transcriptional differences[4–6], CD8+, DN and CD4+ MAIT cells did not comprise separate clusters in blood or liver (Fig. 3h,i and Extended Data Fig. 3h–k), and differentially expressed few genes and proteins (Supplementary Table 6a–h). Previous bulk RNA-seq data indicated higher TCR repertoire diversity in CD8+ relative to DN MAIT cells[5]. After downsampling to equalize numbers of CD8+ and DN cells per donor, we identified a small increase in the frequency of unique clonotypes among CD8+ MAIT cells (Fig. 3j), but no difference in *TRAJ/TRBV* chain usage (Extended Data Fig. 3l,m), and an equivalent Shannon diversity index (Fig. 3k). Therefore, consistent with minor transcriptional differences, CD8+ and DN MAIT cells show similar TCR usage.

Overall, contrasting with mice, human MAIT cells show limited transcriptional heterogeneity within tissues and do not comprise distinct MAIT1 and MAIT17 subsets, or subsets defined by coreceptor expression.

## TCR clonotypes show variable bias in cluster localization

We next examined whether the limited transcriptional heterogeneity within tissues correlated with clonal identity. Given the donor-specific private TCR repertoire, we separately clustered each sample, then used the exact multinomial test to determine whether clonotypes were nonrandomly distributed across clusters.

There was a range of associations between clonotype and cluster, both within and between donors. Some clonotypes predominantly localized in a single cluster (Fig. 4a and Extended Data Fig. 4a). Some showed a subtle but still significant bias in cluster localization (Fig. 4b and Extended Data Fig. 4a). Others were randomly distributed

(Fig. 4c and Extended Data Fig. 4a). Bias in cluster distribution was more frequently significant for larger clonotypes (Fig. 4d), suggesting nonsignificant results for some smaller clonotypes may reflect a lack of statistical power.

For a given clonotype, the extent of bias in cluster localization was not necessarily concordant in blood and liver (Fig. 4a,e and Extended Data Fig. 4a,b). This may be due in part to different clonotype sizes in the two tissues. Some clonotypes had a stable transcriptional phenotype across tissues. For example, a *TRAV1-2/TRAJ12* clonotype from donor 1.2 preferentially localized to the *GNLY*-expressing cluster in blood (Fig. 4f,g and Extended Data Fig. 4c) and liver (Fig. 4h,i and Extended Data Fig. 4d). Therefore, the transcriptional profile of resting MAIT cells is influenced by their clonal identity.

## MAIT cell functional diversity is stimulus-specific

Since we did not identify subsets of resting MAIT cells, we investigated whether functional subsets were present following activation. CD8+ T cells were left unstimulated or stimulated with MR1/5-OP-RU (TCR) or IL-12 + IL-18 (cytokine). After 20 h, CD8+ MAIT cells were sorted (CD26+CD161hiVα7.2+ for unstimulated and cytokine-stimulated, and CD26+CD161hi for TCR-stimulated due to TCR downregulation) for scRNA-seq and scTCR-seq (Exp 3; Supplementary Fig. 2), an approach we termed functional RNA-sequencing (fRNA-seq).

fRNA-seq revealed stimulus-specific transcriptional responses (Fig. 5a and Supplementary Table 7a,b). MAIT cells (27,305 cells) comprised nine clusters—these were present in all donors but largely stimulus-specific (Fig. 5b,c and Extended Data Fig. 5a). Consistent with their homogeneous resting transcriptome, unstimulated cells predominantly localized in one cluster. TCR-stimulated cells localized in clusters 1 and 4. Cells in cluster 1 were more activated than those in cluster 4, displaying increased expression of chemokines and cytokines including *CCL4*, *TNF* and *CSF2* (Fig. 5d,e, Extended Data Fig. 5b–d and Supplementary Table 8). Clusters 2, 3 and 5 largely comprised cytokine-stimulated cells and appeared to indicate different degrees of cell activation. Expression of activation markers (for example, *IL2RA*) and effector molecules (for example, *GZMB*) was low in cluster 3, but high in cluster 5 (Fig. 5d,e, Extended Data Fig. 5b,e and Supplementary Table 8). Cells in cluster 2 expressed high levels of *IFNG* but less *GZMB* than cells in cluster 5. Interferon-stimulated genes were uniquely expressed in cluster 7 (mostly cytokine-stimulated; Fig. 5d and Extended Data Fig. 5b).

As in Exp 1 and 2, we identified a *GNLY*-expressing cluster (cluster 6; Fig. 5d and Extended Data Fig. 5f) that contained cells from all three conditions (Fig. 5c). The cells did not express other cytotoxic molecules (for example, *GZMB*) or markers of activation, inconsistent with our initial hypothesis that *GNLY*-expressing cells at rest are primed for cytotoxicity.

It was suggested that CD56 expression identifies a MAIT cell subset with enhanced cytokine responsiveness[7]. Following cytokine stimulation, CD56 (*NCAM1*)-expressing MAIT cells showed increased *IFNG* production relative to their nonexpressing counterparts (Extended Data Fig. 5g). However, as CD56 expression was qualitatively increased following cytokine stimulation (Extended Data Fig. 5h), we could not establish the usefulness of CD56 as a baseline indicator of functional

---

**Fig. 3 | MAIT cells within the blood and liver show minimal transcriptional heterogeneity. a**, UMAP of blood MAIT cells from matched blood-liver donors (*n* = 6) colored by the nine identified clusters. **b**, Heatmap showing row-scaled log-transformed normalized expression of the top five or all (if <5) marker genes for each blood MAIT cell cluster. **c**, Expression of MAIT1 and MAIT17 genes in blood MAIT cell clusters. Dot color indicates the level of gene expression and dot size indicates the percentage of cells expressing the gene. **d**, UMAPs of blood MAIT cells colored by expression of *GNLY*, *GZMB* and *GZMH*. **e**, UMAP of liver MAIT cells from matched blood-liver donors (*n* = 6) colored by the ten identified clusters. **f**, UMAP of liver MAIT cells colored by expression of *GNLY*. **g**, UMAPs

of liver MAIT cells colored by expression of *IFNG*, *TNF*, *CCL3*, *CCL4*, *HSPA1A* and *EGR1*. **h,i**, Proportion of CD4+, CD8+ and DN cells in each blood (**h**) and liver (**i**) cluster. Coreceptor identity defined by the expression of *CD4*, *CD8A* and *CD8B* genes (Methods). **j**, Number of unique clonotypes in CD8+ and DN MAIT cells from each donor (*n* = 12; CD8+ cell number within each donor downsampled to match the number of DN cells). **k**, Shannon diversity index for TCRαβ clonotypes in CD8+ and DN MAIT cells from each donor (*n* = 12). Boxes span the 25th–75th percentiles, the midline denotes the median and whiskers extend to ±1.5 × IQR. Two-sided Wilcoxon signed-rank test in **j** and **k** (nonsignificant results omitted).

potential. To address this, we stimulated sorted CD56⁻ and CD56⁺ MAIT cells with IL-12 + IL-18 for 20 h, and measured IFNγ, granzyme B, perforin and CD94 expression by flow cytometry. There was a trend toward increased expression of all tested markers in activated CD56⁺ cells relative to CD56⁻ cells (Extended Data Fig. 5i–l). Thus, CD56

expression correlates with MAIT cells primed for cytokine responsiveness. However, CD56⁺ cells did not comprise a transcriptionally distinct cluster of MAIT cells in resting blood or liver (Extended Data Fig. 5m,n). Further experiments are necessary to understand the overall impact of CD56 expression on MAIT cell biology.

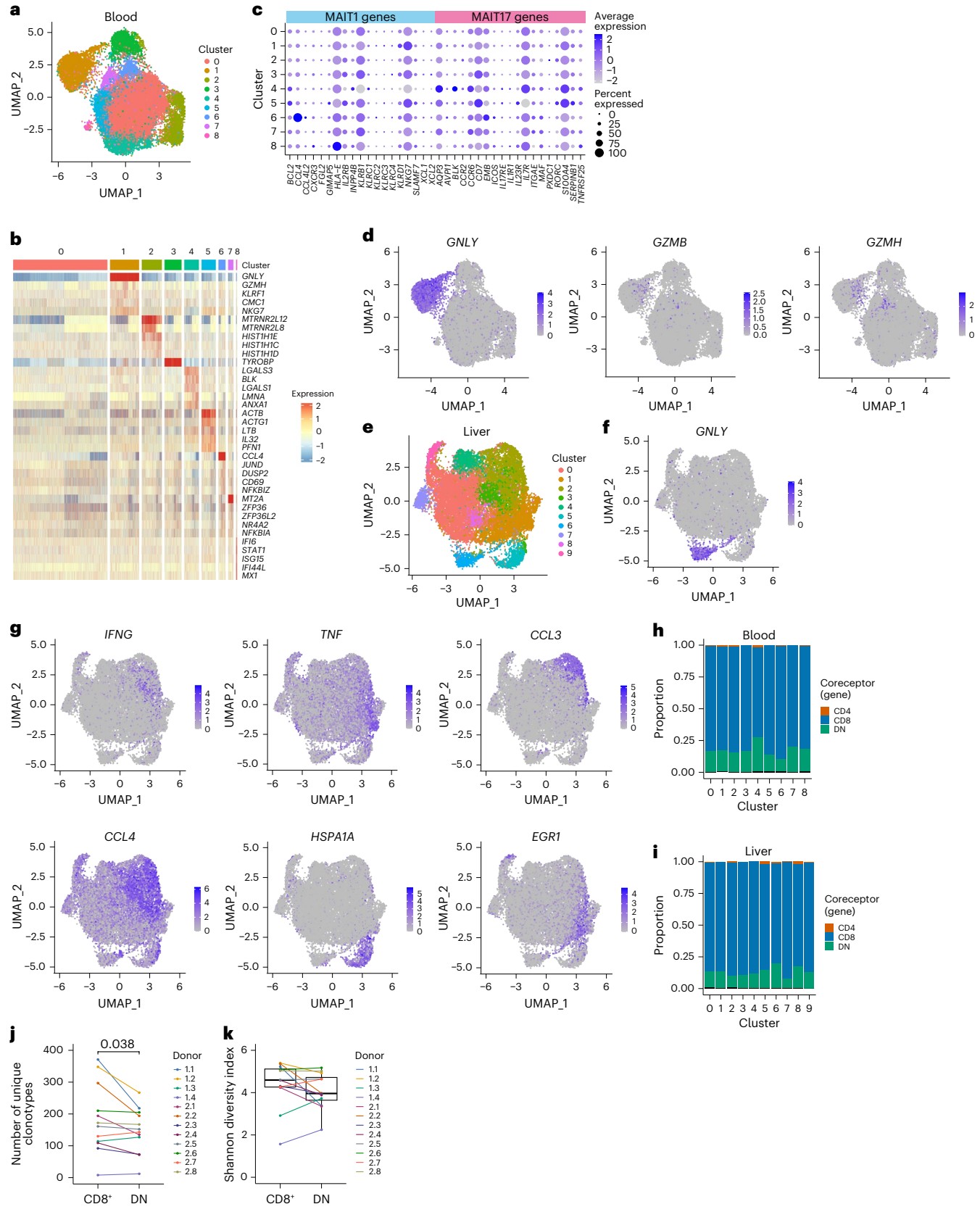

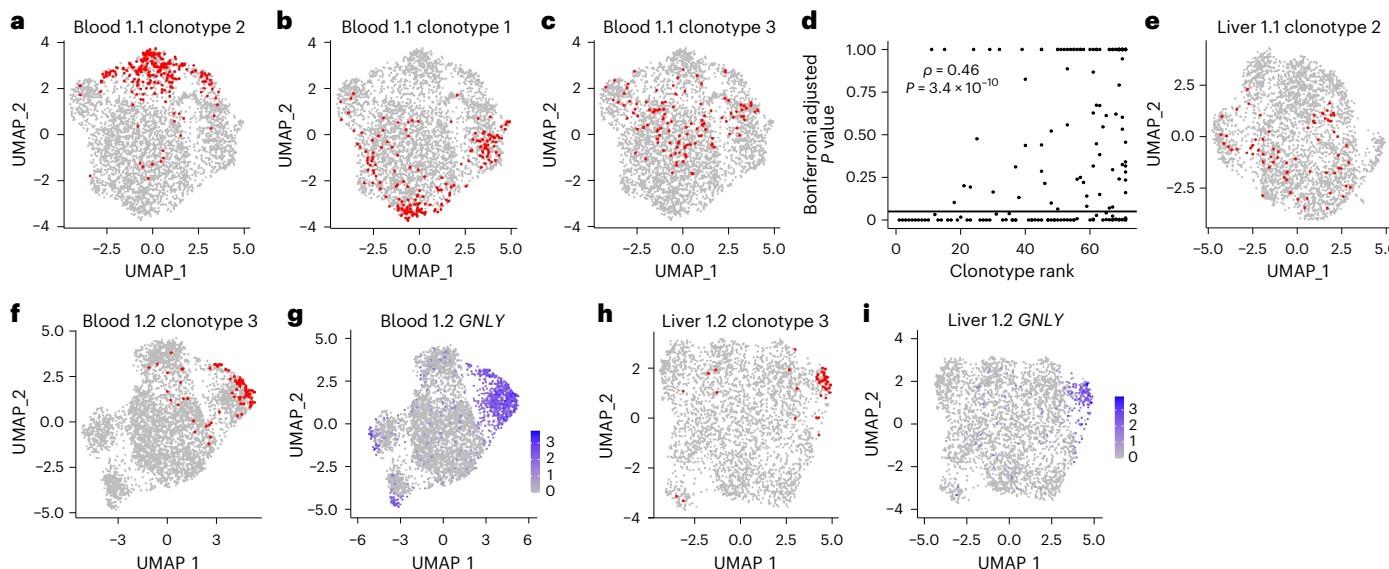

**Fig. 4 | TCRαβ clonotypes are variably associated with transcriptional clusters. a–c**, UMAPs of blood MAIT cells from donor 1.1 showing in red cells from a single TCRαβ clonotype. Plots show clonotype 2 (**a**), clonotype 1 (**b**) and clonotype 3 (**c**). Clonotype 1 is the largest clonotype from a donor, clonotype 2 is the second largest and so on. **d**, Spearman's rank correlation between clonotype size (rank) and the Bonferroni adjusted *P* value for association between clonotype and cluster (exact multinomial test, performed for clonotypes from all donors (*n* = 12) with ≥20 cells). **e**, UMAP of liver MAIT cells from donor 1.1 with cells from clonotype 2 indicated in red (the same clonotype as in **a**). **f,g**, UMAPs of blood MAIT cells from donor 1.2 with cells from clonotype 3 indicated in red (**f**) or expression of *GNLY* shown in blue (**g**). **h,i**, UMAPs of liver MAIT cells from donor 1.2 with cells from clonotype 3 indicated in red (**h**) or expression of *GNLY* shown in blue (**i**).

## Pseudotime analysis reveals linear activation trajectories

As MAIT cell clusters captured cells at different stages of activation, we further explored transcriptional responses to stimulation using pseudotime analysis. The Slingshot[37] algorithm identified a branching trajectory with a single branch point close to unstimulated cells (Fig. 5f), suggesting MAIT cells become transcriptionally distinct early following TCR and cytokine stimulation. Results were validated using SCORPIUS[38] (Extended Data Fig. 6a,b).

Through random forest regression, we identified the genes most important for predicting cell pseudotime on the TCR and cytokine trajectories (Supplementary Table 9). Gene importance was highly correlated between the two trajectories (Fig. 5g), with nine of the top 20 genes overlapping, including *IL2RA* (CD25) and *TNFRSF18* (GITR; both upregulated) and *IL7R* (downregulated). However, several notable genes were important primarily for one trajectory (Fig. 5g–i). *IFNG* and *IL26* were specific to the cytokine trajectory, while *CCL3*, *CCL4* and *TNFRSF9* (4-1BB) showed greater importance for the TCR trajectory. Protein expression, as measured by flow cytometry, was consistent with gene expression (Extended Data Fig. 6c–k)—IFNγ and CD40L were more strongly induced by cytokine stimulation and CCL4, TNF, 4-1BB and CD25 by TCR stimulation. Granzyme B was similarly induced by both stimuli.

## Regulation of TCR- and cytokine-induced transcription

Using SCENIC[29], we identified 159 high-confidence regulons regulating shared and stimulus-specific gene expression. Although global changes in transcription factor activity relative to unstimulated cells were highly correlated for TCR- and cytokine-stimulated cells (Fig. 6a), some regulons showed markedly different activity between conditions.

Relative to unstimulated cells, 65 regulons had increased activity: 11 were TCR-specific, 22 were cytokine-specific and 32 were shared (Fig. 6b). TCR-specific regulons included TCR-induced transcription factors (EGR1, EGR2, NR4A1), and VDR, CREM and STAT5A, which have varied roles in regulating Th17 differentiation and IL-17 production[39–41] (Fig. 6b and Extended Data Fig. 7a). Cytokine-specific regulons

included STAT1, interferon regulatory factors, XBP1 and IKZF1 (Ikaros; Fig. 6b and Extended Data Fig. 7b). Ikaros regulates activated conventional CD8+ T cell responsiveness to IL-12 (ref. 42).

The 32 regulons with increased activity upon TCR and cytokine stimulation were also all differentially active between the two conditions. TBX21 activity was most similar between TCR- and cytokine-stimulated cells—its target genes were enriched for interleukin, Toll-like receptor and NF-κB signaling pathways (Extended Data Fig. 7c). As expected, NFATC1 and STAT4 showed increased activity in TCR- and cytokine-stimulated cells, respectively. We identified new candidate regulators of stimulus-specific MAIT cell functions, namely HIVEP3 for TCR-stimulated cells, and BATF, BCL6 and HIF1A for cytokine-stimulated cells (Fig. 6c–f). HIVEP3 is essential for the development of innate-like T cells including MAIT cells[43]. Among its many roles, BATF promotes effector CD8+ T cell differentiation through the upregulation of key transcription factors, cytokine receptors and signaling molecules[44].

As with upregulated genes, regulon activity varied across stimulated cells. Most regulons progressively increased in activity over pseudotime (Fig. 6g,h). However, activity of some regulons peaked early and subsequently declined, for example, EGR1 and NFKB1 on the TCR trajectory, and STAT1 and IRF1 on the cytokine trajectory. Several early-activated regulons regulated later-activated transcription factors. STAT1 was a predicted regulator of *TBX21*. HIVEP3 was a predicted target of EGR1, supporting its TCR-specific activation and function.

To identify candidate regulators of MAIT cell function, we examined effector gene localization within SCENIC regulons. Unsupervised hierarchical clustering identified two clusters of effector genes (Fig. 6i). Cluster 1 comprised genes preferentially induced by TCR signaling, namely *CSF2*, *TNF*, *CCL3* and *CCL4*, suggesting similar regulation. Genes in cluster 1 were regulated by EGR1, NFATC1 and MYC. In addition, HIVEP3 was a predicted regulator of *CSF2*, *CCL3* and *CCL4*. Cluster 2 comprised both cytokine-specific genes (for example, *IL26*) and genes induced by both stimuli (for example, *GZMB*). As expected, *IFNG* was

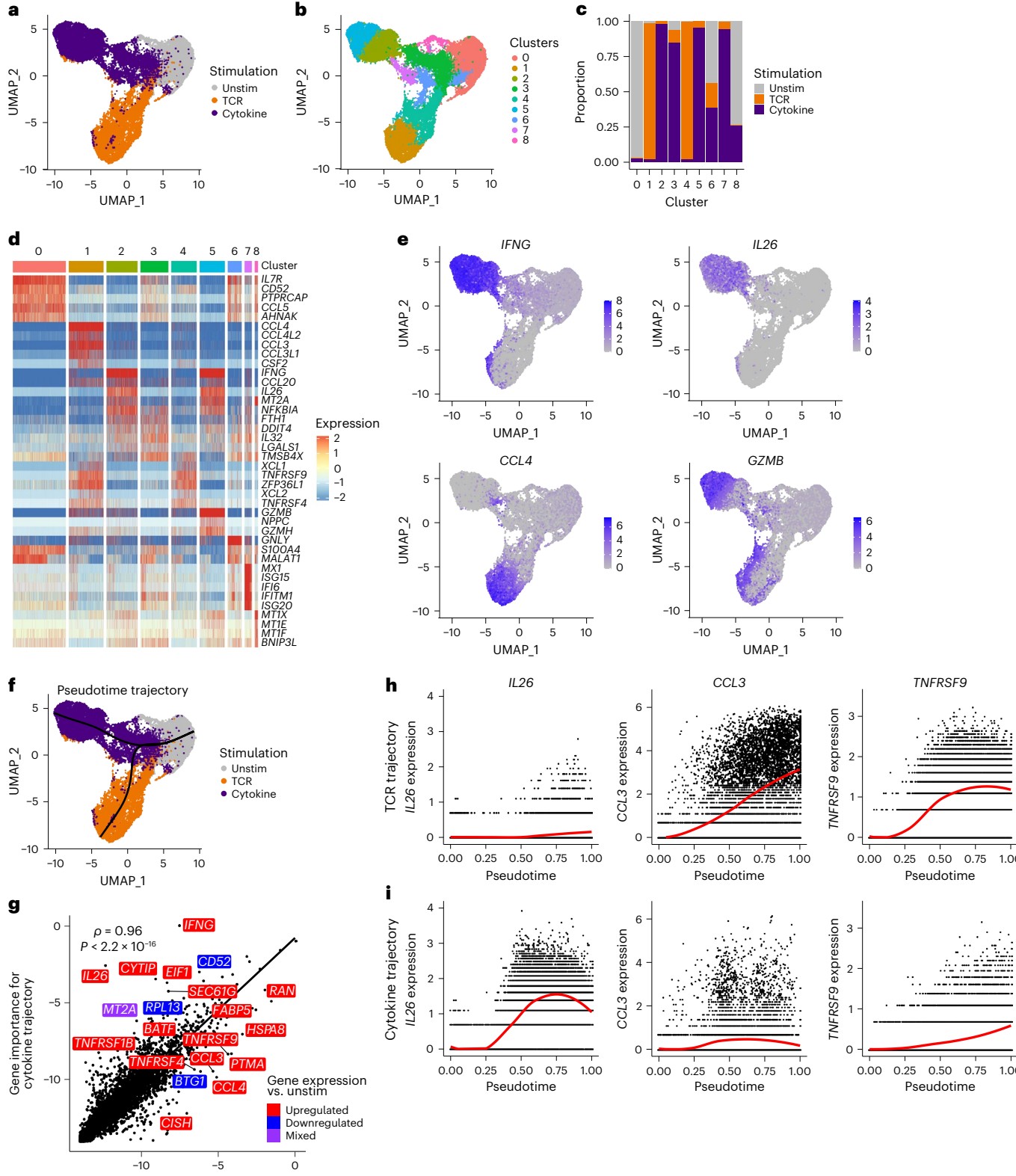

**Fig. 5 | TCR- and cytokine-activated MAIT cells follow distinct linear trajectories. a**,**b**, UMAPs of MAIT cells from all donors colored by stimulation condition (**a**) or the nine identified clusters (**b**). $n$ = 27,305 cells from three donors. **c**, Proportion of cells in each cluster from the three stimulation conditions. **d**, Heatmap showing row-scaled log-transformed normalized expression of the top five marker genes for each cluster. **e**, UMAPs colored by expression of *IFNG*, *IL26*, *CCL4* and *GZMB*. **f**, UMAP of MAIT cells from all donors with the branching pseudotime trajectory identified using Slingshot shown in

black. **g**, Spearman's rank correlation between gene importance (log₂ 1/gene importance rank) on SCORPIUS TCR and cytokine trajectories. Labels indicate the most differentially important genes, ten with higher importance on the TCR trajectory and ten with higher importance on the cytokine trajectory. Colors indicate whether gene expression was upregulated (red), downregulated (blue) or mixed (purple; upregulated in TCR and downregulated in cytokine or vice versa) relative to unstimulated cells. **h**,**i**, Expression of *IL26*, *CCL3* and *TNFRSF9* along SCORPIUS TCR (**h**) and cytokine (**i**) trajectories.

regulated by STAT4, but was also in the BATF regulon, suggesting BATF could contribute to enhanced *IFNG* production in cytokine-stimulated MAIT cells compared with TCR-stimulated MAIT cells.

Overall, TCR- and cytokine-stimulated MAIT cells exhibit shared and stimulus-specific regulation. Our data reveal new candidate regulators of TCR- and cytokine-specific responses and their predicted target genes.

## Clonal identity influences MAIT cell activation potential

TCR clonotypes showed varied associations with resting transcriptional clusters (Fig. 4) and published data suggest functional differences linked to TCRβ usage[7,23,25]. Using fRNA-seq, we investigated whether MAIT cell activation potential (pseudotime position) correlated with TCRβ usage or clonal identity. Within donors, activation capacity was significantly associated with *TRBV* usage, but there was considerable variability among cells with the same *TRBV* gene (Fig. 7a,b and Extended Data Fig. 8a,b). *TRBV* pseudotimes did not correlate between donors (Fig. 7a,b and Extended Data Fig. 8c,d) or between TCR and cytokine trajectories (Fig. 7c), indicating no intrinsic difference in the activation potential of *TRBV* genes.

Therefore, we hypothesized that observed variation could reflect clonal differences. Consistent with this, activation capacity differed between clonotypes on the TCR and cytokine trajectory (Fig. 7d,e and Extended Data Fig. 8e,f). High variability within clonotypes (Fig. 7d,e) suggested additional major influences on MAIT cell activation capacity. Clonotype pseudotimes were not correlated in response to TCR and cytokine stimulation (Fig. 7f), and there was no consistent association between clonotype size and responsiveness to stimulation (Fig. 7g–i). Therefore, larger clones are not intrinsically more functional than smaller clones. Given that variation between clonotypes was observed on the cytokine trajectory as well as the TCR trajectory, differences in clonotype functionality may not solely be associated with the strength of TCR-ligand binding.

## IL-17⁻ and IL-17⁺ cells overlap in function and TCR usage

MAIT1 and MAIT17 subsets were not detected in blood or liver. However, due to minimal IL-17 production, our stimulation experiment (Exp 3) did not allow us to address the source and functionality of IL-17-producing human MAIT cells. Therefore, our second stimulation experiment (Exp 4) included a dual TCR+cytokine condition that induces enhanced IL-17 production[9]. Isolated T cells were left unstimulated or stimulated for 20 h with MR1/5-OP-RU (TCR), IL-12 + IL-18 (cytokine) or both, before MAIT cell sorting (Supplementary Fig. 3). For the TCR+cytokine condition, we performed an additional 3 d (68 h) stimulation, previously shown to increase IL-17 production[9].

We analyzed 96,867 and 42,765 MAIT cells from three donors at 20 h and 68 h, respectively. After 20 h, unstimulated and stimulated cells were phenotypically and transcriptionally distinct (Supplementary Table 10a–f), with TCR+cytokine-stimulated cells localizing between TCR and cytokine single-stimulated cells on the UMAP (Fig. 8a and Extended Data Fig. 9a). Likewise, TCR+cytokine-stimulated cells were distinct from unstimulated cells at 68 h (Fig. 8b, Extended Data Fig. 9b and Supplementary Table 10g,h).

At 20 h, *IFNG* and *IL26* were primarily upregulated following cytokine stimulation; *CCL3*, *CCL4* and *TNFRSF9* (and the corresponding 4-1BB protein) were specific to the TCR condition; and *GZMB* was similarly upregulated by both stimuli, consistent with Exp 3 (Extended Data Fig. 9c–e and Supplementary Table 10a,c). TCR+cytokine stimulation upregulated both TCR- and cytokine-specific genes (Supplementary Table 10e,g). In general, gene and protein expression were highly correlated in the single and dual stimulation conditions (Fig. 8c,d and Extended Data Fig. 9f,g). Notably, the expression of *IL17F* was significantly increased following dual relative to single stimulation (Fig. 8c,d). IL-17-expressing cells comprised a distinct cluster at both timepoints (Fig. 8e,f and Extended Data Fig. 9h–k). Three-day stimulation induced a higher fraction of IL-17⁺ MAIT cells compared with 1 d stimulation (Fig. 8g). *IL17F* was expressed by all cells in the cluster, while a small percentage produced *IL17A* (Extended Data Fig. 9i,k).

To investigate whether IL-17-expressing MAIT cells comprise a distinct subset, we examined transcriptional differences between IL-17⁻ and IL-17⁺ MAIT cells following dual stimulation. At 20 h, IL-17⁻ and IL-17⁺ cells differentially expressed 23 genes (Supplementary Table 11a). Along with *IL17A* and *IL17F*, IL-17-expressing cells showed increased expression of *CCR6* and *CCL20* but reduced *GZMB* and *KLRD1*. Nevertheless, IL-17⁺ cells expressed high levels of *GZMB* and *KLRD1* compared with unstimulated cells (Extended Data Fig. 10a). Expression of *IFNG*, *IL26*, *CCL3* and other effector molecules was comparable in IL-17⁻ and IL-17⁺ cells (Extended Data Fig. 9c). At 68 h, IL-17⁻ and IL-17⁺ cells differentially expressed 28 genes (Supplementary Table 11c), but again differences in effector gene expression were small and only five differentially expressed genes overlapped at the two timepoints. Protein analysis revealed similar findings—IL-17⁻ and IL-17⁺ cells differentially expressed three and six proteins at 20 h and 68 h, respectively (Supplementary Table 11b,d).

The similar transcriptional profiles of IL-17⁻ and IL-17⁺ cells, and the increased frequency of IL-17-expressing cells at 68 h relative to 20 h, suggested that IL-17⁺ cells may represent a functional state obtainable by all MAIT cells under appropriate stimulation conditions. To investigate this, we compared the TCR repertoire of IL-17⁻ and IL-17⁺ cells. Clonotype abundance strongly correlated between the two groups at both timepoints (Fig. 8h,i). The number of unique clonotypes among IL-17-expressing cells was increased at 68 h compared with 20 h (Fig. 8j), indicating that new cells become IL-17⁺. However, there was no difference in the number of unique clonotypes among IL-17⁻ and IL-17⁺ cells (Fig. 8k) or in the Shannon diversity index (Extended Data Fig. 10b) (data downsampled to ensure equivalent numbers of IL-17⁻ and IL-17⁺ cells within each donor at a given timepoint). Moreover, *TRAJ* and *TRBV* usage was comparable (Extended Data Fig. 10c,d).

To examine the regulation of IL-17 gene expression in MAIT cells, we generated a bulk ATAC-seq dataset (*n* = 3 donors) comprising three blood CD8⁺ T cell subsets: naïve T cells, MAIT cells and T_mem cells. As expected given their expression by resting MAIT cells[1], the type 17-associated genes *RORC*, *IL23R* and *CCR6* showed increased accessibility in MAIT cells compared with T_mem and naïve T cells (Extended Data Fig. 10e–g). In addition, MAIT cells showed increased accessibility of peaks associated with *IL17A* and *IL17F* (Fig. 8l,m).

**Fig. 6 | Transcriptional regulation of TCR- and cytokine-stimulated MAIT cells exhibits shared and distinct properties. a,** Pearson's correlation between the log₂ fold change in regulon activity (AUCell scores) between TCR-stimulated and unstimulated MAIT cells, and cytokine-stimulated and unstimulated MAIT cells. Labels show the regulons with the largest difference in log₂ fold change relative to unstimulated cells between the TCR and cytokine trajectory, ten with increased (red), ten with decreased (blue) and ten with mixed (purple; increased in TCR and decreased in cytokine or vice versa) activity following stimulation. **b,** Heatmap showing the activity (row-scaled average AUCell scores) of TCR-specific (orange), cytokine-specific (purple) and shared (maroon) upregulated regulons in each stimulation condition. **c–f,** UMAPs colored by the activity of HIVEP3 (**c**), BATF (**d**), BCL6 (**e**) and HIF1A (**f**) regulons. **g,** Heatmap showing regulon activity (smoothed AUCell scores) over pseudotime on the SCORPIUS TCR trajectory for regulons upregulated upon TCR stimulation. Gray, unstimulated cells; orange, TCR-stimulated cells. **h,** Heatmap showing regulon activity (smoothed AUCell scores) over pseudotime on the SCORPIUS cytokine trajectory for regulons upregulated upon cytokine stimulation. Gray, unstimulated cells; purple, cytokine-stimulated cells. **i,** Regulation of select MAIT cell effector genes. Heatmap is colored by the percent occurrence of each gene within each transcription factor regulon. High-confidence regulons predicted to regulate at least one of the genes in >50% of pySCENIC runs are included. Red asterisks in **g–i** indicate regulons mentioned in the text.

The two differentially accessible *IL17A* peaks were located at the promoter and in the upstream intergenic region, while the two *IL17F* peaks were in the downstream intergenic region. Compared with the *GZMB* promoter peak (Fig. 8n), the *IL17A* promoter peak was of lower magnitude, while *IL17F* lacked a peak at the promoter. Reduced accessibility of *IL17A* and *IL17F* promoters may explain delayed IL-17 secretion relative to rapid granzyme B upregulation following MAIT cell activation.

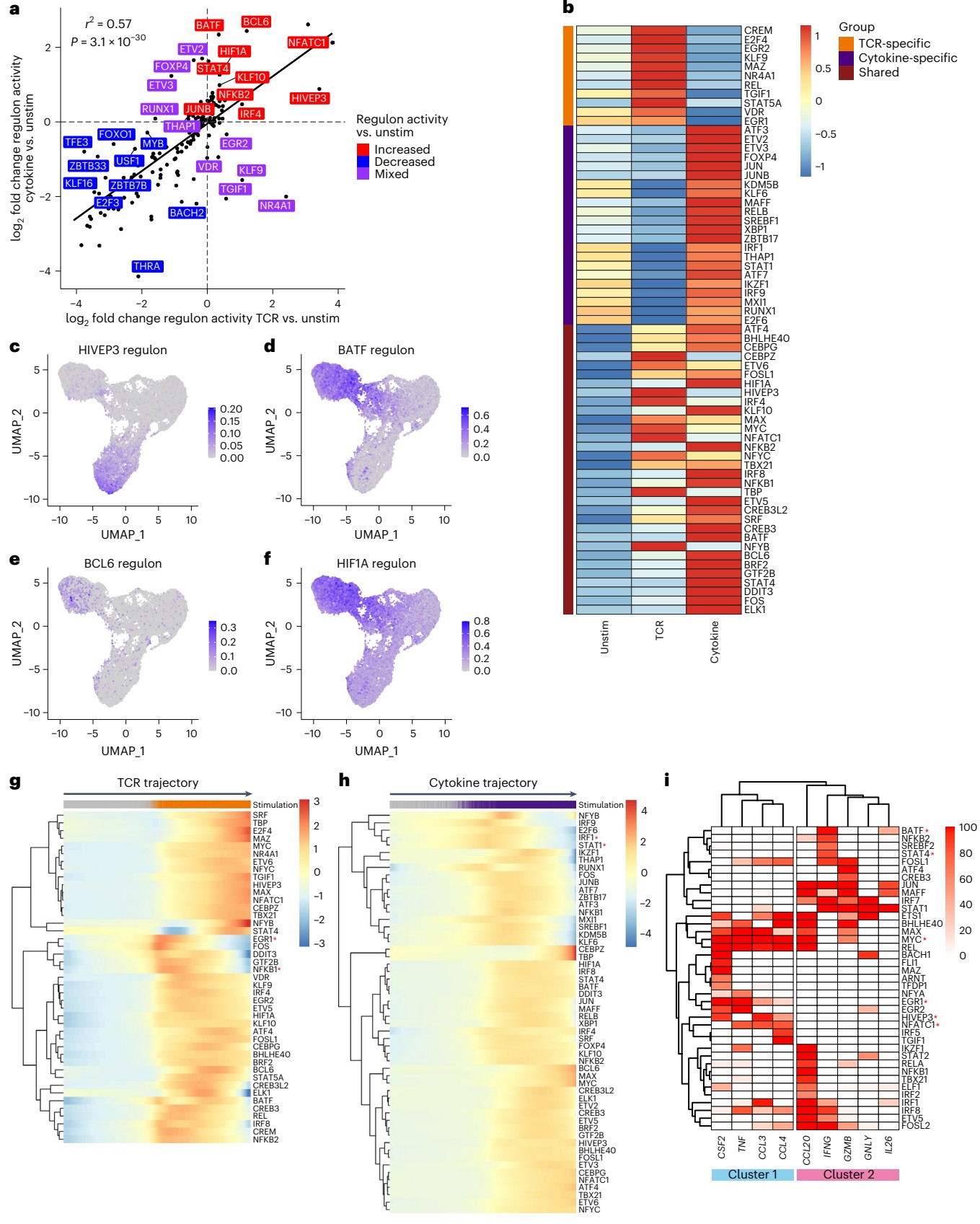

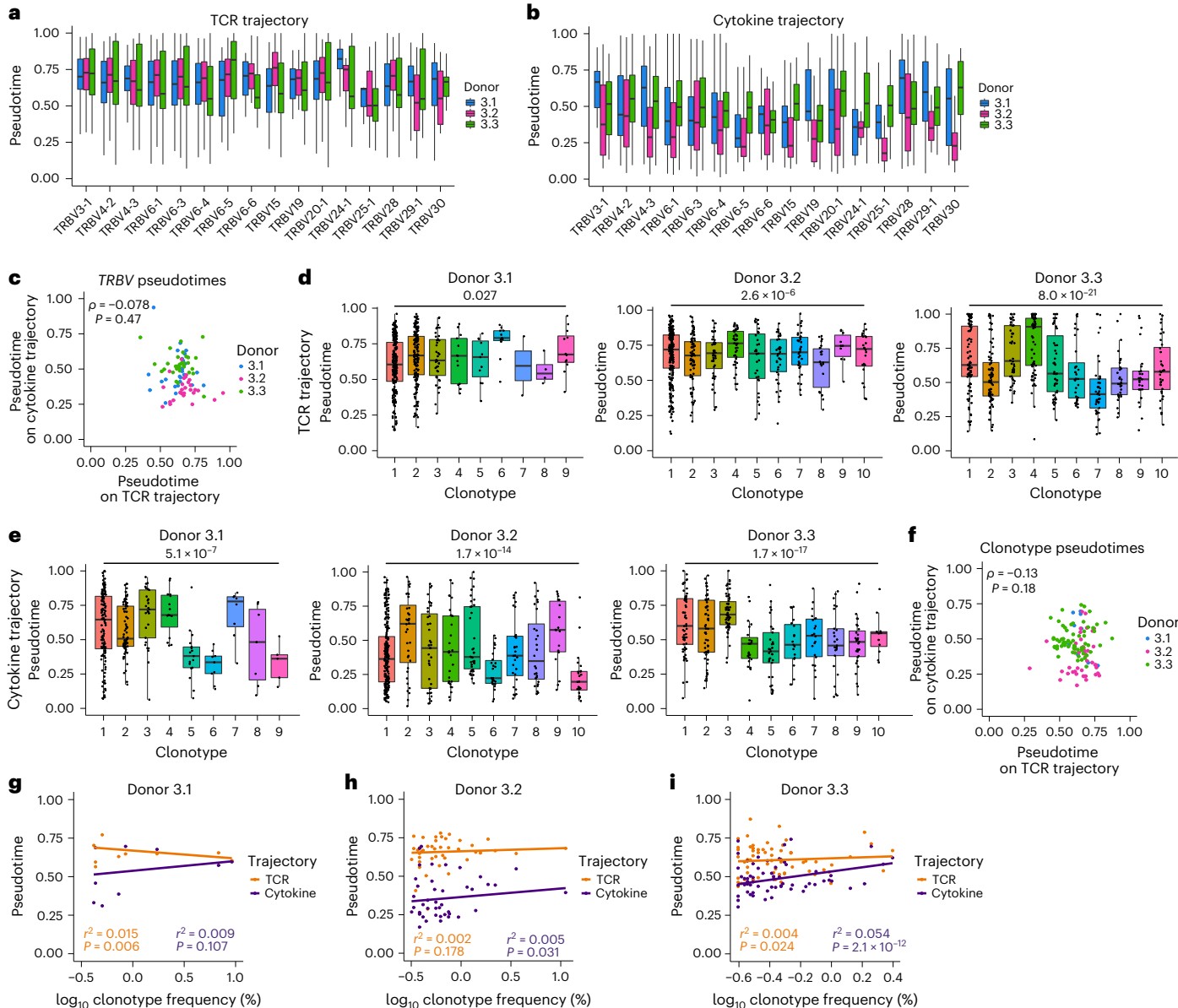

**Fig. 7 | Clonal identity influences MAIT cell activation potential. a,b**, Box plots split by donor ($n = 3$) showing pseudotime values on SCORPIUS TCR (**a**) and cytokine (**b**) trajectories for MAIT cells expressing different *TRBV* gene segments. Kruskal–Wallis test $P$ values for **a** are $8.5 \times 10^{-21}$, $2.2 \times 10^{-9}$ and $1.3 \times 10^{-15}$, and for **b** are 0.045, 0.045 and $2.2 \times 10^{-20}$ for donors 3.1, 3.2 and 3.3, respectively. **c**, Spearman's rank correlation between average *TRBV* pseudotimes on SCORPIUS TCR and cytokine trajectories. **d,e**, Pseudotime values for cells from the largest ten clonotypes in each donor or all clonotypes containing ≥20 cells ($n = 9, 10$ and 10 clonotypes for donors 3.1, 3.2 and 3.3, respectively) on SCORPIUS TCR (**d**) and cytokine (**e**) trajectories. Kruskal–Wallis test was performed for all clonotypes containing ≥20 cells. **f**, Spearman's rank correlation between average clonotype pseudotimes on SCORPIUS TCR and cytokine trajectories. **g–i**, Pearson's correlation between $\log_{10}$ clonotype frequency and pseudotime on SCORPIUS TCR and cytokine trajectories for donors 3.1 (**g**), 3.2 (**h**) and 3.3 (**i**). Plots show stimulated cells only (**a–i**), *TRBV* gene segments with a frequency of >1% in any donor (**a–c**) and clonotypes containing ≥20 cells (**d–i**). In **a**, **b**, **d** and **e**, boxes span the 25th–75th percentiles, the midline denotes the median and whiskers extend to ±1.5 × IQR.

Our data suggest that increased numbers of MAIT cells acquire the capacity to produce IL-17 over time, perhaps due to a requirement for chromatin remodeling, and that aside from IL-17 production, IL-17⁻ and IL-17⁺ MAIT cells show similar transcriptional and functional profiles. This is in stark contrast to resting mouse MAIT cells that comprise distinct MAIT1 and MAIT17 subsets[10–12,33,34].

## Discussion

Our single-cell data from blood and liver, and TCR- and/or cytokine-stimulated MAIT cells, suggest that human MAIT cells comprise a single, highly adaptable, cell population. Transcriptional plasticity is governed by tissue localization, clonal identity and activation state (influenced by type and duration of stimulation). Despite their semi-invariant TCR and shared antigen specificity, diverse TCRβ usage results in private MAIT cell TCR repertoires. This may have important functional consequences, as the clonal identity of an individual MAIT cell influenced its resting and activated transcriptional profile.

Liver MAIT cells were transcriptionally distinct from blood, expressing genes and proteins associated with activation and tissue residency. Basal activation could reflect responses to microbial ligands transported from the gut to the liver via the hepatic portal vein[45]. Liver residency is consistent with bulk RNA-seq

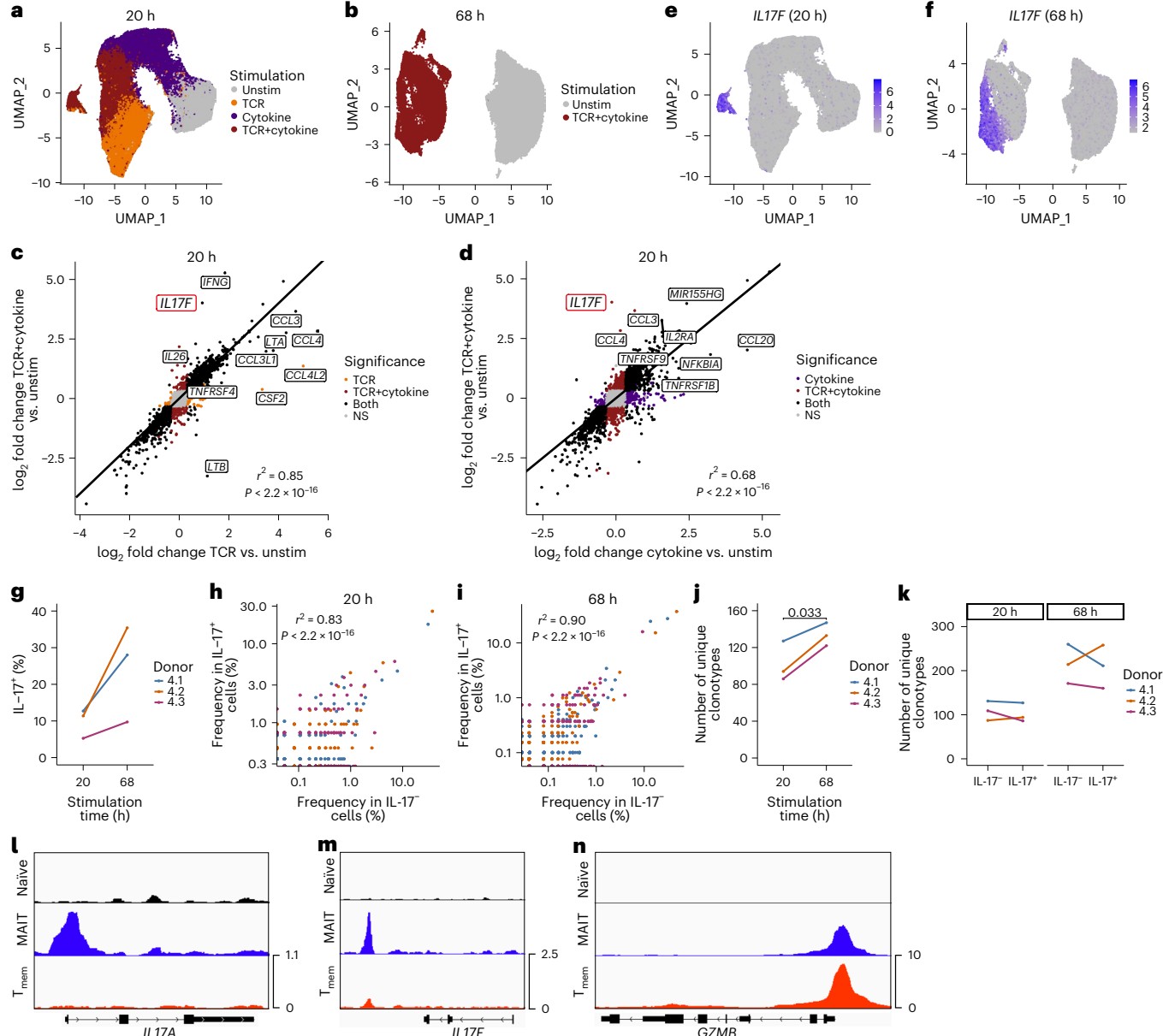

**Fig. 8 | IL-17⁻ and IL-17⁺ MAIT cells are functionally and clonally related.**
**a**, UMAP of 20 h-stimulated MAIT cells colored by stimulation condition.
*n* = 96,867 cells from three donors. **b**, UMAP of 68 h-stimulated MAIT cells
colored by stimulation condition. *n* = 42,765 cells from three donors. **c**, Pearson's
correlation between the log₂ fold change in gene expression between TCR-
stimulated and unstimulated MAIT cells, and TCR+cytokine-stimulated and
unstimulated MAIT cells (20 h stimulation). Labels highlight selected genes
that were differentially regulated by the two stimuli. Point colors indicate
whether the gene was significantly differentially expressed in response to TCR
stimulation only (orange), TCR+cytokine stimulation only (maroon), both (black)
or neither (NS; gray). **d**, Pearson's correlation between the log₂ fold change in
gene expression between cytokine-stimulated and unstimulated MAIT cells, and
TCR+cytokine-stimulated and unstimulated MAIT cells (20 h stimulation). Labels
highlight selected genes that were differentially regulated by the two stimuli.
Point colors indicate whether the gene was significantly differentially expressed
in response to cytokine stimulation only (purple), TCR+cytokine stimulation only

(maroon), both (black) or neither (NS; gray). **e,f**, UMAPs of 20 h-stimulated (**e**)
and 68 h-stimulated (**f**) MAIT cells colored by expression of *IL17F*. **g**, Percentage
of cells within the IL-17-expressing cluster following 20 h or 68 h stimulation.
**h,i**, Pearson's correlation between clonotype frequency in IL-17⁻ (cells within all
other clusters) and IL-17⁺ (cells within the IL-17⁺ cluster) TCR+cytokine-stimulated
MAIT cells at 20 h (**h**) and 68 h (**i**). **j**, Number of unique clonotypes detected
within the IL-17⁺ cluster following 20 h or 68 h stimulation. Cell numbers for each
donor were downsampled to ensure equal numbers of TCR+cytokine-stimulated
cells at the two timepoints. **k**, Number of unique clonotypes within IL-17⁻ (cells
within all other clusters) and IL-17⁺ (cells within the IL-17⁺ cluster) TCR+cytokine-
stimulated MAIT cells following 20 h or 68 h stimulation. Cell numbers for each
donor were downsampled to ensure equal numbers of IL-17⁻ and IL-17⁺ cells at a
given timepoint. **l–n**, Representative ATAC-seq tracks showing *IL17A* (**l**), *IL17F* (**m**)
and *GZMB* (**n**) gene loci in naïve T (black), MAIT (blue) and T_mem (red) cells. *n* = 3
donors in **a–k**. Two-sided paired *t*-test was performed in **g**, **j** and **k** (nonsignificant
results omitted).

analysis of human liver MAIT cells and with mouse parabiosis experi-
ments[12]. However, MAIT cell frequency and TCR usage are similar
in human thoracic duct and matched blood[19], and TCR repertoires
overlapped in blood and liver. In human intestinal and uterine

transplantation, tissue MAIT cells are largely recipient-derived at
>1-year posttransplantation[46,47]. Therefore, the extent of human
MAIT cell tissue residency requires further examination. A small
fraction of liver MAIT cells may be circulating cells in transit through

the liver. However, this is unlikely to have meaningfully impacted our conclusions.

Mouse MAIT cells comprise developmentally, transcriptionally and functionally distinct MAIT1 and MAIT17 subsets[10–12,33,34]. In contrast, human MAIT cells displayed low baseline transcriptional heterogeneity in blood and liver, and clusters were not indicative of MAIT1 and MAIT17 cells or other T cell polarization states. Resting MAIT cell clusters did not clearly associate with the clusters identified following activation. Thus, fRNA-seq adds an important dimension to the analysis of T cell biology. TCR+cytokine stimulation induced IL-17 in a fraction of MAIT cells, but IL-17− and IL-17+ cells similarly expressed other effector molecules and had overlapping TCR repertoires. Therefore, we hypothesize that all human MAIT cells have the capacity to produce IL-17 under appropriate conditions.

TCR and cytokine stimulation induced distinct responses, underpinned by altered regulatory networks. Activated MAIT cells did not comprise discrete functional lineages but were distributed along stimulus-specific activation trajectories. HIVEP3, a TCR-specific transcription factor (regulating *CCL3* and *CCL4*), and BATF, a cytokine-specific transcription factor (regulating *IFNG* and *IL26*), were new predicted regulators of MAIT cell function. A limitation of this analysis is that high-confidence regulons were not identified for RORγt and PLZF, perhaps due to relatively poor gene detection.

Basic TCR repertoire characteristics were consistent with prior studies[21,30]. Surprisingly, the extent of MAIT cell clonality was comparable with $T_{mem}$ cells and individuals displayed largely private TCRαβ repertoires. This challenges the paradigm of MAIT cells as a clonally-restricted population with large numbers of public TCRs and validates previous studies with small cell numbers or bulk TCR repertoire data[20,48]. However, the TCRα chain, key for ligand recognition[31,32], was highly shared between individuals. TCRαβ clonotypes overlapped considerably in blood and liver. While consistent with shared TCRβ usage in matched blood and lymph[19], differential *TRAJ/TRBV* usage was identified in studies without matched blood and tissue (breast[22], kidney and intestine[21]). Identification of private TCR repertoires highlights the importance of matched samples for accurately comparing TCR usage across tissues.

We identified an association between the clonal identity and transcriptome of individual MAIT cells. We hypothesize that the clonotype-cluster association in resting blood and liver reflects differences in the basal activation of clones. Differential activation capacity dependent on TCR clonotype is consistent with altered clonal distribution following *Salmonella* infection[23] and increased clonality with age[20]. However, activation capacity was not correlated with clonotype size. This appears to contrast with the superior proliferation of MAIT cells expressing the most abundant Vβ segments upon in vitro *Escherichia coli* stimulation[7]. Discordant results could reflect differences in experimental approach or the absence of a direct correlation between activation kinetics and proliferative potential. Further study is necessary to understand the driving factors and functional consequences of clonal differences in activation capacity.

A recent paper[49] and a preprint[50] present human MAIT cell scRNA-seq data that are relevant to our findings. However, our study is unique in several key regards, namely the inclusion of TCR data— allowing clonotype to be linked to function—and detailed characterization of responses to multiple stimuli and at multiple timepoints. This is critical for understanding the diversity of MAIT cell functions, including IL-17 production. Consistent with our study, Chandra et al.[50] reported a tissue residency signature in lung MAIT cells and failed to identify a MAIT17 subset. Vorkas et al.[49] analyzed blood MAIT cells following 15 h and 7 d TCR stimulation (direct ex vivo analysis or following cytokine stimulation was not performed). Based on the identification of 12 clusters and gene expression differences between CD4- and CD8-expressing cells (although these did not form separate clusters), the authors concluded that human MAIT cells comprise multiple subsets. However, our trajectory and clonality analysis (based on TCR-confirmed MAIT cells) suggest a continuum of response to stimulation by a single population.

In conclusion, we present a genome-wide single-cell characterization of the transcriptome and TCR repertoire of blood and liver, and resting and activated, human MAIT cells. Our data indicate largely private TCR repertoires, highly shared between matched blood and liver. MAIT cells showed stimulus-specific transcriptional responses, and we identified candidate regulators of the TCR- and cytokine-specific response. While human MAIT cells produce IL-17 following TCR+cytokine stimulation, IL-17+ cells have a similar TCR repertoire and effector profile to IL-17− cells, suggesting they do not comprise a bona fide MAIT17 subset. CD4/CD8 coreceptor expression was not associated with distinct transcriptional states. At rest and following activation, MAIT cell clones show subtle differences in transcriptional profile and functional capacity, which may have important biological consequences, particularly in the context of suboptimal stimulation. Our data provide new insights into human MAIT cell biology, relevant to related innate-like subsets, and a comprehensive resource for further MAIT cell studies in health and disease.

## Online content

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

## Methods

### Data generation

**Liver tissue collection and processing (Exp 1 and 2).** Liver tissue ($n = 7$) and matched blood ($n = 6$) were obtained from patients undergoing liver resection at the Churchill Hospital, Oxford, UK and the University Hospital Basel, Basel, Switzerland (Supplementary Table 1). Patients had no chronic liver disease, active excess alcohol consumption (>14 g per day), infection, immunosuppression or family history of liver disease.

Disease-free liver tissue was collected from the resection margin, cut into small pieces with a scalpel and ground through a 70 μm cell strainer. Cells were washed with R10 (RPMI-1640 (Sigma-Aldrich), 10% FBS (Sigma-Aldrich), 1% penicillin–streptomycin (Thermo Fisher Scientific); $931g$, 10 min, 4 °C) and mononuclear cells isolated by density gradient centrifugation on a discontinuous 35%/70% Percoll (GE Healthcare) gradient ($931g$, 20 min, 21 °C, no brake). Mononuclear cells were collected from the interface and washed with R10 ($596g$, 10 min, 4 °C). Residual red blood cells were lysed with ACK for 3–5 min. Cells were washed twice ($596g$, 10 min, 4 °C) and cryopreserved (90% FBS, 10% DMSO (Sigma-Aldrich)) in liquid nitrogen.

**Ethics statement.** Samples were obtained with written informed consent through the Oxford Gastrointestinal Illnesses Biobank (REC ref. 16/YH/0247) or under Ethikkommission Nordwest- und Zentralschweiz (EKNZ) numbers EKNZ-2014-362, EKNZ-2016-01188 and EKNZ-2019-02118.

**Peripheral blood mononuclear cell (PBMC) isolation.** PBMCs were isolated from fresh whole blood by density gradient centrifugation (Lymphoprep, Axis-Shield) at $931g$ for 30 min with no brake. Cells were cryopreserved in liquid nitrogen and thawed in complete medium (R10, 1X nonessential amino acids (Thermo Fisher Scientific), 1 mM sodium pyruvate (Thermo Fisher Scientific), 10 mM HEPES (pH 7.0–7.5; Thermo Fisher Scientific), 50 μM β-mercaptoethanol (Thermo Fisher Scientific)) on the day of use.

**Stimulation of isolated CD8$^+$/CD3$^+$ T cells for scRNA-seq and scTCR-seq (Exp 3 and 4) or activation marker/cytokine validation.** Pierce streptavidin-coated high-capacity flat-bottom 96-well plates (Thermo Fisher Scientific) were coated with 50 μl biotinylated MR1/5-OP-RU monomer (NIH Tetramer Core Facility) at 10 μg per ml in PBS (Sigma-Aldrich) overnight at 4 °C. Cryopreserved PBMCs were thawed in complete medium. CD8$^+$ T cells were isolated using CD8 MicroBeads (Exp 3; Miltenyi Biotec) and CD3$^+$ T cells using the REAlease CD3 MicroBead Kit (Exp 4 and validation experiments; Miltenyi Biotec) following the manufacturer's instructions. Isolated CD8$^+$/CD3$^+$ T cells were washed in complete medium and resuspended at $1 \times 10^7$ cells per ml. One million (20 h stimulation) or 500,000 (68 h stimulation) cells were added per well to the appropriate 96-well plates (MR1/5-OP-RU-coated plate for TCR and TCR+cytokine stimulation, round-bottom plate for unstimulated and cytokine stimulation). IL-12 (50 ng ml$^{-1}$; R&D Systems) and IL-18 (50 ng ml$^{-1}$; R&D Systems) were added for cytokine stimulation; αCD28 (1 μg ml$^{-1}$; clone: CD28.2; BioLegend) for TCR stimulation; IL-12, IL-18 and αCD28 for TCR+cytokine stimulation; and complete medium for unstimulated cells (final volume 200 μl per well). Cells were incubated for 20 h or 68 h at 37 °C, 5% CO$_2$. For intracellular cytokine staining, brefeldin A (BioLegend) and monensin (BioLegend) were added for the final 4 h.

**Tetramer staining (Exp 1 and 2).** Biotinylated human MR1/5-OP-RU and MR1/6-FP monomers were provided by the NIH Tetramer Core Facility. Tetramers were generated using streptavidin-PE (high concentration) or streptavidin-BV421 (both BioLegend) following the NIH Tetramer Core Facility protocol. Tetramer staining was performed for 40 min at 21 °C in FACS buffer (PBS, 0.5% BSA (Sigma-Aldrich), 1 mM EDTA (Sigma-Aldrich)).

**Surface staining and cell sorting for scRNA-seq and scTCR-seq (Exp 1–4).** TotalSeq-C hashtag antibodies (BioLegend) were used in Exp 2 and 4. Hashtag antibody dilutions were prepared according to the manufacturer's instructions. Namely, antibody vials were centrifuged at 10,000$g$, 30 s, 4 °C, before antibody dilution in FACS buffer. Diluted hashtags were centrifuged at 14,000$g$, 10 min, 4 °C. Cells were incubated in Human TruStain FcX (BioLegend) for 10 min at 4 °C before the addition of diluted hashtag antibodies (0.2 μg per well) for 10 min at 4 °C. Surface fluorochrome-conjugated antibodies were added without washing off the hashtag antibodies. Surface staining was performed in Brilliant Stain Buffer Plus (BD Biosciences) for 30 min at 4 °C. Cells were washed twice in PBS with 0.5% BSA, resuspended in presort buffer (PBS, 1% BSA, 25 mM HEPES) containing 3–5 nM SYTOX Green Nucleic Acid Stain (Thermo Fisher Scientific) and incubated for 20 min at 4 °C. Cells were sorted on a BD FACSAria III with an 85 μm nozzle. Sorted cells were collected in RPMI-1640, 10% FBS, 25 mM HEPES, or HBSS (Thermo Fisher Scientific), 50% FBS, 25 mM HEPES. Sort purity was >99%. For Exp 2 and 4, sorted cells were stained with the TotalSeq-C Human Universal Cocktail V1.0 (BioLegend) according to the manufacturer's instructions. Staining reagents are listed in Supplementary Table 12.

**Stimulation of CD56$^-$ and CD56$^+$ MAIT cells.** CD3$^+$ T cells were isolated using the REAlease CD3 MicroBead Kit following the manufacturer's instructions. Surface antibody and live/dead (SYTOX Green Nucleic Acid Stain) staining was performed as above, then CD56$^-$ and CD56$^+$ MAIT cells (Vα7.2$^+$CD161$^{hi}$) were sorted on a BD FACSAria III with an 85 μm nozzle. Sorted cells were collected in HBSS, 50% FBS, 25 mM HEPES, then centrifuged at 400$g$, 5 min, 21 °C and incubated overnight at 37 °C, 5% CO$_2$. Rested cells were washed in complete medium, plated in a 96-well round-bottom plate and stimulated with IL-12 (50 ng ml$^{-1}$) and IL-18 (50 ng ml$^{-1}$) at 37 °C, 5% CO$_2$ for 20 h, with the addition of brefeldin A and monensin for the final 4 h.

**Surface marker and intracellular cytokine staining for flow cytometry.** Surface staining was performed in Brilliant Stain Buffer Plus for 30 min at 4 °C. Stained cells were washed twice in FACS buffer. For intracellular cytokine staining, cells were fixed in Cytofix/Cytoperm (BD Biosciences) for 20 min at 4 °C, then washed twice in 1X Perm/Wash (BD Biosciences). Intracellular staining was performed in 1X Perm/Wash for 30 min at 4 °C. Cells were acquired on a BD LSR II flow cytometer with BD FACSDiva Software (v8.0.1). Staining reagents are listed in Supplementary Table 12.

**10x Genomics library generation and sequencing.** Sequencing libraries were generated using 10x Genomics Chromium Single Cell V(D)J Reagent Kits (v1.0 Chemistry; Exp 1 and 3) or 10x Genomics Chromium Next GEM Single Cell 5′ Reagent Kits v2 (Dual Index; Exp 2 and 4) following the manufacturer's instructions. For Exp 1 and 3, cells were loaded onto the Chromium Controller (10x Genomics) at a concentration of ~1 × 10$^6$ cells per ml, with 6,000–8,000 cells loaded per channel. For Exp 2 and 4, 17,750–30,000 cells were loaded per channel. Library quality and concentration were assessed using a TapeStation (Agilent) and Qubit 2.0–4 Fluorometer (Thermo Fisher Scientific), respectively. Library generation for Exp 1 and 3 was performed at the Oxford Genomics Centre (Wellcome Centre for Human Genetics, University of Oxford), and for Exp 2 and 4 was performed in-house. Libraries were sequenced on an Illumina HiSeq 4000 (Exp 1) or Illumina NovaSeq 6000 (Exp 2–4) at the Oxford Genomics Centre. Sequencing depths were as follows: Exp 1—39,013–46,998 reads per cell for scRNA-seq, 10,323–30,883 reads per cell for scTCR-seq; Exp 2—76,710–89,378 reads per cell for scRNA-seq, 19,673–41,612 reads per cell for TotalSeq-C feature barcoding antibodies, 5,192–7,065 reads per cell for scTCR-seq; Exp

3−75,638–88,871 reads per cell for scRNA-seq, 12,517–27,197 reads per cell for scTCR-seq; Exp 4–53,120–102,777 reads per cell for scRNA-seq, 9,333–17,717 reads per cell for TotalSeq-C feature barcoding antibodies, 2,904–14,636 reads per cell for scTCR-seq.

**ATAC-seq library generation and sequencing.** Naïve T cells (CD8$^+$CD45RO$^-$CCR7$^+$), MAIT cells (CD8$^+$CCR7$^-$MR1/5-OP-RU$^+$) and T$_{mem}$ cells (CD8$^+$CCR7$^-$MR1/5-OP-RU$^-$) were sorted from CD8-enriched (CD8 MicroBeads) PBMCs ($n = 3$ donors, 50,000 cells per population). ATAC-seq was performed as previously described[51]. Briefly, cells were pelleted at 500$g$ for 10 min at 4 °C and the supernatant was removed. Cells were resuspended in 50 μl cold lysis and transposition mix (25 μl TD buffer, 2.5 μl TDE1 (Illumina, FC-121-1030; product discontinued), 22 μl nuclease-free H$_2$O (Thermo Fisher Scientific), 0.5 μl 1% digitonin (Promega)) and incubated for 30 min at 37 °C with agitation at 300 rpm (Thermo-Shaker TS-100, Biosan). Transposed DNA was purified using the Qiagen MinElute Reaction Cleanup Kit and eluted in 13 μl elution buffer (10 mM Tris−HCl, pH 8). Purified library fragments were PCR amplified for 11 cycles with barcoded primers using NEBNext High-Fidelity 2X PCR Master Mix (New England Biolabs). Amplified DNA was purified using the Qiagen MinElute PCR Purification Kit (23 μl elution volume) and PCR primer contamination was removed using SPRI beads (5 min dry time, 15 μl elution volume; Agencourt AMPure XP PCR Purification, Beckman Coulter). Fragment size distribution was analyzed using a 2100 Bioanalyzer (Agilent) with the High Sensitivity DNA Kit. Libraries were quantified using the KAPA Library Quantification Kit (Roche). Paired-end sequencing (40 bp) was performed on an Illumina NextSeq 500 using the High Output v2 Kit (75 cycles). Libraries were sequenced to a depth of 215–271 million paired-end reads per sample.

## Data analysis

**10x Genomics raw data processing.** FASTQ files were generated from BCL files using Illumina bcl2fastq. For Exp 1 and 3, FASTQ files for gene expression and TCR data were processed using Cell Ranger (v3.0.1–3.0.2; https://support.10xgenomics.com/single-cell-gene-expression/software/pipelines/latest/what-is-cell-ranger) count and vdj pipelines, respectively. For Exp 2 and 4, FASTQ files for all modalities were processed using the Cell Ranger (v7.0.1) multi pipeline. For TCR analysis, the filtered_contig_annotations.csv file was filtered to retain only high-confidence, full-length, productive contigs corresponding to TCRα or TCRβ chains.

**Hashtag demultiplexing.** Hashtag demultiplexing (Exp 2 and 4) was performed using the consensus calling approach from cellhashR[52] (v1.0.3) with the following methods: BFF$_{cluster}$[52], BFF$_{raw}$[52], GMM-Demux[53], MULTI-seq[54], Seurat HTODemux[55] and DropletUtils hashedDrops[55].

**Quality control.** Quality control was performed separately for cells from each channel of the Chromium Controller. Filtered feature-barcode matrices from Cell Ranger count/multi were imported into R using Seurat (v4.0.3–4.3.0)[56]. Cells with low unique molecular identifier counts, low gene counts and/or a high percentage of mitochondrial reads, were removed. For Exp 1 and 2, cells labeled as empty droplets or damaged cells by DropletQC[57] (v0.0.0.9000) were removed (damaged cells in Exp 4 were also removed). For Exp 2 and 4, only cells called as consensus singlets by hashtag demultiplexing were retained. Cells with two TCRα and two TCRβ chains, or more than two TCRα and/or TCRβ chains, were assumed to be doublets and discarded. TCR and BCR genes were removed to ensure downstream clustering analysis was not influenced by TCR or BCR chain usage.

**Normalization, integration, dimensionality reduction and clustering (Exp 1 and 2).** For combined analysis of Exp 1 and 2, data from each donor were normalized separately using sctransform[58] (v0.3.5). Highly variable genes (HVGs) were defined as the 3,000 genes with the largest

residual variance following variance stabilizing transformation. Cells from different donors were integrated using Seurat[56]. Integration features ($n = 3,000$) were selected using matched blood and liver samples, with STACAS[59] (v2.0.1) blacklisted genes subsequently removed. For anchor finding, dimensionality reduction was performed using canonical correlation analysis (MAIT cells only) or reciprocal principal component analysis (PCA; MAIT and T$_{mem}$ cells combined). The number of dimensions used for identifying and weighting anchors was selected empirically by performing integration with multiple input dimensions and evaluating downstream clustering results. Following integration, dimensionality reduction was performed using PCA. Scree plots were used to determine how many PCs to use for UMAP generation and clustering. Cell clusters were identified using Seurat's graph-based clustering approach. Briefly, a shared nearest neighbor graph was constructed using dimensionally-reduced data, and then clusters were determined by optimizing the standard modularity function (Louvain algorithm).

**Normalization, dimensionality reduction, batch correction and clustering (Exp 3 and 4).** Per experiment and timepoint, data from all donors combined were normalized using sctransform[58] (v0.3.2–0.3.5). HVGs were defined as the 3,000 genes with the largest residual variance following variance stabilizing transformation (in Exp 4, STACAS[59] blacklisted genes were removed from HVGs). Dimensionality reduction was performed using PCA. Batch correction for donor was performed using Harmony[60] (v0.1.1) with 50 input PCs ($\theta = 2$, $\lambda = 1$). Scree plots were used to determine how many PCs to use for UMAP generation and clustering. Cell clusters were identified using Seurat's graph-based clustering approach.

**Differential gene expression analysis.** Differential gene expression analysis between clusters was performed using MAST[61] (v1.18.0–1.24.1; FindMarkers function from Seurat) with cellular detection rate as a covariate. Cluster markers were defined as genes with significantly increased expression in one cluster relative to the average of all other clusters (fold change > 1.25 and adjusted $P < 0.05$ based on Bonferroni correction using all genes in the dataset). Differential gene expression analysis between conditions (for example, tissues, coreceptors and stimuli) was performed using MAST[61] with cellular detection rate and donor as covariates. Genes with a fold change > 1.25 and a Bonferroni adjusted $P < 0.05$ were defined as significantly differentially expressed. Input data were log-transformed normalized counts generated by global-scaling normalization (NormalizeData function from Seurat).

**Coreceptor assignment (Exp 1 and 2).** Cells were defined as CD8$^+$, DN or CD4$^+$ based on normalized coreceptor gene or protein (measured using TotalSeq-C antibodies) expression. For assignment based on gene expression, cells were defined as CD8$^+$ if *CD8A* > 0 and/or *CD8B* > 0, CD4$^+$ if *CD4* > 0, and DN if *CD8A*, *CD8B* and *CD4* were undetected. For assignment based on protein expression (Exp 2 only), cells were defined as CD8$^+$ if CD8 > 0.3, CD4$^+$ if CD4 > 2.5 and CD8 < 0.2, and DN if neither CD8$^+$ nor CD4$^+$. Thresholds were selected empirically by examining histograms of normalized count data.

**Pseudotime analysis.** Pseudotime analysis was performed using Slingshot[37] (v2.0.0) and SCORPIUS[38] (v1.0.8). UMAP coordinates and Seurat cluster labels (0.1 resolution) were provided as input to Slingshot, with the main unstimulated cluster specified as the start of the trajectory. Normalized expression values (sctransform) were provided as input to SCORPIUS. Two separate SCORPIUS trajectories were generated from unstimulated and TCR-stimulated cells, and unstimulated and cytokine-stimulated cells. Gene importance along SCORPIUS trajectories was determined using random forest regression (gene_importances function, num_permutations = 10). Differential gene importance was calculated by taking the ratio of gene importance ranks on the TCR and cytokine trajectories (higher rank as numerator). Genes defined

as differentially important had an importance false discovery rate (FDR) < 0.05 and were within the top 150 most highly ranked genes for either the TCR or cytokine trajectory.

**Transcription factor regulon analysis.** Transcription factor regulons were identified using SCENIC[29,62] (pySCENIC v0.11.2–0.12.1). Briefly, the raw expression matrix was filtered to retain genes expressed in >1% of cells and with a count >3 × 0.01 × number of cells. Modules comprising transcription factors and coexpressed genes were generated using GRNBoost2, then pruned to remove indirect targets lacking enrichment for the corresponding transcription factor motif (cisTarget). This resulted in a set of transcription factor regulons. Due to stochasticity in gene regulatory network inference using GRNBoost2, each pySCENIC run can identify a different number of regulons, as well as different target genes for each transcription factor. Thus, pySCENIC was run 100 times. High-confidence regulons were defined as regulons that occurred in >80% of runs and that contained at least five high-confidence target genes. High-confidence target genes were those found within a regulon in >80% of runs. Cells were scored for the activity of each high-confidence regulon (including only high-confidence target genes) using AUCell (v1.16.0–1.20.1). Regulons differentially active between tissues or stimulation conditions were determined using MAST[61] with donor as a covariate. Regulons with a Bonferroni adjusted $P < 0.01$ were defined as differentially active. Smoothed regulon activity scores (AUCell scores) over SCORPIUS trajectories were generated by loess regression (loess function from the stats R package).

**Gene set enrichment analysis (GSEA) and over-representation analysis.** GSEA[63] (v4.3.2) for published mouse and human $T_{RM}$ cell gene signatures[27,28] was performed using pseudobulk gene counts (normalized gene counts summed for all cells within a sample) with 1,000 gene permutations. Over-representation analysis for gene ontology (GO) terms and Reactome pathways was performed using clusterProfiler[64] (v4.7.1) and ReactomePA[65] (v1.36.0), respectively. Gene symbols were converted to Entrez IDs using the Bioconductor org.Hs.eg.db annotation package (v3.13.0–3.16.0). Background genes were defined as genes expressed (count >0) in ≥1% of cells (or for Exp 3, genes expressed in at least five cells). Redundant enriched GO terms were removed using the simplify function from clusterProfiler.

**Gene lists.** MAIT1 and MAIT17 gene signatures were generated by overlapping MAIT1 and MAIT17 genes from two published scRNA-seq datasets[33,34]. Mouse gene symbols were converted to human gene symbols using the biomaRt R package (getLDS function; v2.54.1) with the ENSEMBL_MART_ENSEMBL BioMart database and the hsapiens_gene_ensembl and mmusculus_gene_ensembl datasets. Human Th1 and Th17 gene signatures were generated by combining genes from the NanoString nCounter Human Immunology V2 Panel Gene List (https://nanostring.com/support-documents/ncounter-human-immunology-v2-panel-gene-list) and a meta-analysis published by Radens et al.[35]. Interferon-stimulated genes were obtained from Schoggins and Rice[66].

**Nucleotide and functional TCR clonotypes.** Tables of TCRα and TCRβ usage for each cell in a sample were combined to generate one table per donor for nucleotide clonotype calling, and one table for all donors for functional clonotype calling. MAIT cells were required to have a *TRAV1-2* TCRα chain and at least one TCRβ chain. $T_{mem}$ cells were required to have at least one TCRα chain and at least one TCRβ chain. Cells expressing *TRAV1-2* paired with *TRAJ33*, *TRAJ12* or *TRAJ20* and with a 12 amino acid CDR3α region were assumed to be contaminating MAIT cells and were removed before $T_{mem}$ cell clonotype calling.

Nucleotide clonotypes (TCRαβ) were defined as cells with identical TCR gene segment usage, and CDR3α and CDR3β nucleotide sequences. TCRα clonotypes and TCRβ clonotypes were defined as cells with identical TCRα segment usage and CDR3α sequences, or

identical TCRβ segment usage and CDR3β sequences, respectively. TCRαβ clonotypes were numbered according to size, with clonotype 1 being the largest, clonotype 2 being the second largest and so on. Clonotypes of identical size were randomly ordered for numbering. TCRαβ clonotypes were assigned ranks in a similar manner, but clonotypes of identical size were given the same rank.

Functional clonotypes (TCRαβ) were defined as cells with at least one identical TCRα and TCRβ chain amino acid sequence (gene segment usage and CDR3 sequences). Functional TCRα and functional TCRβ clonotypes were defined as cells with at least one matching TCRα or TCRβ chain amino acid sequence, respectively. Given the presence of TCR dropout, functional clonotypes were permitted to contain a mixture of cells with one or two TCRα or TCRβ chains, providing all detected chains matched those within the clonotype.

**TCR analyses.** TCR analyses were performed only for cells with a defined TCR clonotype. The Shannon diversity index was calculated using the diversity function from the vegan R package (v2.6.4). To test for an association between clonotype (clonotypes from $n = 12$ donors with ≥20 cells) and cluster, a multinomial test was performed using the EMT R package (v1.3)—MonteCarlo = FALSE when the number of distinct possible outcomes (events) $< 1 × 10^6$, else MonteCarlo = TRUE with ntrial = $10 ×$ events or $1 × 10^8$ (whichever smaller). $P$ values were Bonferroni adjusted for the number of clonotypes tested per donor. Sequence logos were generated using ggseqlogo (v0.1). The overall height of the stacked letters at each position indicates the sequence conservation, while the relative abundance of each amino acid is indicated by the height of individual letters within the stack. Acidic bases are shown in red, basic residues in blue, hydrophobic in black and polar in green. The number of N-nucleotides and P-nucleotides in 36-nucleotide CDR3α sequences was determined using IMGT/JunctionAnalysis[67] (v2.3.0).

**ATAC-seq.** Read quality was checked using FastQC (v0.11.5; Babraham Bioinformatics) and adapter sequences were removed using Trimmomatic[68] (v0.36). Reads were mapped to hg38 using Bowtie 2 (ref. 69; --very-sensitive -X 2000 --no-mixed --no-discordant --no-unal; v2.3.4.1). BAM files from multiple sequencing lanes were merged using samtools[70] (v1.6) and duplicate reads were removed using Picard Mark-Duplicates (v2.15.0). BAM files were filtered to remove reads mapping to the mitochondrial genome (samtools) and blacklisted regions (bedtools[71] v2.26.0). Blacklisted regions comprised the ENCODE blacklist[72] and a custom ATAC-seq blacklist generated by J. Buenrostro[73]. Read start sites were adjusted to correspond to the center of the transposase binding site—reads on the forward strand were offset by +5 bp and reads on the reverse strand were offset by −4 bp using deepTools[74] alignmentSieve (v3.1.0). Cut sites were identified as the 5′ ends of forward and reverse reads. Peak calling was performed with MACS2 (ref. 75; --nomodel -p 0.1 -f BAMPE --call-summits; v2.1.1) and an optimal peak list defined using the irreproducibility discovery rate framework[76]. Cut sites within peaks were quantified using featureCounts[77] (v1.6.0). Peaks were annotated to genes using HOMER[78] annotatePeaks.pl (v4.8). Differentially accessible peaks (fold change > 2, FDR < 0.05) were identified using edgeR[79] (v3.24.3). BAM files were converted into BigWig files using deepTools[74] bamCoverage (--normalizeUsing CPM --ignoreForNormalization chrM chrX chrY --binSize 1; v3.5.1).

**Statistics and reproducibility.** Statistical tests are listed in the relevant sections of the Methods and/or in the figure legends. Standard statistical tests were performed in R (v4.1.1–4.2.0) using the stats or rstatix (v0.7.2) packages. All tests were two-sided. Initial blood-liver (Exp 1) and stimulation (Exp 3) scRNA-seq experiments were followed by validation experiments (Exp 2 and 4)—the findings of these were highly concordant. Genes of interest from Exp 1 and 3 were validated at the protein level by CITE-seq (Exp 2 and 4) and flow cytometry.

No statistical method was used to predetermine sample size. No samples were excluded from the analyses. The experiments were not randomized. The investigators were not blinded to allocation during experiments and outcome assessment.

**Plots.** Most plots were generated using the ggplot2 R package (v3.3.4–3.4.2). FACS plots were generated in FlowJo (v10.8.1; BD Biosciences). Heatmaps were generated using pheatmap (v1.0.12) or ComplexHeatmap (v2.14.0). GSEA plots were generated in GraphPad Prism (v9.5.1; GraphPad Software, LLC). ATAC-seq traces were generated using Integrative Genomics Viewer (v2.16.0).

### Reporting summary
Further information on research design is available in the Nature Portfolio Reporting Summary linked to this article.

### Data availability
Sequencing data generated in this study have been deposited in NCBI's Gene Expression Omnibus (GEO) and are accessible through GEO SuperSeries accession number GSE194189. The Bioconductor org.Hs.eg.db annotation package and ENSEMBL_MART_ENSEMBL BioMart database are publicly available. Source data are provided with this paper.

### Code availability
No custom software was generated for this manuscript. The software used for all analyses is listed in the relevant sections of the Methods.

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

### Acknowledgements
We thank biobankers of the Translational Gastroenterology Unit and surgeons of Oxford University Hospitals NHS Foundation Trust for collecting patient samples, especially M. Silva for facilitating this; S. Slevin for liver processing; K. Lynch for liver patient identification and sample processing; M. Salio for the plate-bound MR1 stimulation protocol; H. Ferry for sorting; D. Sims and D. Agarwal for discussions regarding computational analysis; and the NIH Tetramer Core Facility for providing MR1 monomers. We thank the Oxford Genomics Centre at the Wellcome Centre for Human Genetics (funded by Wellcome grant ref. 203141/A/16/Z) for the generation and initial processing of the sequencing data. Analysis was performed using computer systems at the MRC WIMM Centre for Computational Biology. L.C.G. and A.A. are supported by Wellcome (109028/Z/15/Z and 216417/Z/19/Z, respectively). M.E.B.F. is supported by Beyond

Celiac and the Academy of Medical Sciences. M.F.S. is supported by the Swiss National Science Foundation (PZOOP3_167828 and PZOOP3_189490), Goldschmidt-Jacobson Stiftung and Uniscientia Stiftung. N.M.P. is supported by an Oxford-UCB Postdoctoral Fellowship. P.K. is supported by Wellcome (222426/Z/21/Z), the NIH (U19 I082360), the NIHR Oxford Biomedical Research Centre and an NIHR Senior Fellowship.

## Author contributions

L.C.G., N.M.P. and P.K. designed the project and the experiments. L.C.G., A.A., M.E.B.F., M.J.L., G.F.H., M.F.S., N.M.P. and P.K. recruited patients and/or performed liver tissue processing. L.C.G. performed all other experiments and all data analysis. L.C.G. wrote the manuscript. All authors critically reviewed and approved the manuscript.

## Competing interests

P.K. has acted as a consultant to UCB, Biomunex, AstraZeneca and Infinitopes. N.M.P. has acted as a consultant to Infinitopes. The other authors declare no competing interests.

## Additional information

**Extended data** is available for this paper at https://doi.org/10.1038/s41590-023-01575-1.

**Correspondence and requests for materials** should be addressed to Lucy C. Garner or Paul Klenerman.

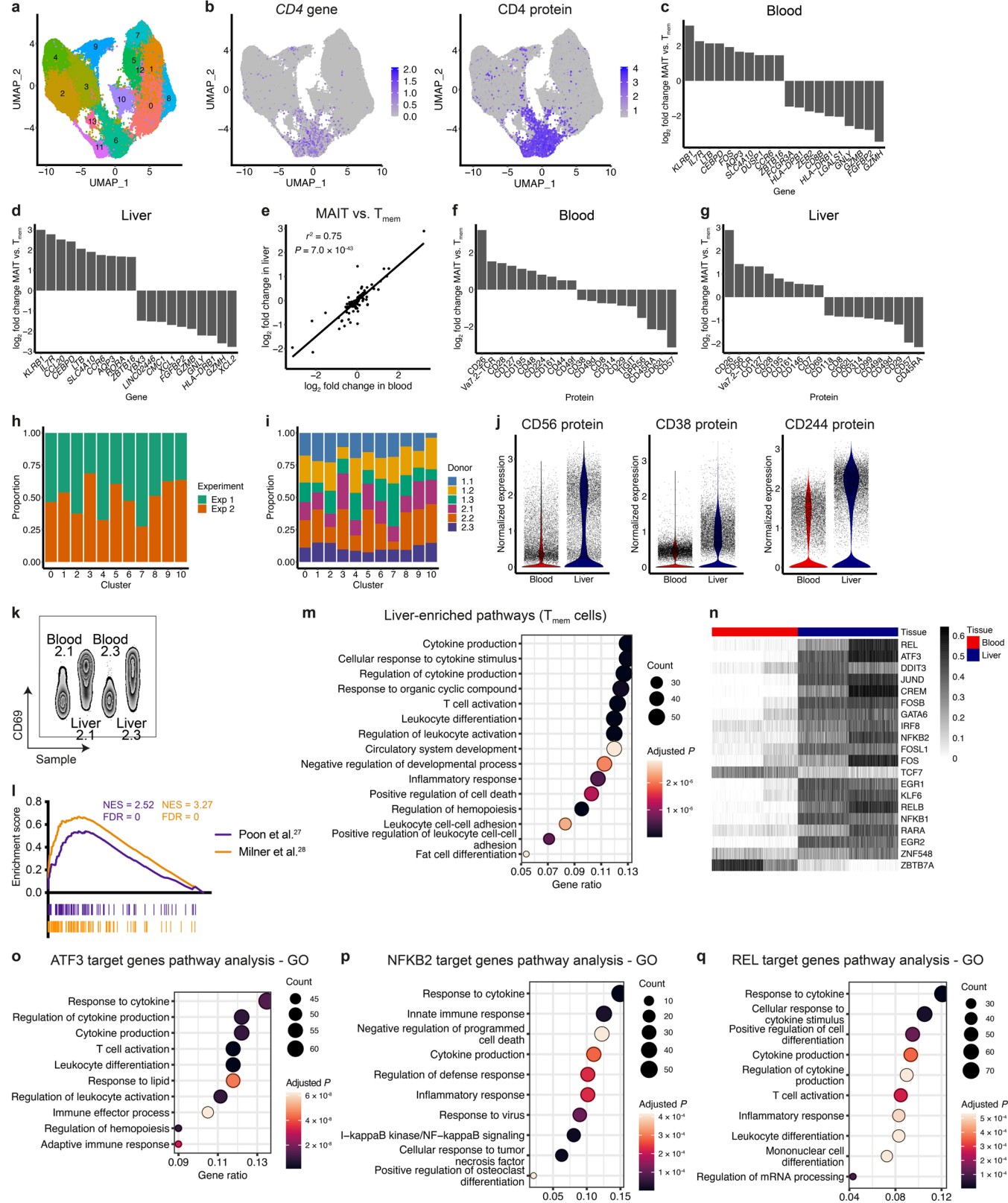

**Extended Data Fig. 1 | See next page for caption.**

**Extended Data Fig. 1 | MAIT and T$_{mem}$ cells, and blood and liver MAIT cells, exhibit distinct phenotypes and transcriptomes. a**, UMAP of blood and liver MAIT cells and conventional memory T (T$_{mem}$) cells colored by the 14 identified clusters. $n$ = 89,456 cells from 12 donors (11 blood, 7 liver). **b**, UMAPs of blood and liver MAIT and T$_{mem}$ cells colored by CD4 gene (left) and protein (right) expression. **c**,**d**, Bar plots showing the log$_2$ fold change in expression of the top ten genes upregulated in MAIT cells compared with T$_{mem}$ cells, and vice versa, in the blood (**c**) and liver (**d**). **e**, Pearson's correlation between the log$_2$ fold change in protein expression between MAIT and T$_{mem}$ cells in the blood, and MAIT and T$_{mem}$ cells in the liver. **f**,**g**, Bar plots showing the log$_2$ fold change in expression of the top ten proteins upregulated in MAIT cells compared with T$_{mem}$ cells, and vice versa, in the blood (**f**) and liver (**g**). **h**,**i**, Proportion of cells in each cluster (analysis of MAIT cells from six matched blood-liver donors) from Exp 1 and Exp 2 (**h**) and from each donor (**i**). **j**, Expression of CD56, CD38 and CD244 (2B4) proteins in blood and liver MAIT cells. **k**, Flow cytometry plot showing CD69 expression on blood and liver MAIT cells from two representative donors. **l**, Gene set enrichment analysis of liver compared with blood T$_{mem}$ cells using published human and mouse tissue-resident memory T cell gene signatures. NES, normalized enrichment score. **m**, Over-representation analysis on the genes significantly upregulated in liver T$_{mem}$ cells compared with blood T$_{mem}$ cells. Top 15 gene ontology (GO) terms and associated Benjamini–Hochberg adjusted $P$ values are shown. **n**, Heatmap showing activity (row-scaled AUCell scores) of the 20 most differentially active regulons (largest difference in average AUCell score) between matched blood and liver MAIT cells in Exp 2. $n$ = 3 donors. **o-q**, Over-representation analysis on predicted ATF3 (**o**), NFKB2 (**p**) and REL (**q**) target genes. Top ten GO terms and associated Benjamini–Hochberg adjusted $P$ values are shown.

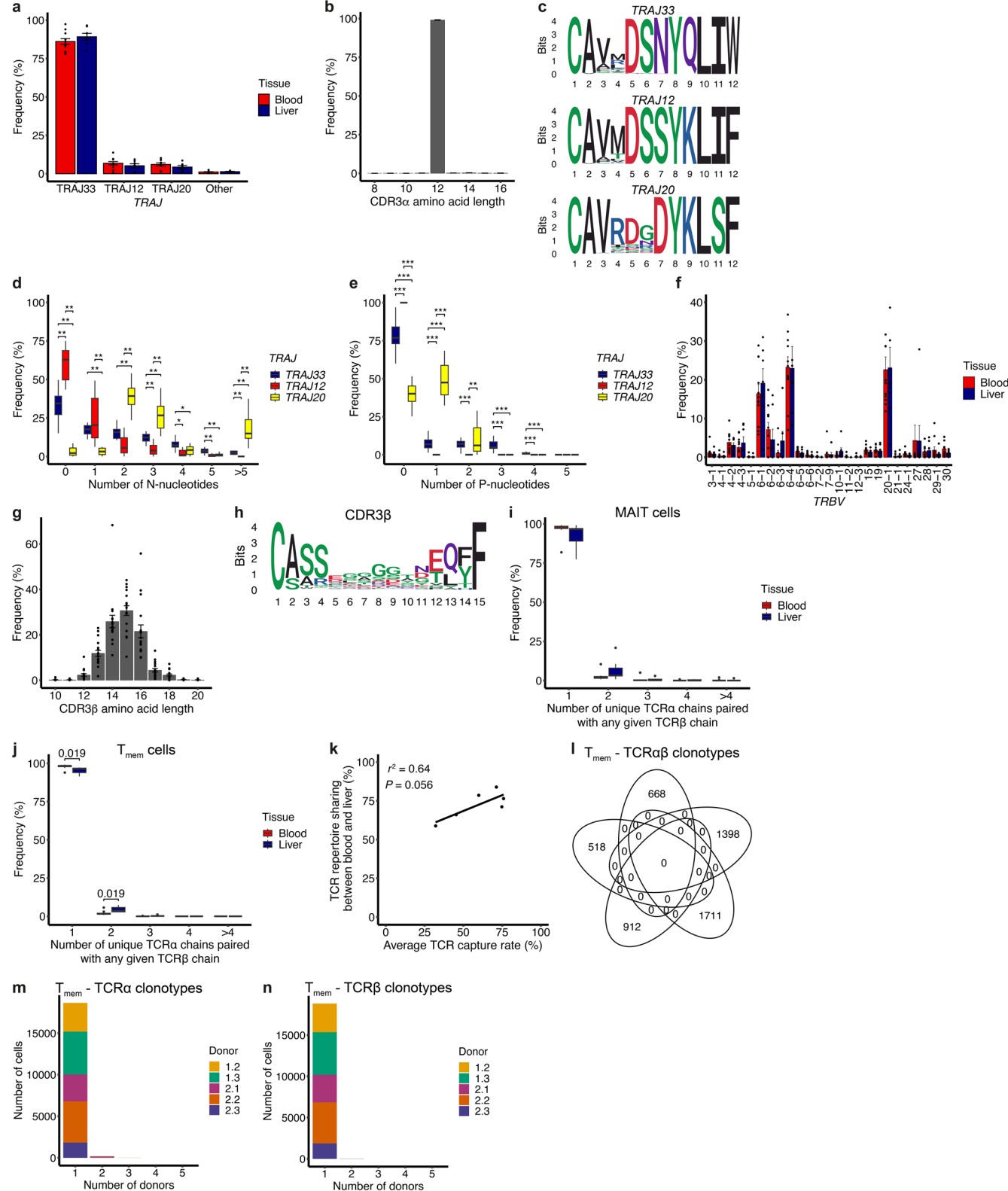

**Extended Data Fig. 2 | See next page for caption.**

**Extended Data Fig. 2 | MAIT cells have a restricted TCRα but diverse TCRβ chain, resulting in private TCRαβ repertoires. a**, Proportion of blood and liver cells expressing *TRAJ33*, *TRAJ12*, *TRAJ20* and other *TRAJ* gene segments. **b**, Distribution of CDR3α amino acid lengths. **c**, Sequence logos generated from all *TRAJ33*, *TRAJ12* or *TRAJ20* CDR3α amino acid sequences of length 12 ($n$ = 26,529, 1,852 and 1,451 sequences, respectively). **d**,**e**, Frequency of N-nucleotides (**d**) and P-nucleotides (**e**) in *TRAJ33*, *TRAJ12* and *TRAJ20* TCRs from 12 donors. **f**, Proportion of blood and liver MAIT cells expressing different *TRBV* gene segments. Plot includes *TRBV* gene segments with a frequency >1% in at least one sample. **g**, Distribution of CDR3β amino acid lengths. **h**, Sequence logo generated from all MAIT cell CDR3β amino acid sequences of length 15 ($n$ = 9,300 sequences). **i**,**j**, TCR chain pairing at the population level. Number of unique TCRα chains paired with any given TCRβ chain in blood and liver MAIT

(**i**; $n$ = 11 blood, 7 liver samples) or T$_{mem}$ (**j**; $n$ = 10 blood, 5 liver samples) cells. **k**, Pearson's correlation between the average TCR capture rate (percentage of cells with a paired *TRAV1-2* TCR) for a donor ($n$ = 6) and percentage MAIT cell TCR repertoire sharing between matched blood and liver. **l**, Venn diagram showing the number of TCRαβ clonotypes shared between the five T$_{mem}$ cell donors. **m**,**n**, Number of T$_{mem}$ cells from each donor belonging to TCRα clonotypes (**m**) or TCRβ clonotypes (**n**) found in 1, 2, 3, 4, or 5 (all) donors. **a-c** and **f-i** show data from $n$ = 18 samples (11 blood, 7 liver), 12 donors. Data in **a**, **b**, **f**, **g** are presented as mean ± s.e.m. In **d**, **e**, **i**, **j**, boxes span the 25th–75th percentiles, the midline denotes the median and whiskers extend to ± 1.5 × IQR. Points in **i** and **j** indicate outliers. Two-sided Wilcoxon rank-sum test (**a**, **f**, **i**, **j**) and two-sided Wilcoxon signed-rank test (**d**, **e**) for all pairwise comparisons. Benjamini-Hochberg adjusted $P$ values are shown (nonsignificant results omitted). *$P$ < 0.05, **$P$ < 0.01, ***$P$ < 0.001.

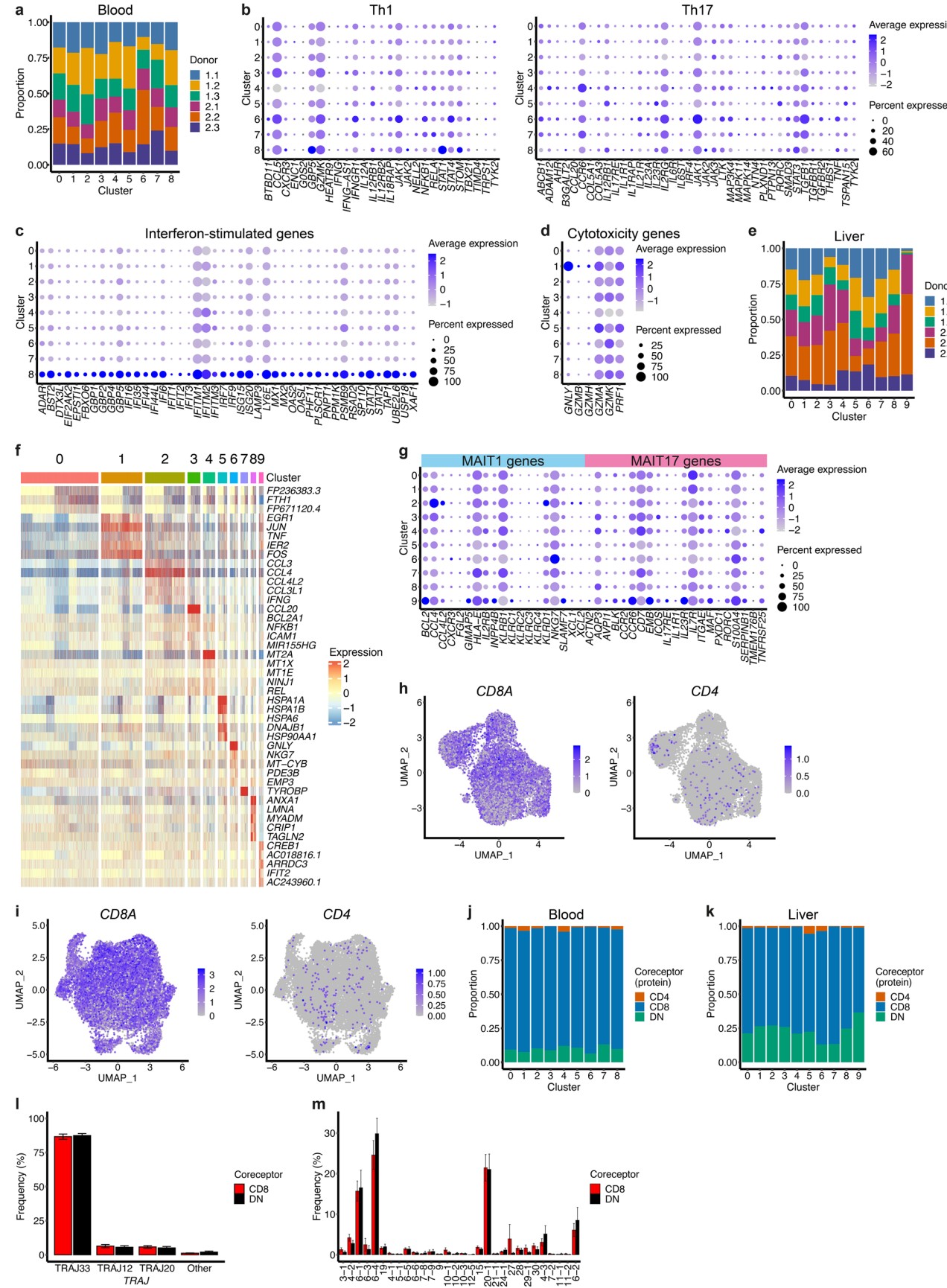

**Extended Data Fig. 3 | See next page for caption.**

**Extended Data Fig. 3 | MAIT cells within the blood and liver show minimal transcriptional heterogeneity. a**, Proportion of cells in each blood MAIT cell cluster from each donor (*n* = 6). **b-d**, Expression of Th1 and Th17 (**b**), interferon-stimulated (**c**) and cytotoxicity (**d**) genes in blood MAIT cell clusters. Dot color indicates the level of gene expression and dot size indicates the percentage of cells expressing the gene. **e**, Proportion of cells in each liver MAIT cell cluster from each donor (*n* = 6). **f**, Heatmap showing row-scaled log-transformed normalized expression of the top five or all (if <5) marker genes for each liver MAIT cell cluster. **g**, Expression of MAIT1 and MAIT17 genes in liver MAIT cell clusters. Dot color indicates the level of gene expression and dot size indicates the percentage of cells expressing the gene. **h,i**, UMAPs of blood (**h**) and liver (**i**) MAIT cells colored by expression of *CD8A* (left) and *CD4* (right). CD4+ cells positioned in front of CD4− cells to allow better visibility. **j,k**, Proportion of CD4+, CD8+ and DN cells in each blood (**j**) and liver (**k**) cluster. Coreceptor identity defined based on the expression of CD4 and CD8 proteins (Methods). **l,m**, Frequency of CD8+ and DN MAIT cells expressing the indicated *TRAJ* (**l**) and *TRBV* (**m**) genes. *n* = 12 donors. *TRBV* genes expressed in >1% of CD8+ or DN cells from any donor are included. Mean ± s.e.m. is shown. Two-sample Wilcoxon signed-rank test for all pairwise comparisons (nonsignificant results omitted).

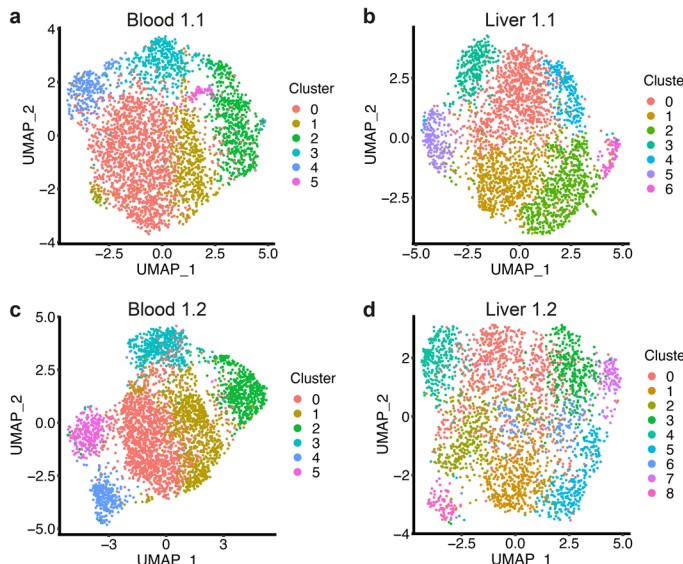

**Extended Data Fig. 4 | Clusters identified in individual blood or liver samples. a-d,** Individual sample UMAPs colored by the identified clusters—blood 1.1 (**a**), liver 1.1 (**b**), blood 1.2 (**c**), liver 1.2 (**d**).

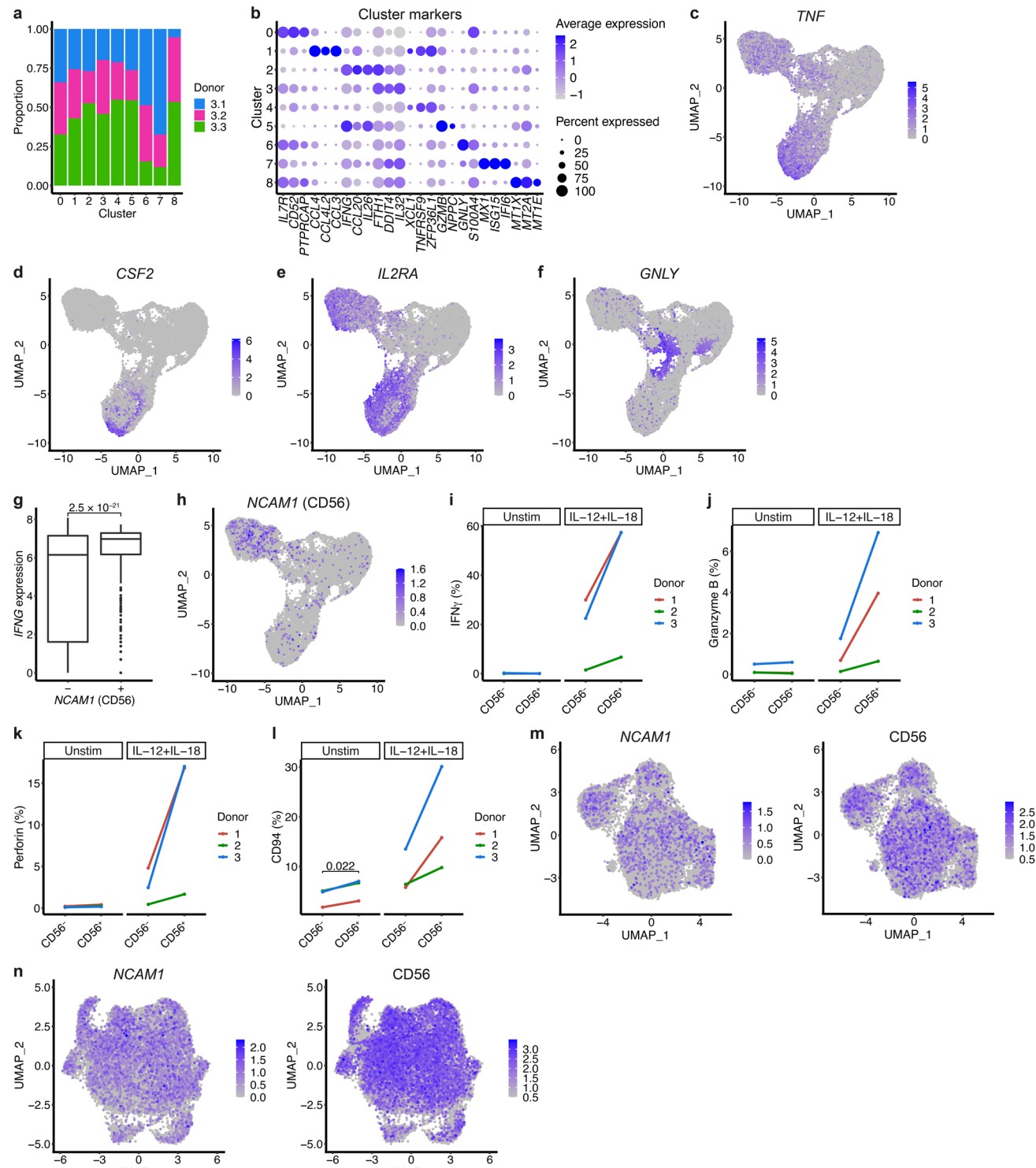

**Extended Data Fig. 5 | Gene and protein expression by TCR- and cytokine-stimulated MAIT cells. a**, Proportion of cells from each donor ($n = 3$) in each cluster (unstimulated, TCR-stimulated and cytokine-stimulated MAIT cells combined). **b**, Expression of the top three marker genes per cluster. Dot color indicates the level of gene expression and dot size indicates the percentage of cells expressing the gene. **c-f**, UMAPs colored by expression of *TNF* (**c**), *CSF2* (**d**), *IL2RA* (**e**) and *GNLY* (**f**). **g**, Expression of *IFNG* in cytokine-stimulated MAIT cells negative ($n = 11,122$ cells) and positive ($n = 387$ cells; log-transformed normalized expression >0) for the expression of *NCAM1* (CD56). Boxes span the 25th–75th percentiles, the midline denotes the median and whiskers extend to ±1.5 × IQR. Points indicate outliers. Two-sample Wilcoxon rank-sum test. **h**, UMAP colored by expression of *NCAM1* (CD56). **i-l**, Percentage of sorted CD56⁻ and CD56⁺ MAIT cells expressing IFNγ (**i**), granzyme B (**j**), perforin (**k**) and CD94 (**l**) when left unstimulated or stimulated with IL-12 + IL-18 for 20 h (as measured by flow cytometry). $n = 3$ donors. Two-sided paired *t*-test between CD56⁻ and CD56⁺ cells in both conditions (nonsignificant results omitted). **m,n**, CD56 gene (left) and protein (right) expression in blood (**m**) and liver (**n**) MAIT cells. Protein expression measured in Exp 2 only.

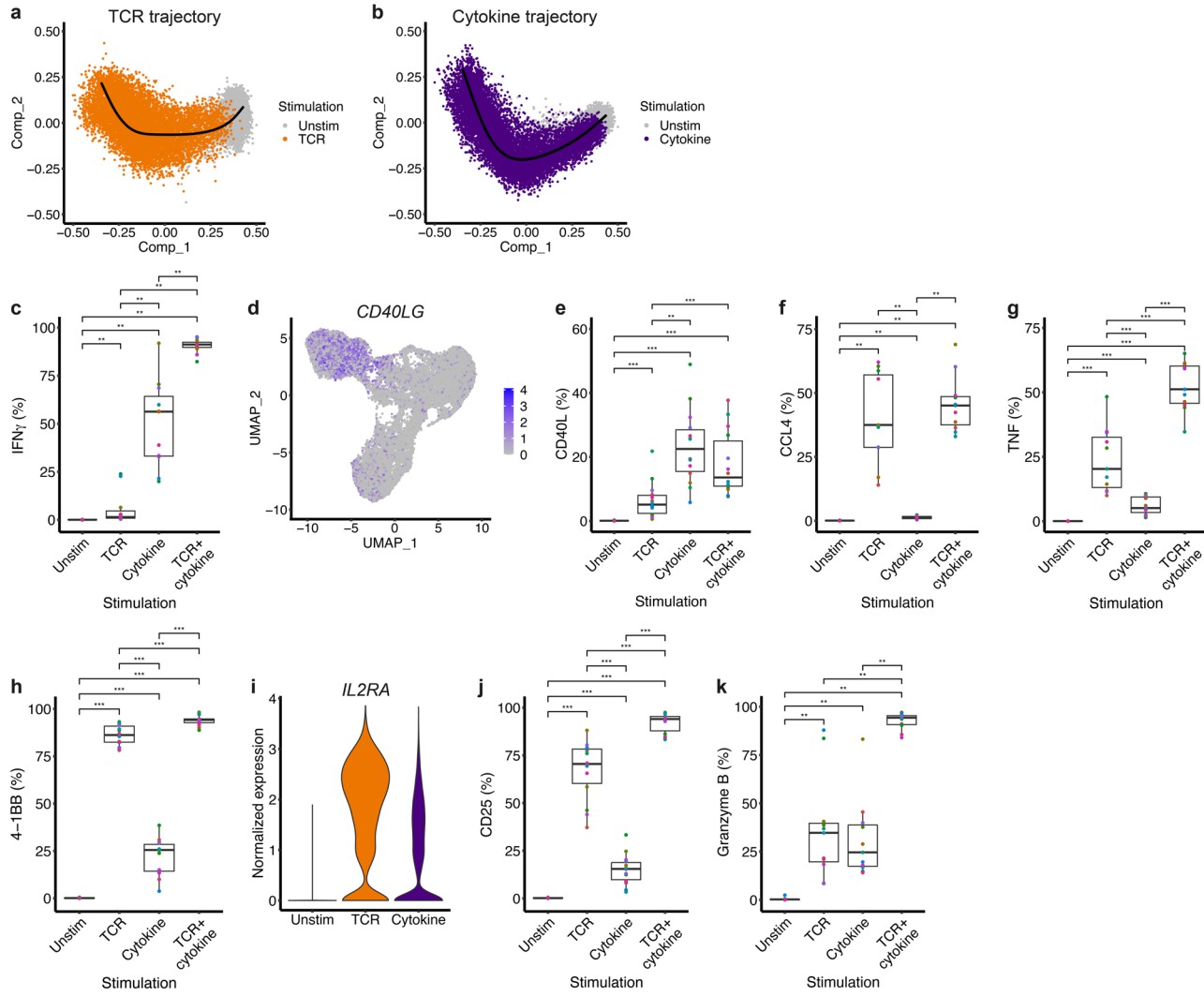

**Extended Data Fig. 6 | MAIT cell activation trajectories and validation of stimulus-specific and shared activation markers and cytokines.**
**a,b**, Multidimensional scaling plots of unstimulated and TCR-stimulated MAIT cells (**a**), or unstimulated and cytokine-stimulated MAIT cells (**b**). SCORPIUS TCR (**a**) and cytokine (**b**) trajectories are shown in black. $n$ = 3 donors. **c,e-h,j,k**, Percentage of MAIT cells expressing IFNγ (**c**), CD40L (**e**), CCL4 (**f**), TNF (**g**), 4-1BB (**h**), CD25 (**j**) and granzyme B (**k**) when left unstimulated or stimulated with plate-bound MR1/5-OP-RU (TCR), IL-12 + IL-18 (cytokine) or both (TCR+cytokine) for 20 h. Protein expression was measured by flow cytometry on all MAIT cells from

11 donors in two independent experiments (**c, f, g, k**) or CD4⁻ MAIT cells from 14 donors in three independent experiments (**e, h, j**). Boxes span the 25th–75th percentiles, the midline denotes the median and whiskers extend to ±1.5 × IQR. Two-sample Wilcoxon signed-rank test. Benjamini-Hochberg adjusted $P$ values are shown (nonsignificant results omitted). **P < 0.01, ***P < 0.001. **d**, UMAP of unstimulated, TCR-stimulated and cytokine-stimulated MAIT cells colored by expression of *CD40LG*. **i**, Violin plot showing expression of *IL2RA* by unstimulated, TCR-stimulated and cytokine-stimulated MAIT cells.

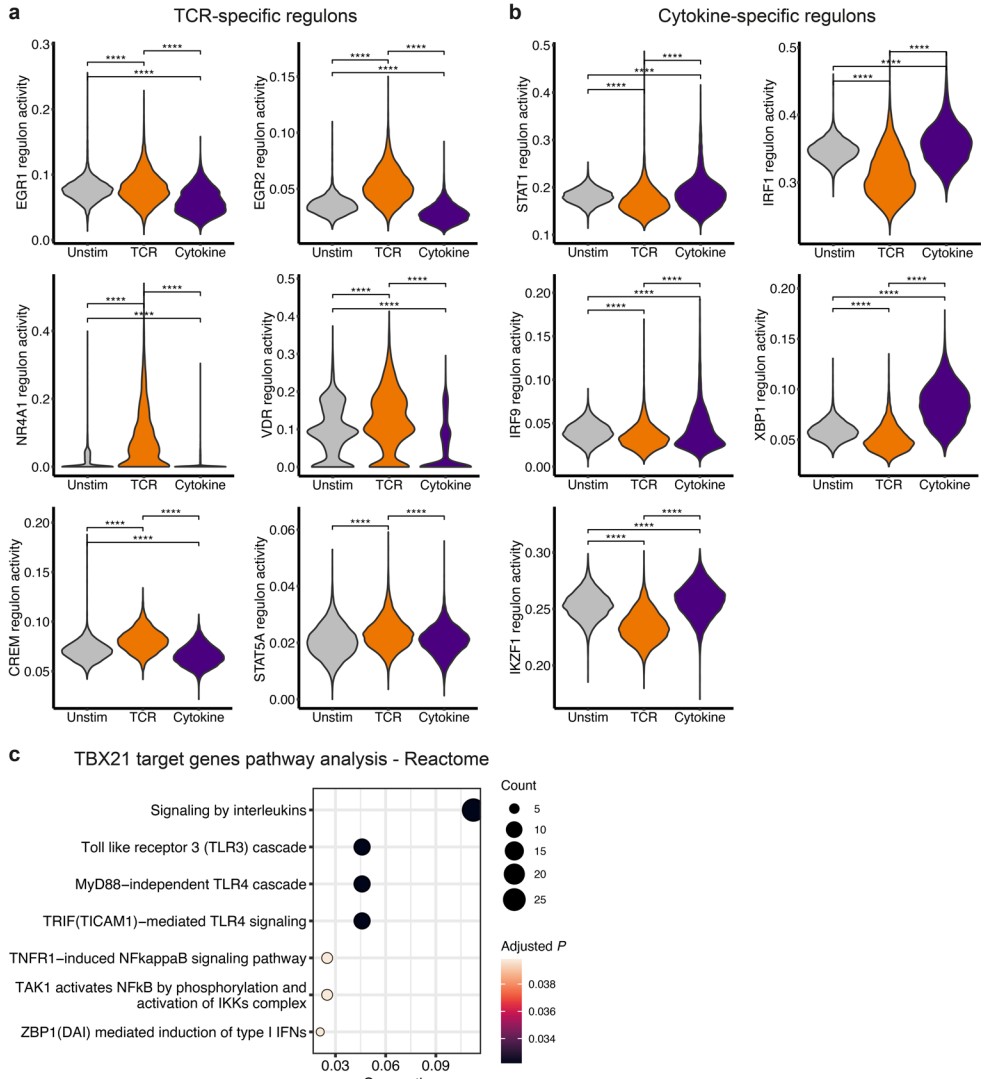

**Extended Data Fig. 7 | Transcriptional regulation of TCR- and cytokine-stimulated MAIT cells exhibits shared and distinct properties. a,b**, Violin plots showing the activity (AUCell scores) of selected TCR-specific (**a**) and cytokine-specific (**b**) transcription factor regulons in unstimulated, TCR-stimulated and cytokine-stimulated MAIT cells. $n = 3$ donors. Differential activity analysis was performed for all pairwise comparisons using MAST. Bonferroni adjusted $P$ values are shown (nonsignificant results omitted). ****$P < 0.0001$. **c**, Over-representation analysis on predicted T-bet (TBX21) target genes. Significant Reactome pathways and associated Benjamini-Hochberg adjusted $P$ values are shown.

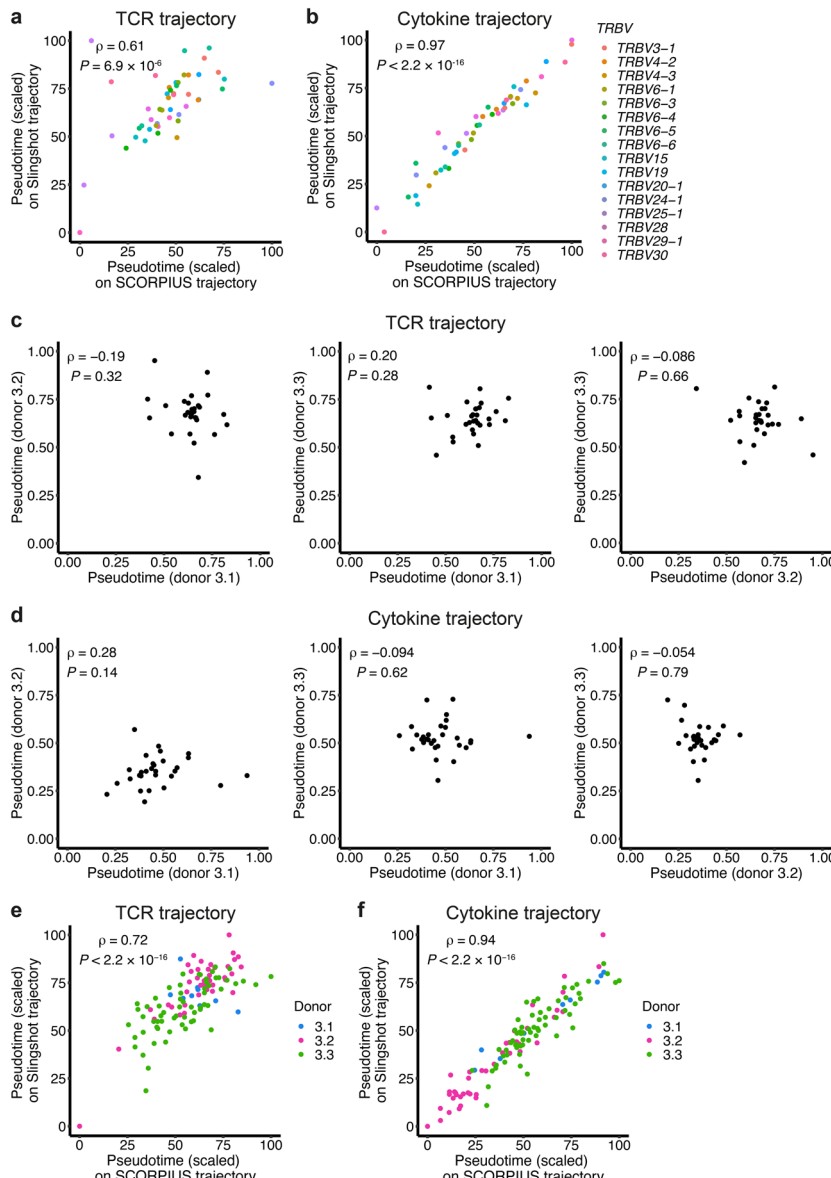

**Extended Data Fig. 8 | Influence of TCR usage on MAIT cell activation potential. a,b**, Spearman's rank correlation between average *TRBV* pseudotimes on SCORPIUS and Slingshot (two trajectory analysis methods) TCR (**a**) and cytokine (**b**) trajectories. Pseudotime values scaled between 0 and 100. **c,d**, Spearman's rank correlation between average *TRBV* pseudotimes for pairs of donors on SCORPIUS TCR (**c**) and cytokine (**d**) trajectories. **e,f**, Spearman's rank correlation between average clonotype pseudotimes on SCORPIUS and Slingshot TCR (**e**) and cytokine (**f**) trajectories. Pseudotime values scaled between 0 and 100. Plots show stimulated cells only, *TRBV* gene segments with a frequency >1% in any donor (**a-d**) and clonotypes containing ≥20 cells (**e, f**).

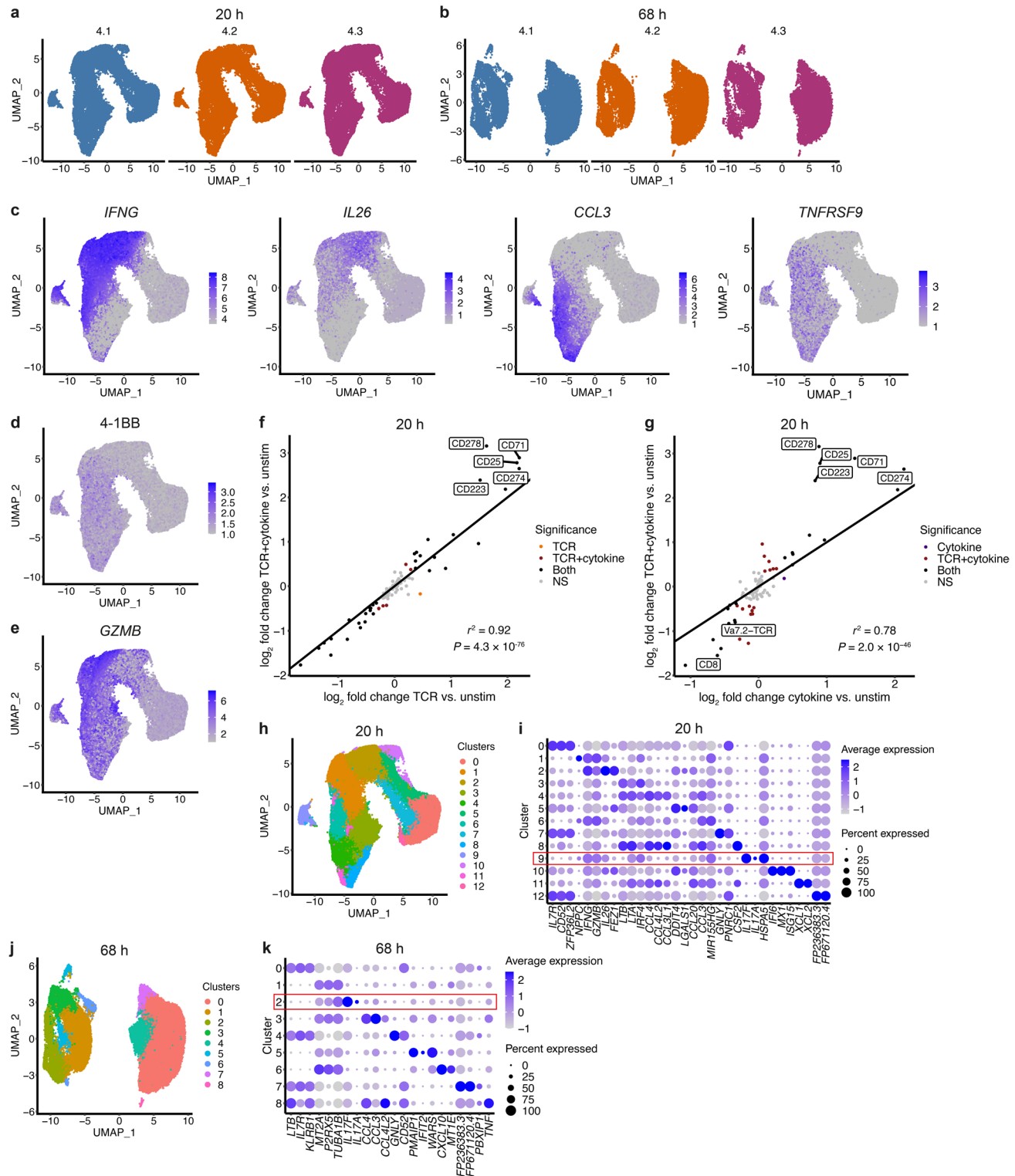

**Extended Data Fig. 9 | See next page for caption.**

**Extended Data Fig. 9 | TCR-, cytokine- and TCR+cytokine-stimulated MAIT cells. a,b**, UMAPs of 20 h- (**a**) and 68 h- (**b**) stimulated MAIT cells split and colored by donor. **c**, UMAPs of 20 h-stimulated MAIT cells colored by expression of *IFNG*, *IL26*, *CCL3* and *TNFRSF9*. **d**, UMAP of 20 h-stimulated MAIT cells colored by expression of 4-1BB protein. **e**, UMAP of 20 h-stimulated MAIT cells colored by expression of *GZMB*. **f**, Pearson's correlation between the log$_2$ fold change in protein expression between TCR-stimulated and unstimulated MAIT cells, and TCR+cytokine-stimulated and unstimulated MAIT cells. Labels highlight selected genes that were differentially regulated by the two stimuli. Point colors indicate whether the protein was significantly differentially expressed in response to TCR stimulation only (orange), TCR+cytokine stimulation only (maroon), both (black) or neither (NS; gray). **g**, Pearson's correlation between the log$_2$ fold change in protein expression between cytokine-stimulated and unstimulated MAIT cells, and TCR+cytokine-stimulated and unstimulated MAIT cells. Labels highlight selected genes that were differentially regulated by the two stimuli. Point colors indicate whether the protein was significantly differentially expressed in response to cytokine stimulation only (purple), TCR+cytokine stimulation only (maroon), both (black) or neither (NS; gray). **h**, UMAP of 20 h-stimulated MAIT cells colored by the 13 identified clusters. **i**, Dot plot showing the top three marker genes per cluster in **h**. Red box indicates the IL-17-expressing cluster. Dot color indicates the level of gene expression and dot size indicates the percentage of cells expressing the gene. **j**, UMAP of 68 h-stimulated MAIT cells colored by the nine identified clusters. **k**, Dot plot showing the top three marker genes per cluster in **j**. Red box indicates the IL-17-expressing cluster. Dot color indicates the level of gene expression and dot size indicates the percentage of cells expressing the gene.

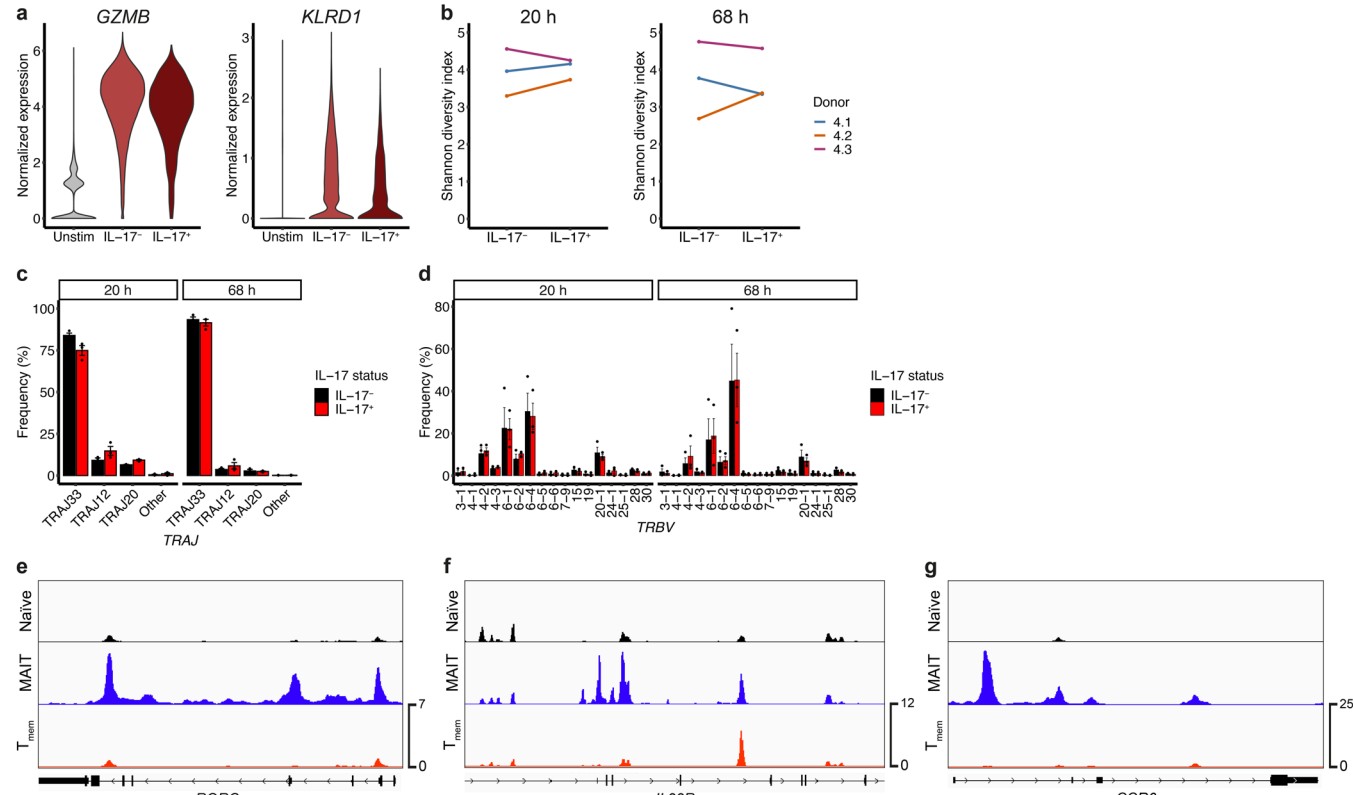

**Extended Data Fig. 10 | IL-17-expressing MAIT cells and type 17 gene loci.**
**a**, Violin plots showing *GZMB* (left) and *KLRD1* (CD94; right) expression by unstimulated MAIT cells, and IL-17⁻ and IL-17⁺ TCR+cytokine-stimulated MAIT cells at 20 h. **b**, Shannon diversity index for TCRαβ clonotypes in IL-17⁻ and IL-17⁺ MAIT cells from the TCR+cytokine stimulation condition at 20 h (left) and 68 h (right). Cell numbers for each donor were downsampled to ensure equal numbers of IL-17⁻ and IL-17⁺ cells at a given timepoint. **c,d**, Percentage of IL-17⁻ and IL-17⁺ TCR+cytokine-stimulated MAIT cells expressing different *TRAJ* (**c**) and *TRBV* (**d**) gene segments. *n* = 3 donors. *TRBV* genes expressed in >1% of IL-17⁻ or IL-17⁺ cells from any donor are included. Mean ± s.e.m. is shown. Two-sample Wilcoxon signed-rank test for all pairwise comparisons (nonsignificant results omitted). **e-g**, Representative ATAC-seq tracks showing *RORC* (**e**), *IL23R* (**f**) and *CCR6* (**g**) gene loci in naïve T (black), MAIT (blue) and T$_{mem}$ (red) cells.

# Reporting Summary

## Statistics

For all statistical analyses, confirm that the following items are present in the figure legend, table legend, main text, or Methods section.

| n/a | Confirmed | |
|---|---|---|
| ☐ | ☒ | The exact sample size (*n*) for each experimental group/condition, given as a discrete number and unit of measurement |
| ☐ | ☒ | A statement on whether measurements were taken from distinct samples or whether the same sample was measured repeatedly |
| ☐ | ☒ | The statistical test(s) used AND whether they are one- or two-sided *Only common tests should be described solely by name; describe more complex techniques in the Methods section.* |
| ☐ | ☒ | A description of all covariates tested |
| ☐ | ☒ | A description of any assumptions or corrections, such as tests of normality and adjustment for multiple comparisons |
| ☐ | ☒ | A full description of the statistical parameters including central tendency (e.g. means) or other basic estimates (e.g. regression coefficient) AND variation (e.g. standard deviation) or associated estimates of uncertainty (e.g. confidence intervals) |
| ☐ | ☒ | For null hypothesis testing, the test statistic (e.g. *F*, *t*, *r*) with confidence intervals, effect sizes, degrees of freedom and *P* value noted *Give P values as exact values whenever suitable.* |
| ☒ | ☐ | For Bayesian analysis, information on the choice of priors and Markov chain Monte Carlo settings |
| ☒ | ☐ | For hierarchical and complex designs, identification of the appropriate level for tests and full reporting of outcomes |
| ☐ | ☒ | Estimates of effect sizes (e.g. Cohen's *d*, Pearson's *r*), indicating how they were calculated |

*Our web collection on statistics for biologists contains articles on many of the points above.*

## Software and code

Policy information about availability of computer code

| | |
|---|---|
| Data collection | Flow cytometry data were collected using BD FACSDiva Software (v8.0.1). Sequencing data were collected using HiSeq Control Software (v2.2.68; Illumina) or NovaSeq Control Software (v1.6.0-1.8.0; Illumina). |
| Data analysis | Flow cytometry data were analysed using FlowJo (v10.8.1; BD Biosciences).<br><br>R version 4.1.1-4.2.0. R packages: AUCell (v1.16.0-1.20.1), biomaRt (v2.54.1), cellhashR (v1.0.3), clusterProfiler (v4.7.1), clustree (v0.4.3-0.5.0), ComplexHeatmap (v2.14.0), DropletQC (v0.0.0.9000), edgeR (v3.24.3), EMT (v1.3), ggplot2 (v3.3.4-3.4.2), ggpubr (v0.4.0), ggrepel (v0.9.1-0.9.3), ggseqlogo (v0.1), gridExtra (v2.3), Harmony (v0.1.1), MAST (v1.18.0-1.24.1), org.Hs.eg.db (v3.13.0-3.16.0), pheatmap (v1.0.12), plotrix (v3.8.2), ReactomePA (v1.36.0), readxl (v1.4.2), rstatix (v0.7.2), scales (v1.1.1-1.2.1), scater (v1.20.1-1.26.1), sctransform (v0.3.2-0.3.5), SCORPIUS (v1.0.8), Seurat (v4.0.3-4.3.0), Slingshot (v2.0.0), STACAS (v2.0.1), tidyverse (v1.3.1-2.0.0), vegan (v2.6.4), venn (v1.11), VennDiagram (v1.7.3), viridis (v0.6.1-0.6.2).<br><br>Python version 3.7.10-3.10.8. Python packages: pandas (v1.2.5-1.5.2), pySCENIC (v0.11.2-0.12.1).<br><br>Command-line software: bedtools (v2.26.0), Bowtie 2 (v2.3.4.1), Cell Ranger (v3.0.1-7.0.1; 10x Genomics), deepTools (v3.1.0-3.5.1), FastQC (v0.11.5), featureCounts (v1.6.0), HOMER (v4.8), MACS2 (v2.1.1), Picard (v2.15.0), samtools (v1.6), Trimmomatic (v0.36).<br><br>Other software: GraphPad Prism (v9.5.1; GraphPad Software, LLC), IMGT/JunctionAnalysis (v2.3.0; https://www.imgt.org/IMGT_jcta/analysis), GSEA (v4.3.2), Integrative Genomics Viewer (v2.16.0). |

For manuscripts utilizing custom algorithms or software that are central to the research but not yet described in published literature, software must be made available to editors and reviewers. We strongly encourage code deposition in a community repository (e.g. GitHub). See the Nature Portfolio guidelines for submitting code & software for further information.

## Data

Policy information about availability of data

All manuscripts must include a data availability statement. This statement should provide the following information, where applicable:

- Accession codes, unique identifiers, or web links for publicly available datasets
- A description of any restrictions on data availability
- For clinical datasets or third party data, please ensure that the statement adheres to our policy

Sequencing data generated in this study have been deposited in NCBI's Gene Expression Omnibus (GEO) and are accessible through GEO SuperSeries accession number GSE194189. The Bioconductor org.Hs.eg.db annotation package and ENSEMBL_MART_ENSEMBL BioMart database are publicly available.

## Research involving human participants, their data, or biological material

Policy information about studies with human participants or human data. See also policy information about sex, gender (identity/presentation), and sexual orientation and race, ethnicity and racism.

| | |
|---|---|
| Reporting on sex and gender | Blood-liver scRNA-seq experiments: the sex of liver patients is listed in Supplementary Table 1. Blood was additionally obtained from one male and four females. Stimulation scRNA-seq experiments: for Exp 3, blood was obtained from three females; for Exp 4, blood was obtained from two males and one female. Experiments were underpowered for formal sex-based analysis. Sex information is not available for the donors used in flow cytometry validation experiments. |
| Reporting on race, ethnicity, or other socially relevant groupings | Race and ethnicity information is not available for the donors in this study. |
| Population characteristics | Liver patient information, including age, sex, diagnosis and treatment is detailed in Supplementary Table 1. Age and gender were not accounted for in the analyses. Donor was included as a covariate for differential expression analyses between conditions e.g. cell types, tissues, stimuli. |
| Recruitment | Human participants were recruited at the Churchill Hospital, Oxford, UK and the University Hospital Basel, Basel, Switzerland. Patients had no chronic liver disease, active excess alcohol consumption (> 14 g/day), infection, immunosuppression or family history of liver disease. The requirement for written informed consent may introduce selection bias. Patient recruitment was influenced by practical considerations, such as the time of day of the surgery and the amount of liver tissue available. |
| Ethics oversight | Samples were obtained with written informed consent through the Oxford Gastrointestinal Illnesses Biobank (REC ref. 16/YH/0247) or under Ethikkommission Nordwest- und Zentralschweiz (EKNZ) numbers EKNZ-2014-362, EKNZ-2016-01188 and EKNZ-2019-02118. |

Note that full information on the approval of the study protocol must also be provided in the manuscript.

# Field-specific reporting

Please select the one below that is the best fit for your research. If you are not sure, read the appropriate sections before making your selection.

☒ Life sciences          ☐ Behavioural & social sciences          ☐ Ecological, evolutionary & environmental sciences

For a reference copy of the document with all sections, see nature.com/documents/nr-reporting-summary-flat.pdf

# Life sciences study design

All studies must disclose on these points even when the disclosure is negative.

| | |
|---|---|
| Sample size | Statistical methods were not used to predetermine sample size. scRNA-seq and flow cytometry experiments included ≥ 3 donors per group (tissue or stimulation condition). Initial blood-liver (Exp 1) and stimulation (Exp 3) scRNA-seq experiments were followed by validation experiments (Exp 2 and 4) – the findings of these were highly concordant. |
| Data exclusions | No samples were excluded from scRNA-seq, ATAC-seq or flow cytometry datasets.<br><br>scRNA-seq cell exclusions: cells with low unique molecular identifier counts, low gene counts and/or a high percentage of mitochondrial reads were assumed to be dead/dying cells and were removed. For Exp 1 and 2, cells labelled as empty droplets or damaged cells by DropletQC were removed (damaged cells in Exp 4 also removed). For Exp 2 and 4, only cells called as consensus singlets by hashtag demultiplexing (cellhashR) were retained. Cells with two TCRα and two TCRβ chains, or more than two TCRα and/or TCRβ chains, were assumed to be doublets and discarded. |
| Replication | Initial blood-liver (Exp 1) and stimulation (Exp 3) scRNA-seq experiments were followed by validation experiments (Exp 2 and 4) – the findings of these were highly concordant. Genes of interest from Exp 1 and 3 were validated at the protein level by CITE-seq (Exp 2 and 4) and flow cytometry. As described in the methods, pySCENIC transcription factor regulon analysis was performed 100 times and results aggregated to define high-confidence regulons. |

ATAC-seq was performed once with three donors.

Activation markers and cytokines identified by scRNA-seq were validated by flow cytometry in 14 donors (three independent experiments) and 11 donors (two independent experiments), respectively. Stimulation of sorted CD56- and CD56+ MAIT cells was performed once with three donors. All attempts at replication were successful.

Randomization | For blood-liver scRNA-seq experiments, randomization to groups was not possible since liver patients were compared with healthy donors. For stimulation experiments, cells from each donor were present in all conditions.

Blood-liver scRNA-seq experiments (Exp 1 and 2) were each performed in two batches - donors were randomly distributed across batches. Stimulation scRNA-seq experiments (Exp 3 and 4) were each performed as a single batch. Donor was included as a covariate in differential expression analyses.

Blinding | Experiments and analyses were not performed blinded as the same investigator performed sample collection, sample processing, data generation and data analysis.

# Reporting for specific materials, systems and methods

We require information from authors about some types of materials, experimental systems and methods used in many studies. Here, indicate whether each material, system or method listed is relevant to your study. If you are not sure if a list item applies to your research, read the appropriate section before selecting a response.

## Materials & experimental systems

| n/a | Involved in the study |
|---|---|
| ☐ | ☒ Antibodies |
| ☒ | ☐ Eukaryotic cell lines |
| ☒ | ☐ Palaeontology and archaeology |
| ☒ | ☐ Animals and other organisms |
| ☒ | ☐ Clinical data |
| ☒ | ☐ Dual use research of concern |
| ☒ | ☐ Plants |

## Methods

| n/a | Involved in the study |
|---|---|
| ☒ | ☐ ChIP-seq |
| ☐ | ☒ Flow cytometry |
| ☒ | ☐ MRI-based neuroimaging |

## Antibodies

Antibodies used | Specificity; fluorophore; clone; manufacturer; catalogue number; RRID

Anti-human CCL4; APC; REA511; Miltenyi Biotec; 130-129-199; AB_2922004
Anti-human CCR7; PE-CF594; 150503; BD Biosciences; 562381; AB_11153301
Anti-human CD3ε; APC-Cy7; HIT3A; BioLegend; 300318; AB_314054
Anti-human CD3ε; APC-Cy7; UCHT1; BioLegend; 300426; AB_830755
Anti-human CD3ε; BV650; UCHT1; BioLegend; 300468; AB_2629574
Anti-human CD3ε; BV650; UCHT1; BioLegend; 300464; AB_2566036
Anti-human CD3ε; FITC; UCHT1; BioLegend; 300406; AB_314060
Anti-human CD3ε; PerCP-Cy5.5; UCHT1; BioLegend; 300430; AB_893299
Anti-human CD4; BV650; OKT4; BioLegend; 317436; AB_2563050
Anti-human CD4; BV711; OKT4; BioLegend; 317440; AB_2562912
Anti-human CD4; BV785; OKT4; BioLegend; 317442; AB_2563242
Anti-human CD8α; AF700; SK1; BioLegend; 344724; AB_2562790
Anti-human CD8α; BV650; SK1; BioLegend; 344730; AB_2564510
Anti-human CD14; APC-Cy7; M5E2; BioLegend; 301820; AB_493695
Anti-human CD14; FITC; M5E2; BioLegend; 301804; AB_314186
Anti-human CD19; APC-Cy7; HIB19; BioLegend; 302218; AB_314248
Anti-human CD19; FITC; HIB19; BioLegend; 302206; AB_314236
Anti-human CD25; PerCP-Cy5.5; BC96; BioLegend; 302626; AB_2125478
Anti-human CD26; APC; BA5b; BioLegend; 30271; AB_10916120
Anti-human CD45; AF700; HI30; BioLegend; 304024; AB_493761
Anti-human CD45RO; BV786; UCHL1; BD Biosciences; 564290; AB_2738733
Anti-human CD56; BV421; HCD56; BioLegend; 318328; AB_11218798
Anti-human CD69; PerCP-Cy5.5; FN50; BioLegend; 310926; AB_2074956
Anti-human CD94; FITC; DX22; BioLegend; 305504; AB_314534
Anti-human CD137 (4-1BB); BV421; 4B4-1; BD Biosciences; 564091; AB_2722503
Anti-human CD154 (CD40L); BV785; 24-31; BioLegend; 310842; AB_2572187
Anti-human CD161; APC; 191B8; Miltenyi Biotec; 130-113-590; AB_2733346
Anti-human CD161; PE; 191B8; Miltenyi Biotec; 130-113-592; AB_2733625
Anti-human CD161; PE-Vio770; 191B8; Miltenyi Biotec; 130-113-594; AB_2751134
Anti-human Granzyme B; AF647; GB11; BioLegend; 515406; AB_2566333
Anti-human Granzyme B; Pacific Blue; GB11; BioLegend; 515408; AB_2562196
Anti-human IFNγ; BV605; 4S.B3; BioLegend; 502536; AB_2563881
Anti-human IFNγ; BV711; 4S.B3; BioLegend; 502540; AB_2563506

Anti-human Perforin; BV711; dG9; BioLegend; 308130; AB_2687190
Anti-human TCR γδ; BV480; B1; BD Biosciences; 566076; AB_2739491
Anti-human TCR γδ; FITC; B1; BD Biosciences; 559878; AB_397353
Anti-human TCR Vα7.2; PE-Cy7; 3C10; BioLegend; 351712; AB_2561994
Anti-human TCR Vα24-Jα18; FITC; 6B11; BioLegend; 342906; AB_1731856
Anti-human TCR Vδ2; FITC; B6; BioLegend; 331406; AB_1089230
Anti-human TNF; BV605; MAb11; BioLegend; 502909; AB_315261

Validation

All antibodies are commercially available and validation statements can be found on the manufacturers' websites using the catalogue number or in the Antibody Registry database (https://antibodyregistry.org) using the RRID. Antibodies were titrated to achieve optimal separation between negative and positive populations.

# Flow Cytometry

## Plots

Confirm that:

☒ The axis labels state the marker and fluorochrome used (e.g. CD4-FITC).

☒ The axis scales are clearly visible. Include numbers along axes only for bottom left plot of group (a 'group' is an analysis of identical markers).

☒ All plots are contour plots with outliers or pseudocolor plots.

☒ A numerical value for number of cells or percentage (with statistics) is provided.

## Methodology

Sample preparation

Liver tissue collection and processing (Exp 1 and 2):
Liver tissue (n=7) and matched blood (n=6) were obtained from patients undergoing liver resection at the Churchill Hospital, Oxford, UK and the University Hospital Basel, Basel, Switzerland (Supplementary Table 1). Patients had no chronic liver disease, active excess alcohol consumption (> 14 g/day), infection, immunosuppression or family history of liver disease.

Disease-free liver tissue was collected from the resection margin, cut into small pieces with a scalpel, and ground through a 70 μm cell strainer. Cells were washed with R10 (RPMI-1640 [Sigma-Aldrich], 10% FBS [Sigma-Aldrich], 1% penicillin-streptomycin [Thermo Fisher Scientific]; 931g, 10 min, 4 °C) and mononuclear cells isolated by density gradient centrifugation on a discontinuous 35%/70% Percoll (GE Healthcare) gradient (931g, 20 min, 21 °C, no brake). Mononuclear cells were collected from the interface and washed with R10 (596g, 10 min, 4 °C). Residual red blood cells were lysed with ACK for 3-5 min. Cells were washed twice (596g, 10 min, 4 °C) and cryopreserved (90% FBS, 10% DMSO [Sigma-Aldrich]) in liquid nitrogen.

Ethics statement:
Samples were obtained with written informed consent through the Oxford Gastrointestinal Illnesses Biobank (REC ref. 16/YH/0247) or under Ethikkommission Nordwest- und Zentralschweiz (EKNZ) numbers EKNZ-2014-362, EKNZ-2016-01188 and EKNZ-2019-02118.

Peripheral blood mononuclear cell (PBMC) isolation:
PBMCs were isolated from fresh whole blood by density gradient centrifugation (Lymphoprep, Axis-Shield) at 931g for 30 min with no brake. Cells were cryopreserved in liquid nitrogen and thawed in complete medium (R10, 1X nonessential amino acids [Thermo Fisher Scientific], 1 mM sodium pyruvate [Thermo Fisher Scientific], 10 mM HEPES [pH 7.0-7.5; Thermo Fisher Scientific], 50 μM β-mercaptoethanol [Thermo Fisher Scientific]) on the day of use.

Stimulation of isolated CD8+/CD3+ T cells for scRNA-seq and scTCR-seq (Exp 3 and 4) or activation marker/cytokine validation:
Pierce streptavidin-coated high-capacity flat-bottom 96-well plates (Thermo Fisher Scientific) were coated with 50 μl biotinylated MR1/5-OP-RU monomer (NIH Tetramer Core Facility) at 10 μg/ml in PBS (Sigma-Aldrich) overnight at 4 °C. Cryopreserved PBMCs were thawed in complete medium. CD8+ T cells were isolated using CD8 MicroBeads (Exp 3; Miltenyi Biotec) and CD3+ T cells using the REAlease CD3 MicroBead Kit (Exp 4 and validation experiments; Miltenyi Biotec) following the manufacturer's instructions. Isolated CD8+/CD3+ T cells were washed in complete medium and resuspended at 1 × 107 cells/ml. One million (20 h stimulation) or 500,000 (68 h stimulation) cells were added per well to the appropriate 96-well plates (MR1/5-OP-RU-coated plate for TCR and TCR+cytokine stimulation, round-bottom plate for unstimulated and cytokine stimulation). IL-12 (50 ng/ml; R&D Systems) and IL-18 (50 ng/ml; R&D Systems) were added for cytokine stimulation; αCD28 (1 μg/ml; clone: CD28.2; BioLegend) for TCR stimulation; IL-12, IL-18, and αCD28 for TCR+cytokine stimulation; and complete medium for unstimulated cells (final volume 200 μl/well). Cells were incubated for 20 h or 68 h at 37 °C, 5% CO2. For intracellular cytokine staining, brefeldin A (BioLegend) and monensin (BioLegend) were added for the final 4 h.

Tetramer staining (Exp 1 and 2):
Biotinylated human MR1/5-OP-RU and MR1/6-FP monomers were provided by the NIH Tetramer Core Facility. Tetramers were generated using streptavidin-PE (high concentration) or streptavidin-BV421 (both BioLegend) following the NIH Tetramer Core Facility protocol. Tetramer staining was performed for 40 min at 21 °C in FACS buffer (PBS, 0.5% BSA [Sigma-Aldrich], 1 mM EDTA [Sigma-Aldrich]).

Surface staining and cell sorting for scRNA-seq and scTCR-seq (Exp 1-4):
TotalSeq-C hashtag antibodies (BioLegend) were used in Exp 2 and 4. Hashtag antibody dilutions were prepared according to the manufacturer's instructions. Namely, antibody vials were centrifuged at 10,000g, 30 s, 4 °C, before antibody dilution in

FACS buffer. Diluted hashtags were centrifuged at 14,000g, 10 min, 4 °C. Cells were incubated in Human TruStain FcX (BioLegend) for 10 min at 4 °C before the addition of diluted hashtag antibodies (0.2 µg/well) for 10 min at 4 °C. Surface fluorochrome-conjugated antibodies were added without washing off the hashtag antibodies. Surface staining was performed in Brilliant Stain Buffer Plus (BD Biosciences) for 30 min at 4 °C. Cells were washed twice in PBS with 0.5% BSA, resuspended in presort buffer (PBS, 1% BSA, 25 mM HEPES) containing 3-5 nM SYTOX Green Nucleic Acid Stain (Thermo Fisher Scientific) and incubated for 20 min at 4 °C. Cells were sorted on a BD FACSAria III with an 85 µm nozzle. Sorted cells were collected in RPMI-1640, 10% FBS, 25 mM HEPES, or HBSS (Thermo Fisher Scientific), 50% FBS, 25 mM HEPES. Sort purity was > 99%. For Exp 2 and 4, sorted cells were stained with the TotalSeq-C Human Universal Cocktail V1.0 (BioLegend) according to the manufacturer's instructions. Staining reagents are listed in Supplementary Table 12.

Stimulation of CD56- and CD56+ MAIT cells:
CD3+ T cells were isolated using the REAlease CD3 MicroBead Kit following the manufacturer's instructions. Surface antibody and live/dead (SYTOX Green Nuclear Acid Stain) staining were performed as above, then CD56- and CD56+ MAIT cells (Vα7.2 +CD161hi) sorted on a BD FACSAria III with an 85 µm nozzle. Sorted cells were collected in HBSS, 50% FBS, 25 mM HEPES, then centrifuged at 400g, 5 min, 21 °C and incubated overnight at 37 °C, 5% CO2. Rested cells were washed in complete medium, plated in a 96-well round-bottom plate and stimulated with IL-12 (50 ng/ml) and IL-18 (50 ng/ml) at 37 °C, 5% CO2 for 20 h, with the addition of brefeldin A and monensin for the final 4 h.

Surface marker and intracellular cytokine staining for flow cytometry:
Surface staining was performed in Brilliant Stain Buffer Plus for 30 min at 4 °C. Stained cells were washed twice in FACS buffer. For intracellular cytokine staining, cells were fixed in Cytofix/Cytoperm (BD Biosciences) for 20 min at 4 °C, then washed twice in 1X Perm/Wash (BD Biosciences). Intracellular staining was performed in 1X Perm/Wash for 30 min at 4 °C. Cells were acquired on a BD LSR II flow cytometer with BD FACSDiva Software (v8.0.1). Staining reagents are listed in Supplementary Table 12.

| | |
|---|---|
| Instrument | Cells sorting was performed on a BD FACSAria III with an 85 µm nozzle. Cells were analysed on a BD LSR II. |
| Software | FACSDiva (v8.0.1; BD Biosciences) for data collection, FlowJo (v10.8.1; BD Biosciences) for data analysis. |
| Cell population abundance | Postsort purity, determined for representative samples, was > 99%. |
| Gating strategy | See Supplementary Figures 1-3.<br><br>Exp 1 and 2 - FSC-A vs. SSC-A (lymphocytes), CD45 vs. SSC-A (hematopoietic), FSC-H vs. FSC-A (singlets), CD3ε vs. Dump (live CD3+ T cells; Dump = CD14, CD19, TCR γδ, TCR Vα24-Jα18, TCR Vδ2, SYTOX Green Nucleic Acid Stain). MAIT cells are MR1/5-OP-RU+ and Tmem cells are MR1/5-OP-RU-CCR7-.<br><br>Exp 3 - FSC-A vs. SSC-A (lymphocytes), CD3ε vs. Dump (live CD3+ T cells; Dump = CD14, CD19, TCR γδ, TCR Vα24-Jα18, TCR Vδ2, SYTOX Green Nucleic Acid Stain), FSC-H vs. FSC-A (singlets), CD4 vs. CD8α (CD8+ T cells). MAIT cells are CD26 +CD161hiVα7.2+.<br><br>Exp 4 - FSC-A vs. SSC-A (lymphocytes), CD3ε vs. Dump (live CD3+ T cells; Dump = CD14, CD19, TCR γδ, TCR Vα24-Jα18, TCR Vδ2, SYTOX Green Nucleic Acid Stain), FSC-H vs. FSC-A (singlets). MAIT cells are CD26+CD161hiVα7.2+.<br><br>CD56- and CD56+ MAIT cell sorting and poststimulation analysis:<br>Sorting - FSC-A vs. SSC-A (lymphocytes), CD3ε vs. Dump (live CD3+ T cells; Dump = CD14, CD19, TCR γδ, TCR Vα24-Jα18, TCR Vδ2, SYTOX Green Nucleic Acid Stain), FSC-H vs. FSC-A (singlets). CD56- and CD56+ MAIT cells are Vα7.2+CD161hiCD56- and Vα7.2+CD161hiCD56+, respectively.<br>Poststimulation analysis - FSC-H vs. FSC-A (singlets), FSC-A vs. SSC-A (lymphocytes), CD3ε vs. live/dead (live CD3+ T cells). CD56- and CD56+ MAIT cells are Vα7.2+CD161hiCD56- and Vα7.2+CD161hiCD56+, respectively.<br><br>Stimulation and activation marker staining - FSC-H vs. FSC-A (singlets), FSC-A vs. SSC-A (lymphocytes), CD3ε vs. Dump (live CD3+ T cells; Dump = CD14, CD19, TCR γδ, TCR Vα24-Jα18, TCR Vδ2, SYTOX Green Nucleic Acid Stain). MAIT cells are CD26 +CD161hiVα7.2+.<br><br>Stimulation and cytokine staining - FSC-H vs. FSC-A (singlets), FSC-A vs. SSC-A (lymphocytes), CD3ε vs. live/dead (live CD3+ T cells). MAIT cells are Vα7.2+CD161hi.<br><br>ATAC-seq - FSC-A vs. SSC-A (lymphocytes), FSC-A vs. Dump (non-monocyte, non-B, live; Dump = CD14, CD19, live/dead), FSC-H vs. FSC-A (singlets), CD3ε vs. TCR γδ (non-γδ T cells), CD4 vs. CD8α (CD8+ T cells). Naive T cells are CD45RO-CCR7+, MAIT cells are CCR7-MR1/5-OP-RU+ and Tmem cells are CCR7-MR1/5-OP-RU-. |

☒ Tick this box to confirm that a figure exemplifying the gating strategy is provided in the Supplementary Information.

