## [Peer Review File · Nature Immunology]

Peer Review Information

Journal: Nature Immunology

Manuscript Title: Single-cell analysis of human MAIT cell transcriptional, functional, and clonal diversity

Corresponding author name(s): Dr Lucy Garner, Professor Paul Klenerman

Reviewer Comments & Decisions:

Decision Letter, initial version:
--

18th Feb 2022

Dear Lucy & Paul,

Thank you for providing a point-by-point response to the referees comments on your Resource manuscript entitled, "Human MAIT cells show clonal diversity but transcriptional and functional homogeneity". As noted previously, while they find your work of considerable potential interest, they have raised quite substantial concerns that must be addressed - in particular to increase the number of human donors for the paired blood-liver analyses, and for functional validations of some of the relevant/interesting gene products identified in the transcriptomic analysis. In light of these comments, we cannot accept the manuscript for publication, but would be very interested in considering a revised version as proposed in your response letter.

We invite you to submit a substantially revised manuscript, however please bear in mind that we will be reluctant to approach the referees again in the absence of major revisions.

Specifically, the revision should include new experiments to address (as indicated in your response):

- scRNA-seq and scTCR-seq on matched blood and liver MAIT cells from three further donors, using hashing and/or CITE-seq antibodies to distinguish CD4+, CD8+ and DN MAIT cells.
- A repeat of the fRNA-seq experiment on three new healthy donors, with an additional TCR + cytokine stimulation condition.
- Flow cytometry for protein-level validation of key transcriptomic differences between blood and liver MAIT cells, and TCR- and cytokine-activated MAIT cells.
- Flow cytometry to measure the expression of IFN γ (and other activation markers) following cytokine stimulation of sorted CD56+ and CD56- MAIT cells.
- siRNA knockdown of candidate stimulus-specific transcription factors identified by regulon analysis e.g. HIVEP3, BATF.

Please include the additional textual clarifications as indicated in your response letter.

When you revise your manuscript, please take into account all reviewer and editor comments, please highlight all changes in the manuscript text file in Microsoft Word format.

Additionally, referee #1 pointed out to us that last week a similar paper was published, "presenting somewhat different conclusions:

<https://www.jimmunol.org/content/early/2022/02/11/jimmunol.2100522>

This paper by Vorkas et al sort of challenges the conclusions by Garner et al, as Vorkas finds evidence of heterogeneity and discrete subsets." We would like you to respond to this comment as well.

* If you have not done so already please begin to revise your manuscript so that it conforms to our Resource format instructions at <http://www.nature.com/ni/authors/index.html>. Refer also to any guidelines provided in this letter.

The Reporting Summary can be found here:

When submitting the revised version of your manuscript, please pay close attention to our [href="https://www.nature.com/nature-research/editorial-policies/image-integrity">Digital Image Integrity Guidelines. and to the following points below:](https://www.nature.com/nature-research/editorial-policies/image-integrity)

[REDACTED]

If you wish to submit a suitably revised manuscript we would hope to receive it within 6 months. If you cannot send it within this time, please let us know. We will be happy to consider your revision so long as nothing similar has been accepted for publication at Nature Immunology or published elsewhere.

Nature Immunology is committed to improving transparency in authorship. As part of our efforts in this direction, we are now requesting that all authors identified as 'corresponding author' on published papers create and link their Open Researcher and Contributor Identifier (ORCID) with their account on the Manuscript Tracking System (MTS), prior to acceptance. ORCID helps the scientific community achieve unambiguous attribution of all scholarly contributions. You can create and link your ORCID from the home page of the MTS by clicking on 'Modify my Springer Nature account'. For more information please visit www.springernature.com/orcid.

Thank you for the opportunity to review your work.

Sincerely,

Laurie

Laurie A. Dempsey, Ph.D.
Senior Editor
Nature Immunology
l.dempsey@us.nature.com
ORCID: 0000-0002-3304-796X

Referee expertise:

Referee #1: Innate-like T cells

Referee #2: Innate-like T cells

Referee #3: Innate-like T cells

Reviewers' Comments:

Reviewer #1:

Remarks to the Author:

This paper by Garner et al. presents a comprehensive investigation into the single cell transcriptome of human MAIT cells. The paper presents two complementary data sets; The first is based on paired

blood and liver samples from three donors, and the second one is based on blood MAIT cells activated through TCR or via IL-12+IL-18. In addition to the scRNAseq they have also nicely utilized the 10X platform to generate complementary scTCR sequencing from the same cells. This allows analysis of the TCR alpha and beta chain pairing at the single cells level, and also analysis of the transcriptional signature of different TCR clonotype cells. Overall, the paper is very well written and presented. This reviewer believes it can be a useful resource for the field, but still have some concerns and suggestions that may help improve the paper.

The overall message in the title and abstract is that "human MAIT cells show clonal diversity but transcriptional and functional homogeneity". The statement about clonal diversity is accurate although perhaps not novel as such. Perhaps a more interesting aspect is the potential link between clonotypes and responsiveness. I will comment more on that below. The transcriptional homogeneity is well supported, although perhaps a more interesting message is the consistent pattern in all three donors that the two tissue sites are associated with distinct transcriptional profiles while the TCR repertoire is shared between the sites within each donor. The final part of the title "functional homogeneity" is not well supported in this data set as function was strictly speaking not assessed. Rather, the authors performed TCR and cytokine stimulations and assessed the transcriptional response.

It is very nice to have the paired blood-liver samples. At the same time data from only three pairs is a bit limited. For example, the interesting findings that the MAIT cell TCR repertoire show a high degree of intra-donor similarity between blood and liver (Fig 3 a), while there is little sharing of clonotypes between donors, would be much more solid if resting on more than three donors. Likewise, the striking difference between the TCRalpha and beta clonotypes, where most of the TCRalpha clonotypes are public while there are very few if any public TCRbeta clones, would be much stronger with a few additional donors. A higher number of blood-liver pairs may solidify or refute some of these patterns.

The authors themselves point out the limitation that the blood and liver tissue samples come from cancer patients with liver metastases. It is well known that chronic conditions of different kinds can have significant effects on the MAIT cell compartment, including the activation state. How can we be sure that the current data set does not reflect a disease state of the MAIT cell compartment? Perhaps comparisons between the blood data from the blood-liver pairs, and the unstimulated control data from the healthy donor blood in vitro stimulation data set can be helpful?

In a way the conclusions presented about homogeneity in the MAIT cell compartment is very well aligned with the literature, as there is strikingly little phenotypic diversity among MAIT cells in blood. A couple of studies have reported differences in MAIT cell functional phenotype depending on expression of CD8a, CD8b or CD4 (or lack of expression i.e. DN). In general, the lack of separation of these coreceptor genes in the present UMAPs does not really stand in contrast with published phenotypic differences and apoptosis sensitivity. One can always discuss the definition of what is a subset, but the lack of finding one based on transcriptomics alone does not convincingly show the absence. For the CD4+ MAIT cells that is probably even more true as CD4 transcripts can be hard to detect efficiently.

The data with variable cluster bias of different clonotypes in Fig. 5 is interesting (also in the light of the cluster association with activation potential in Fig. 8). Just double checking that the clustering in Fig. 5 was done without the TCR data, so it is not so that the clustering is dependent on the TCR data (i.e. the finding generating itself so to say)?

The data in Fig. 6 is consistent with the literature and adds considerable detail to the different types of responses triggered by TCR vs cytokine stimulus. The identification of predicted transcription factor regulons associated with the two types of stimuli, the HIVEP3- and BATF-dependent regulons, may prove to be the most interesting finding in the paper. The reader of course wonders if the authors have been able to experimentally test the importance of these transcription factors?

The apparent differences between TCR clonotypes in responsiveness to both TCR stimulus and cytokine stimulus is interesting, and reasonably consistent with previous findings that the TCRbeta chain can influence MAIT cell triggering. The finding that responsiveness to also cytokine stimulation differs between clonotypes is puzzling. Does this somehow imply that a previous TCR-dependent experience will make some MAIT cells more prone to cytokine responsiveness?

Reviewer #2:

Remarks to the Author:

Title: Human MAIT cells show clonal diversity but transcriptional and functional homogeneity

Summary:

Garner and colleagues present a valuable resource and comprehensive analysis of paired single cell transcriptomic and TCR sequence data from human blood and liver MAIT cells (MR1/5-OP-RU+). The authors include an analysis of blood MAIT cells at rest and post TCR- or cytokine-mediated stimulation from an additional cohort, termed 'functional RNA-seq'.

The authors reveal that MAIT cells from blood and liver express distinct transcriptomes, namely liver MAIT cells possessed a tissue-resident-like transcriptional and regulatory profile. Despite this, between donors and within distinct tissues, MAIT cells expressed transcriptional homogeneity. As expected, a high degree of conservation existed among TCRA clonotypes between tissues and donors. In contrast, TCRab clonotypes were diverse and private but shared across tissues. Some private TCRab clonotypes appeared to associate with specific transcriptional clusters. The authors reveal that blood MAIT cells progress along distinct, consistent trajectories upon TCR or cytokine stimulation that is somewhat related to private TCR clonotypes, but likely not solely TCR signal strength.

The paper is well written, with appropriate methodology and statistics, and provides a large dataset from which the field can draw knowledge (as the authors claim, their study includes analysis of more MAIT cells than previous studies and single cell rather than bulk sequencing). The study also brings together scRNAseq, TCR sequencing and analysis of both activated and unstimulated cells from blood and liver. These provide an additional data and analyses while transcriptomic analysis of human MAIT cells is still relatively new. Certainly there is a lack of comprehensive analysis of MAIT cells across various tissues and in vitro stimulation conditions. However, the novel findings and value as an overarching resource for the MAIT cell field need to be better highlighted. Specific comments are provided below.

Major comments:

1. In figure 1 and figures 4-5 human MAIT cells are compared to gene signatures from murine T cells. If available, the manuscript would benefit from a comparison between relevant human T cell subsets. For example, human Th1 and Th17 cells, or human liver-derived memory T cells. Tissue residency gene signatures are available for human samples.

2. The authors acknowledge several times that their findings are consistent with previously published literature. Specifically, tissue resident signatures being more abundant in liver than blood was reported using bulk RNAseq (Salou 2019). The lack of distinct MAIT1 and MAIT17 subsets in humans was reported by Leeansyah 2015 and Dias 2017). The diverse TCRb and fixed TCRA repertoire is also known. However, comparisons with memory T cells are interesting and will be important to the field in understanding MAIT cell responses and memory, particularly given the interest in vaccination strategies targeting these cells. As above, the text could better clarify what is novel here and the implications for understanding MAIT cell biology.

3. One issue with this dataset and analysis serving as a MAIT cell resource is that the matched blood and liver are not from healthy donors, but from patients with various types of cancer. Additionally, there is no information on medications that may impact MAIT cells from these donors. Although it is acknowledged that obtaining healthy liver is difficult, healthy blood MAIT cells could easily be included for comparison, which would add to the value as a resource. It appears the stimulation experiments were performed using healthy blood donors (although of younger age), and so a more comprehensive analysis may be possible with existing data. At a minimum relevant clinical information should be added and the implications discussed (for example, in line 294 the authors make conclusions about "resting" MAIT cells, but do not consider the possible response to disease or medications in the patient. The authors should consider changing the title to better indicate the MAIT cells being analysed.

Minor comments:

1. How can the authors be sure some liver MAIT cells were not blood MAIT cells in transit through the liver? The authors should state if their experimental approach accounted for or considered this as a possibility.

2. A previous study Dias et al., 2018 suggests that some DN MAIT cells may be functionally distinct yet derived from a subset of CD8+ MAIT cells. To investigate these findings the authors may perform a comparison between TCR clonotypes and 'transcriptionally defined' DN or CD8+ coreceptor subsets.

3. It is unclear to this reviewer why only CD8+ MAIT cells were examined for functional RNA-seq analysis, especially as it has been reported that CD4+ MAIT cells are less sensitive to TCR stimulation than other subsets Kurioka et al., 2017. The authors should reiterate for clarity that these data are limited to a subset of MAIT cells.

4. Generally, the manuscript lacks verification of the transcriptomic differences observed between blood and liver MAIT cells. Verification of key genes (CD69, ICOS, CXCR6, etc.) by flow cytometry or other means would further strengthen the validity of these observations.

5. Some tidying up of the wording might improve the abstract. It is unclear what is meant by "this model" in line 3. "Surprising clonal diversity" I believe has been demonstrated previously and thus may be a little overstated.

6. In the introduction first few paragraphs the references are all review articles. It would be helpful in a resource to also cite the original publications appropriate to each point.

7. Line 55-61. The wording is a little unclear around the human/mouse differences and hypothesis on

a “distinct type 17 subset”.

8. Line 67: “produced in other tissues” should be “produced by MAIT cells in other tissues”. Also, perhaps this should be tempered to “described to date” as I don’t believe this has been thoroughly examined.

9. Line 637. The heading “Data Generation” in methods has no associated text

Reviewer #3:

Remarks to the Author:

The paper by Garner and coworkers is rich with data about human MAIT cells, using scRNAseq, scTCRseq and a term they have created; functional RNAseq. They have examined a large number of MAIT cells from blood and liver of three humans and subjected these cells to two types of activation. As expected from study like this, the volume of information is large but it is also difficult to extract the key messages. Regardless seeing this information integrated at the single cell level with antigen mediated TCR activation versus cytokine mediated activation in the context of TCR clonotypes at a large scale in human samples is valuable.

Many of the findings are predictable based on earlier reports e.g the predominant use of TRAJ33 then TRAJ12 and 20; and the existence of greater diversity of TCR-b compared to TCR-a which explains the greater promiscuity for TCR-a pairing to different TCR-b than the other way round. There are other studies recently published examining human MAIT cell transcriptomics at single cell level and it is appropriate to refer to these where data in this paper aligns or otherwise. E.g. Vorkas et. Al. J.Imm; Chandra et. Al. BioRxiv.

My main concern is the limited depth, novelty and conceptual advance of the study. This would be a useful resource paper but unsure if Nature Immunology is the right journal for this.

Some specific comments and questions:

- 1) The comparison of TCR and cytokine stimulation is very interesting and reveals many differences. We already know these stimuli drive different outcomes from MAIT cells but not to the level provided here. In a physiological setting MAIT cells are likely to receive both TCR and cytokine signals together, which raises the question of whether they would follow one or other path shown in Figure 6 or a distinct or intermediate path. This would add great value because it would provide insight into which pathway dominates and how MAIT cells would be likely respond in vivo where both signals would be present.
- 2) Extended data figure 4h. It is stated that CD4 and CD8 did not occupy distinct clusters however they do appear in distinct locations on the UMAP1/2 analysis, rather than intermingled so this explanation is oversimplified.
- 3) In the activation analysis would it be possible to examine how CD4 or CD8 or DN MAIT cells respond? this would add value to earlier section looking at non-activated MAIT cells separated by these markers because differences may require activation.
- 4) In discussion the statement that MAIT cells comprise a single functional population seems like oversimplification because we have seen a great range of subsets, with 11 different clusters where functional differences were observed. The reason is because the authors suggest the differences are due to activation state but this isn’t clear from my reading. Why can’t it be that different sublineages

have different expression of activation markers

5) There are some major gaps in the field that the introduction of this paper itself poses, however was not clearly addressed in the results and discussion. For instance whether IL-17+ human MAIT cells represent a distinct lineage or a state induced by activation was not explored in depth in this dataset, in part due to limitations in the technique such as lack of sequencing depth (as recognised by the authors). The inability to detect PLZF and RORc for instance, may hamper the comprehensiveness as a resource.

6) Throughout this paper and particularly in the activation analysis, there are no validation studies to further demonstrate the utility of this data set. It is very interesting that the authors have identified multiple candidates that are signatures of these activation trajectories but follow-up to show active expression of these markers at protein level will enhance this. A few examples, i) the jury is out with CD56 being a reliable marker for enhanced function within MAIT cells, as the authors probed for relative expression of NCAM1 in their dataset pre/post stimulation, there isn't follow up on supporting or opposing this question in the field. ii) the author team together with two other reports in 2019 (Lamichhane et al 2019 – described TCR stimulation producing More Inflammatory Cytokines than IL-12 and IL-18 Stimulation, was this also seen? Iii) Is the tissue repair signature also up in liver cells?

Author Rebuttal to Initial comments

See Inserted PDF.

Reviewer #1

(Remarks to the Author)

This paper by Garner et al. presents a comprehensive investigation into the single cell transcriptome of human MAIT cells. The paper presents two complementary data sets; The first is based on paired blood and liver samples from three donors, and the second one is based on blood MAIT cells activated through TCR or via IL-12+IL-18. In addition to the scRNAseq they have also nicely utilized the 10X platform to generate complementary scTCR sequencing from the same cells. This allows analysis of the TCR alpha and beta chain pairing at the single cells level, and also analysis of the transcriptional signature of different TCR clonotype cells. Overall, the paper is very well written and presented. This reviewer believes it can be a useful resource for the field, but still have some concerns and suggestions that may help improve the paper.

We thank the reviewer for their positive comments on the manuscript, and appreciate the constructive feedback, which we feel has substantially strengthened the revised manuscript. We have added new data (n = 3 additional matched blood-liver samples; additional stimulation conditions and timepoints; protein validation; ATAC-seq data on blood MAIT cells) and amended the text, which we hope addresses the concerns raised.

The overall message in the title and abstract is that “human MAIT cells show clonal diversity but transcriptional and functional homogeneity”. The statement about clonal diversity is accurate although perhaps not novel as such. Perhaps a more interesting aspect is the potential link between clonotypes and responsiveness. I will comment more on that below. The transcriptional homogeneity is well supported, although perhaps a more interesting message is the consistent pattern in all three donors that the two tissue sites are associated with distinct transcriptional profiles while the TCR repertoire is shared between the sites within each donor. The final part of the title “functional homogeneity” is not well supported in this data set as function was strictly speaking not assessed. Rather, the authors performed TCR and cytokine stimulations and assessed the transcriptional response.

We agree that the title and abstract needed revision, and have removed the phrases “clonal diversity” and “functional homogeneity”. However, we have retained the word “functional”, since we have now included flow cytometry validation of selected tissue-specific proteins, activation markers, and cytokines. In addition, the new blood-liver (Exp 2) and stimulation (Exp 4) datasets both include 130 CITE-seq antibodies (the expression of which highly correlate with the corresponding genes). Our revised title is: “Single-cell analysis of human MAIT cell transcriptional, functional, and clonal diversity”. Protein-level validation can be found in Fig. 1i, Extended Data Fig. 2b, 2e-g, 2j-k, 8c, 8e-j, 11d, 11f-g, and Supplementary Tables 2,3,6,10,11.

It is very nice to have the paired blood-liver samples. At the same time data from only three pairs is a bit limited. For example, the interesting findings that the MAIT cell TCR repertoire show a high degree of intra-donor similarity between blood and liver (Fig 3 a), while there is little sharing of clonotypes between donors, would be much more solid if resting on more than three donors. Likewise, the striking difference between the TCRalpha and beta clonotypes, where most of the TCRalpha clonotypes are public while there are very few if any public TCRbeta clones, would be much stronger with a few additional donors. A higher number of blood-liver pairs may solidify or refute some of these patterns.

We took note of this clear comment (also similarly made by other reviewers) and have effectively repeated the entire experiment to generate a much bigger dataset across a larger number of donors. Experiment 1 (original paper) contained findings in the tissue section from 3 liver donors (4 blood).

We obtained a further 3 liver samples and 8 blood samples (Exp 2) to address the reviewer's questions. To gain maximum value from this process, we integrated the two datasets as far as possible. The findings from 6 blood-liver pairs mirror what was originally found and we feel this is now a very robust dataset – accounting for any issues with patient/donor variability and sampling. These data are contained in Fig. 1-4. In total, across all experiments, we have scRNA-seq and scTCR-seq data on 18 individuals (12 for blood-liver comparisons and 6 for in vitro stimulation) in the revised manuscript.

The authors themselves point out the limitation that the blood and liver tissue samples come from cancer patients with liver metastases. It is well known that chronic conditions of different kinds can have significant effects on the MAIT cell compartment, including the activation state. How can we be sure that the current data set does not reflect a disease state of the MAIT cell compartment? Perhaps comparisons between the blood data from the blood-liver pairs, and the unstimulated control data from the healthy donor blood in vitro stimulation data set can be helpful?

This important point was also addressed with the new liver samples, which came from donors with benign lesions. In addition, we included a comparison showing that blood from entirely healthy donors is comparable to that from patients who underwent liver surgery (Fig. 1b). We have also included a table (Supplementary Table 1) to indicate patient age, gender, and diagnosis – in the final total of 7 liver patients, two patients had primary hepatocellular carcinoma, one had an isolated metastasis from a colonic tumour (this patient had had prior chemotherapy > 2m prior to surgery, a treatment not used in any of the others), and the others had surgery for benign conditions (e.g. haemangioma).

In a way the conclusions presented about homogeneity in the MAIT cell compartment is very well aligned with the literature, as there is strikingly little phenotypic diversity among MAIT cells in blood. A couple of studies have reported differences in MAIT cell functional phenotype depending on expression of CD8a, CD8b or CD4 (or lack of expression i.e. DN). In general, the lack of separation of these coreceptor genes in the present UMAPs does not really stand in contrast with published phenotypic differences and apoptosis sensitivity. One can always discuss the definition of what is a subset, but the lack of finding one based on transcriptomics alone does not convincingly show the absence. For the CD4⁺ MAIT cells that is probably even more true as CD4 transcripts can be hard to detect efficiently.

We appreciate the issue raised by the reviewer. We have now used two different approaches to define CD8⁺, DN, and CD4⁺ MAIT cells – either using expression of the *CD8A/B* and *CD4* genes, or expression of the CD4 and CD8 CITE-seq antibodies (Fig. 3h,i and Extended Data Fig. 4h-k). The frequency of CD4⁺ MAIT cells identified using expression of the *CD4* gene is in line with previous flow cytometry studies (e.g. Gherardin et al., Immunol Cell Biol, 2018). While we do identify a small number of genes and proteins that are differentially expressed between CD8⁺, DN, and CD4⁺ MAIT cells (Supplementary Table 6), we cannot identify a reliable clustering that splits them taking all the data into account in an agnostic, unsupervised way. We acknowledge that this does not preclude some phenotypic/functional differences. However, as unsupervised clustering is robust for the separation of conventional CD4⁺ and CD8⁺ T cells, any difference in MAIT cell biology that does exist based on differential expression of these co-receptors must be markedly more subtle than is seen in the conventional T cell population.

For analysis of CD4⁺ MAIT cells, there is a risk of contamination of this small population with the much larger conventional CD4⁺ T cell subset. Therefore, for differential expression analysis, we analysed only cells with a *TRAV1-2-TRAJ33/12/20* TCR.

The data with variable cluster bias of different clonotypes in Fig. 5 is interesting (also in the light of the cluster association with activation potential in Fig. 8). Just double checking that the clustering in Fig. 5 was done without the TCR data, so it is not so that the clustering is dependent on the TCR data (i.e. the finding generating itself so to say)?

Yes, we can confirm the two analyses were done independently and TCR genes were removed before clustering analysis. Fig. 4d now includes analysis of the clonotype-cluster association for 12 donors.

The data in Fig. 6 is consistent with the literature and adds considerable detail to the different types of responses triggered by TCR vs cytokine stimulus. The identification of predicted transcription factor regulons associated with the two types of stimuli, the HIVEP3- and BATF-dependent regulons, may prove to be the most interesting finding in the paper. The reader of course wonders if the authors have been able to experimentally test the importance of these transcription factors?

This was a major focus of effort in the lab for this revision. We used Dharmacon Accell siRNAs (SMARTpool) from Horizon Discovery targeting HIVEP3 and BATF, as well as STAT4 as a positive control. However, unfortunately, even with our positive control, the amount of knockdown achieved was insufficient to produce a reliable functional effect (IFN γ production measured in response to 20-hour IL-12 + IL-18 stimulation). Either additional optimisation or use of a completely different approach will be required to address this question in a satisfactory manner. We appreciate this would be an important area for future work, but given the substantial technical difficulty in performing the experiment and the time constraints, we had to prioritise the major dataset expansion for the revision of this paper.

The apparent differences between TCR clonotypes in responsiveness to both TCR stimulus and cytokine stimulus is interesting, and reasonably consistent with previous findings that the TCRbeta chain can influence MAIT cell triggering. The finding that responsiveness to also cytokine stimulation differs between clonotypes is puzzling. Does this somehow imply that a previous TCR-dependent experience will make some MAIT cells more prone to cytokine responsiveness?

The gradual expansion of MAIT cells through the first decade of life in response to the microbiome suggests that differences in exposures (either over time or in space) could lead to a diversity of experience for the MAIT cell pool. As the reviewer suggests, we hypothesise that the activation history of a clone could influence its subsequent responsiveness to TCR and/or cytokine stimulation. Clearly, this will require a dedicated study to properly test this hypothesis.

Additional reviewer point: A similar paper was published, presenting somewhat different conclusions: <https://www.jimmunol.org/content/early/2022/02/11/jimmunol.2100522>. This paper by Vorkas et al sort of challenges the conclusions by Garner et al, as Vorkas finds evidence of heterogeneity and discrete subsets."

We have carefully reviewed this paper and examined the conclusions, together with other relevant studies published since our initial submission, and have included a paragraph on this in the discussion (lines 746-767). We believe that differences in scope of work and experimental approach distinguish our current manuscript from prior studies. Most importantly, our inclusion of TCR analysis is a unique strength, as is our use of multiple stimulation conditions (additional TCR + cytokine stimulation in the revised manuscript). Thus, we feel our revised manuscript will be an invaluable resource for the field.

With regards to the different conclusions drawn by us as compared to Vorkas and colleagues, we believe there are a number of important technical differences e.g. around the definition of MAIT cells, which explain the different interpretations. As observed by Vorkas et al., we do see upregulation of *FOXP3* expression following MAIT cell activation (see below). However, since these *FOXP3*-expressing cells also express other effector molecules, such as *IFNG*, we believe that *FOXP3* is a marker of activation in this context, as observed for conventional T cells (<https://pubmed.ncbi.nlm.nih.gov/17154262/>). Some of the activation-induced clusters in the Vorkas publication align with those observed in our stimulation datasets, but based on our trajectory and clonal analysis, we argue that these represent cells along an activation continuum, rather than discrete functional subsets.

Reviewer #2

(Remarks to the Author)

Title: Human MAIT cells show clonal diversity but transcriptional and functional homogeneity

Summary:

Garner and colleagues present a valuable resource and comprehensive analysis of paired single cell transcriptomic and TCR sequence data from human blood and liver MAIT cells (MR1/5-OP-RU+). The authors include an analysis of blood MAIT cells at rest and post TCR- or cytokine-mediated stimulation from an additional cohort, termed 'functional RNA-seq'.

The authors reveal that MAIT cells from blood and liver express distinct transcriptomes, namely liver MAIT cells possessed a tissue-resident-like transcriptional and regulatory profile. Despite this, between donors and within distinct tissues, MAIT cells expressed transcriptional homogeneity. As expected, a high degree of conservation existed among TCRa clonotypes between tissues and donors. In contrast, TCRab clonotypes were diverse and private but shared across tissues. Some private TCRab clonotypes appeared to associate with specific transcriptional clusters. The authors reveal that blood MAIT cells progress along distinct, consistent trajectories upon TCR or cytokine stimulation that is somewhat related to private TCR clonotypes, but likely not solely TCR signal strength.

The paper is well written, with appropriate methodology and statistics, and provides a large dataset from which the field can draw knowledge (as the authors claim, their study includes analysis of more MAIT cells than previous studies and single cell rather than bulk sequencing). The study also brings together scRNAseq, TCR sequencing and analysis of both activated and unstimulated cells from blood and liver. These provide an additional data and analyses while transcriptomic analysis of human MAIT cells is still relatively new. Certainly there is a lack of comprehensive analysis of MAIT cells across various tissues and in vitro stimulation conditions. However, the novel findings and value as an overarching resource for the MAIT cell field need to be better highlighted. Specific comments are provided below.

We appreciate the positive comments, as well as the need to better highlight the novelty of our study, as well as its value as a resource. We have modified the text to emphasise novel findings (as well as de-emphasised other data that was largely confirmatory of prior findings e.g. moved general characterisation of the TCR repertoire to Extended Data Fig. 3) and have included a large amount of new data that we believe significantly adds to the value of the resource. The new data includes two additional CITE-seq datasets – one from resting blood and liver MAIT cells, and one from stimulated MAIT cells at multiple timepoints and with a combined TCR + cytokine stimulation – as well as ATAC-seq data from blood T cells.

Major comments:

1. In figure 1 and figures 4-5 human MAIT cells are compared to gene signatures from murine T cells. If available, the manuscript would benefit from a comparison between relevant human T cell subsets. For example, human Th1 and Th17 cells, or human liver-derived memory T cells. Tissue residency gene signatures are available for human samples.

We have now added additional analyses using human Th1, Th17, and T_{RM} cell gene signatures (Fig. 1j; Extended Data Fig. 4b). We originally chose the murine gene signatures, since MAIT1 and MAIT17 cells have not been characterised in humans and because functional tissue residency is hard to establish in humans. Our new analysis based on human signatures is consistent with the original findings and strengthens our conclusions.

2. The authors acknowledge several times that their findings are consistent with previously published literature. Specifically, tissue resident signatures being more abundant in liver than blood was reported using bulk RNAseq (Salou 2019). The lack of distinct MAIT1 and MAIT17 subsets in humans was reported by Leeansyah 2015 and Dias 2017). The diverse TCRb and fixed TCRa repertoire is also known. However, comparisons with memory T cells are interesting and will be important to the field in understanding MAIT cell responses and memory, particularly given the interest in vaccination strategies targeting these cells. As above, the text could better clarify what is novel here and the implications for understanding MAIT cell biology.

Based on the reviewer's suggestion, we have included additional analyses comparing MAIT cells to T_{mem} cells in the blood and liver with respect to both the transcriptome and TCR repertoire (Fig. 1a-c, Extended Data Fig. 2a-g,l,m; Supplementary Table 2,3). We feel this new data highlight what is unique (and not) about MAIT cell biology within tissues and regarding clonal architecture. Further, we have tried to highlight better what is novel both in the originally and newly added data (and de-emphasised less critical confirmatory data). We feel one particular advantage of the scRNA-seq approach is that it can clarify whether a given change e.g. in the tissue, is the result of an enrichment in a specific cell subset, or a more global adaptation.

3. One issue with this dataset and analysis serving as a MAIT cell resource is that the matched blood and liver are not from healthy donors, but from patients with various types of cancer. Additionally, there is no information on medications that may impact MAIT cells from these donors. Although it is acknowledged that obtaining healthy liver is difficult, healthy blood MAIT cells could easily be included for comparison, which would add to the value as a resource. It appears the stimulation experiments were performed using healthy blood donors (although of younger age), and so a more comprehensive analysis may be possible with existing data. At a minimum relevant clinical information should be added and the implications discussed (for example, in line 294 the authors make conclusions about "resting" MAIT cells, but do not consider the possible response to disease or medications in the patient. The authors should consider changing the title to better indicate the MAIT cells being analysed.

We acknowledge this limitation in the original data (an issue also raised by other reviewers). To address this, we have included liver samples from three additional donors undergoing liver surgery for benign lesions (no underlying disease or relevant therapy). Additionally, we have performed additional sequencing on blood MAIT cells from four healthy donors in order to directly compare the transcriptional profile of blood MAIT cells from liver donors and entirely healthy donors. Our analysis shows them to be highly comparable (Fig. 1b). We have included relevant clinical data in Supplementary Table 1.

Minor comments:

1. How can the authors be sure some liver MAIT cells were not blood MAIT cells in transit through the liver? The authors should state if their experimental approach accounted for or considered this as a possibility.

Our experimental approach aimed to remove as much blood as possible from the liver samples by allowing it to drain out of the tissue prior to processing. However, we acknowledge that a small fraction of circulating MAIT cells may have remained within the tissue. It is therefore possible that the observed TCR sharing between blood and liver is augmented by the presence of blood MAIT cells within the liver, while a second population of liver-resident MAIT cells gives rise to the tissue residency gene signature. This is unlikely given the high fraction of MAIT cells belonging to clonotypes found in both blood and liver. However, to computationally test this hypothesis, we have

performed gene set enrichment analysis between liver and blood MAIT cells, limiting our analysis only to cells from clonotypes found in both tissues. Enrichment of tissue residency genes in the liver confirms that MAIT cells expressing the same TCRs can exhibit different transcriptional profiles dependent on tissue localisation (see below). For the sake of length, these analyses have not been included in the final manuscript, as the conclusions are comparable to those from analysis using the entire dataset.

Milner et al., Nature, 2017 ($p = 0$, NES = 2.85):

Poon et al., Nat Immunol, 2023 ($p = 0$, NES = 2.73):

2. A previous study Dias et al., 2018 suggests that some DN MAIT cells may be functionally distinct yet derived from a subset of CD8+ MAIT cells. To investigate these findings the authors may perform a comparison between TCR clonotypes and 'transcriptionally defined' DN or CD8+ coreceptor subsets.

We thank the reviewer for the suggestion and we have now included this analysis in Fig. 3j,k and Extended Data Fig. 4l,m. We were careful to avoid biases in the analysis by downsampling the number of CD8+ MAIT cells within each donor to match the number of DN cells. Consistent with the

findings of Dias et al., PNAS, 2018, we see a small increase in the number of unique clonotypes in CD8⁺ relative to DN MAIT cells. However, we observe no difference in the Shannon diversity index, or in the use of *TRAJ* and *TRBV* chains between these populations. Overall, our findings indicate that the two populations are highly overlapping in TCR usage, which could be consistent with one population being derived from the other, but this hypothesis would need to be directly tested in a further study.

3. It is unclear to this reviewer why only CD8⁺ MAIT cells were examined for functional RNA-seq analysis, especially as it has been reported that CD4⁺ MAIT cells are less sensitive to TCR stimulation than other subsets Kurioka et al., 2017. The authors should reiterate for clarity that these data are limited to a subset of MAIT cells.

We originally focussed on CD8⁺ MAIT cells, since these comprise a large fraction of human MAIT cells and we wanted to perform an enrichment step prior to cell sorting, which was easily achieved using CD8 MicroBeads. However, we recognised this was a potential limitation and included all CD3⁺ MAIT cells in the new stimulation experiment (Exp 4). In addition, we have clarified in the text that we specifically sorted CD8⁺ MAIT cells for Exp 3 (line 350).

Unfortunately, we were unable to robustly assess differences in the activation potential of CD8⁺, DN, and CD4⁺ MAIT cells, given the very small number of CD4⁺ cells detected and the strong downregulation of CD8 upon MAIT cell activation (for more details, please see our response to reviewer 3, comment 3).

4. Generally, the manuscript lacks verification of the transcriptomic differences observed between blood and liver MAIT cells. Verification of key genes (CD69, ICOS, CXCR6, etc.) by flow cytometry or other means would further strengthen the validity of these observations.

This limitation of our initial work was also helpfully highlighted by Reviewer 1. We have addressed this point in two ways. Firstly, we performed flow cytometry validation of CD69 expression in the liver, as well as of a number of TCR-specific, cytokine-specific, or shared activation markers and cytokines (Extended Data Fig. 2k; Extended Data Fig. 8c,e-j). Secondly, we included a panel of 130 CITE-seq antibodies in the new blood-liver and activation datasets (Exp 2 and 4) – we used these to verify the expression of specific proteins, and also performed differential protein expression analysis between cell types, tissues, stimulation conditions etc. (Extended Data Fig. 2b,e-g,j; Extended Data Fig. 11d,f,g; Supplementary Tables 2,3,6,10,11).

5. Some tidying up of the wording might improve the abstract. It is unclear what is meant by “this model” in line 3. “Surprising clonal diversity” I believe has been demonstrated previously and thus may be a little overstated.

We have edited the abstract, also in the light of the added data, and removed these phrases.

6. In the introduction first few paragraphs the references are all review articles. It would be helpful in a resource to also cite the original publications appropriate to each point.

We have addressed this with references to the primary literature.

7. Line 55-61. The wording is a little unclear around the human/mouse differences and hypothesis on a “distinct type 17 subset”.

We have modified this text (lines 65-68).

8. Line 67: “produced in other tissues” should be “produced by MAIT cells in other tissues”. Also, perhaps this should be tempered to “described to date” as I don’t believe this has been thoroughly examined.

We have modified this (lines 73-74).

9. Line 637. The heading “Data Generation” in methods has no associated text

Apologies the formatting was unclear – this was supposed to act as a subheading and we have now underlined it to indicate this more clearly.

Reviewer #3

(Remarks to the Author)

The paper by Garner and coworkers is rich with data about human MAIT cells, using scRNAseq, scTCRseq and a term they have created; functional RNAseq. They have examined a large number of MAIT cells from blood and liver of three humans and subjected these cells to two types of activation. As expected from study like this, the volume of information is large but it is also difficult to extract the key messages. Regardless seeing this information integrated at the single cell level with antigen mediated TCR activation versus cytokine mediated activation in the context of TCR clonotypes at a large scale in human samples is valuable.

Many of the findings are predictable based on earlier reports e.g the predominant use of TRAJ33 then TRAJ12 and 20; and the existence of greater diversity of TCR-b compared to TCR-a which explains the greater promiscuity for TCR-a pairing to different TCR-b than the other way round. There are other studies recently published examining human MAIT cell transcriptomics at single cell level and it is appropriate to refer to these where data in this paper aligns or otherwise. E.g. Vorkas et. Al. J.Imm; Chandra et. Al. BioRxiv.

My main concern is the limited depth, novelty and conceptual advance of the study. This would be a useful resource paper but unsure if Nature Immunology is the right journal for this.

We thank the reviewer for their comments and have taken on board the concerns raised. We have endeavoured to better highlight our novel findings, as well as de-emphasise findings which are consistent with prior studies (although not previously described as comprehensively e.g. general TCR characterisation has been reduced and plots largely moved to Extended Data Fig. 3). Additionally, we have included a large amount of new data, which we believe adds to the robustness of our findings and the value of this study as a resource for the MAIT cell community and beyond. New data includes two CITE-seq datasets – one from resting blood and liver MAIT cells, and one from stimulated MAIT cells at multiple timepoints and with a combined TCR + cytokine stimulation – as well as protein-level functional validation and ATAC-seq data from blood T cells.

Some specific comments and questions:

1) The comparison of TCR and cytokine stimulation is very interesting and reveals many differences. We already know these stimuli drive different outcomes from MAIT cells but not to the level provided here. In a physiological setting MAIT cells are likely to receive both TCR and cytokine signals together, which raises the question of whether they would follow one or other path shown in Figure 6 or a distinct or intermediate path. This would add great value because it would provide insight into which pathway dominates and how MAIT cells would be likely respond in vivo where both signals would be present.

To address this excellent point, our new stimulation CITE-seq dataset (Exp 4) includes a combined TCR and cytokine stimulation condition at two timepoints (20 hours and 68 hours) (Fig. 8; Extended Data Fig. 11,12). We observed that TCR + cytokine-stimulated cells lie between TCR and cytokine single-stimulated cells on the UMAP (Fig. 8a), and that changes in gene and protein expression following stimulation were highly correlated between the single and dual stimulation conditions (Fig. 8c,d; Extended Data Fig. 11f,g). *IL17F* was a notable exception, being largely restricted to the TCR + cytokine dual stimulation condition. IL-17-expressing MAIT cells comprised a distinct cluster at both 20 hours and 68 hours, and we explored gene expression and TCR usage by this population (Fig. 8; Extended Data Fig. 12).

2) Extended data figure 4h. It is stated that CD4 and CD8 did not occupy distinct clusters however

they do appear in distinct locations on the UMAP1/2 analysis, rather than intermingled so this explanation is oversimplified.

We agree that this was oversimplified. However, our overall conclusion is supported by our larger dataset, which now comprises matched blood and liver from six donors. We were unable to identify distinct CD8⁺, DN, and CD4⁺ MAIT cell clusters in the blood or liver (Extended Data Fig. 4h,i). Co-receptor fractions were similar across clusters, regardless of whether co-receptor identity was assigned using gene expression (Fig. 3h,i) or protein expression (Extended Data Fig. 4j,k). Moreover, our analysis identified very few genes and proteins differentially expressed between these populations (Supplementary Table 6). Therefore, while we acknowledge that these populations may show some differences, they did not appear to segregate in an unsupervised analysis – as is seen for conventional T cell populations (Extended Data Fig. 2b).

3) In the activation analysis would it be possible to examine how CD4 or CD8 or DN MAIT cells respond? this would add value to earlier section looking at non-activated MAIT cells separated by these markers because differences may require activation.

This was an important aim of the new revised experiment (Exp 4). Originally, we planned to sort CD8⁺, DN, and CD4⁺ MAIT cells prior to stimulation. However, attempts to perform sorting prior to stimulation led to marked cell death. Overnight resting of sorted cells prior to stimulation reduced this issue, but we were concerned that this resting step may result in considerable transcriptional changes even within the unstimulated population.

As a result, we decided to perform stimulation prior to sorting and to include CITE-seq antibodies for detection of CD8 and CD4 expression. Unfortunately, this had its own issues. Firstly, due to background expression of the CD8 and CD4 CITE-seq antibodies, it was challenging to define specific thresholds of expression for distinguishing expressing and non-expressing cells. Secondly, CD8 expression was strongly downregulated following stimulation at both the RNA and protein level (see violin plots below). Thus, it was impossible to distinguish CD8⁺ cells that had transiently downregulated CD8 from bona fide DN MAIT cells. For these reasons, we are reluctant to draw strong conclusions about functional differences between these subsets. Nevertheless, we did not observe clusters restricted to CD8⁺, DN, or CD4⁺ cells following activation.

20 hours:

CD8A (left) and CD8B (right) gene

CD8 protein

68 hours:

CD8A (left) and CD8B (right) gene

CD8 protein

4) In discussion the statement that MAIT cells comprise a single functional population seems like oversimplification because we have seen a great range of subsets, with 11 different clusters where functional differences were observed. The reason is because the authors suggest the differences are due to activation state but this isn't clear from my reading. Why can't it be that different sublineages have different expression of activation markers

This is an important point which we have aimed to explain and display better. There are clusters seen at rest and also following stimulation. However, at baseline, very few genes showed cluster-specific expression and clusters did not reflect any recognised T cell differentiation states. Moreover, baseline clusters did not clearly map onto the functional changes observed following activation. Following stimulation, functional differences were dominated by the nature of the stimulus and cells appeared to fall along linear trajectories representing continuums of activation. This interpretation is further supported by our new dataset, where we observed an IL-17-expressing cluster specifically in the dual TCR + cytokine stimulation condition (Fig. 8). Compared with IL-17⁻ cells, IL-17⁺ cells showed comparable expression of other effector molecules, such as *IFNG*, and similar TCR usage, suggesting this cluster reflects a state of activation rather than a bona fide MAIT17 subset. Thus, at least for the stimulations used in the current study, we believe our data support a model whereby all measured effector functions are accessible to every MAIT cell under the appropriate conditions.

5) There are some major gaps in the field that the introduction of this paper itself poses, however was not clearly addressed in the results and discussion. For instance whether IL-17⁺ human MAIT cells represent a distinct lineage or a state induced by activation was not explored in depth in this dataset, in part due to limitations in the technique such as lack of sequencing depth (as recognised by the authors). The inability to detect PLZF and RORc for instance, may hamper the comprehensiveness as a resource.

We thank the reviewer for highlighting a disconnect in the original manuscript between questions we raised versus what we were able to address. To address this limitation, in the revised manuscript, the new stimulation experiment (which includes a TCR + cytokine stimulation condition and a longer stimulation condition) was specifically designed to allow investigation of IL-17 biology in MAIT cells (Fig. 8; Extended Data Fig. 12). Our results show that IL-17-expressing cells form a distinct cluster amongst dual TCR + cytokine stimulated cells at both 20 hours and 68 hours. However, IL-17⁻ and IL-17⁺ cells show similar expression of other effector molecules, such as *IFNG*, and overlapping TCR usage. Our ATAC-seq data provides a possible rationale for delayed IL-17 production relative to other effector molecules, such as granzyme B. Overall, our data suggests that IL-17⁺ cells reflect a specific activation state rather than a bona fide MAIT17 subset. We feel this has strongly enhanced the utility and comprehensiveness of this dataset as a resource.

With respect to detection efficiency of *ZBTB16* and *RORC*, all of our single-cell datasets have been sequenced to a high depth (mean read counts per experiment listed in the methods). We acknowledge that transcription factors can show poor capture rates using droplet-based scRNA-seq approaches. However, we do clearly detect expression of both *ZBTB16* and *RORC* in MAIT cells (see below).

6) Throughout this paper and particularly in the activation analysis, there are no validation studies to further demonstrate the utility of this data set. It is very interesting that the authors have identified multiple candidates that are signatures of these activation trajectories but follow-up to show active expression of these markers at protein level will enhance this. A few examples, i) the jury is out with CD56 being a reliable marker for enhanced function within MAIT cells, as the authors probed for relative expression of NCAM1 in their dataset pre/post stimulation, there isn't follow up on supporting or opposing this question in the field.

This limitation of our original submission was also highlighted by the other reviewers, and we have attempted to address this through two approaches. We have added flow cytometry validation of several TCR-specific, cytokine-specific, and shared activation markers and cytokines (Extended Data Fig. 8c,e-j). In addition, our new blood-liver and stimulation datasets (Exp 2 and 4) both include a 130 antibody CITE-seq panel, which allowed us to validate the expression of other proteins (Extended Data Fig. 2b,e-g,j; Extended Data Fig. 11d,f,g; Supplementary Tables 2,3,6,10,11).

We have directly addressed the CD56 question by sorting CD56⁻ and CD56⁺ MAIT cells and stimulating them with IL-12 + IL-18. In the three donors examined, CD56 does appear to identify MAIT cells with enhanced functional capacity following cytokine stimulation (IFN γ , granzyme B, perforin, CD94; Extended Data Fig. 7i-l). However, these cells do not appear to comprise a transcriptionally distinct cluster at rest or following activation. A dedicated study will be required to fully understand how and why this marker is associated with cytokine responsiveness.

ii) the author team together with two other reports in 2019 (Lamichhane et al 2019 – described TCR stimulation producing More Inflammatory Cytokines than IL-12 and IL-18 Stimulation, was this also seen?

The key conclusion of our work in Leng et al., Cell Rep, 2019 (and also Hinks et al., Cell Rep, 2019) is that TCR stimulation led to a stronger association with a tissue repair signature – the Lamichhane et al., Cell Rep, 2019 dataset also agrees with this, although there are important differences in stimulation timings used (TCR condition – 6 hours in Lamichhane, 24 hours in Leng). In general, we observe broad transcriptional responses to both TCR and cytokine stimulation, with key genes identified as specific to the TCR (e.g. *CCL3*) or cytokine (e.g. *IL26*) trajectory. Our new stimulation dataset (Fig. 8) reveals that dual TCR + cytokine stimulation induces a transcriptional response that is essentially an amalgam of the response to the two single stimulation conditions. *IL17F* production is a notable exception, being almost exclusively limited to the dual stimulation condition.

iii) Is the tissue repair signature also up in liver cells?

Yes, we do see a small, statistically significant, increase in expression of the tissue repair gene signature in liver compared with blood MAIT cells (see below). However, for space reasons, we have not included this figure in the manuscript.

Decision Letter, first revision:

15th May 2023

Dear Lucy & Paul,

Thank you for submitting your revised manuscript "Single-cell analysis of human MAIT cell transcriptional, functional, and clonal diversity" (NI-RS33452A). It has now been seen by the original referees and their comments are below. The reviewers find that the paper has improved in revision, and therefore we'll be happy in principle to publish it in Nature Immunology, pending minor revisions to satisfy the referees' final requests and to comply with our editorial and formatting guidelines.

We will now perform detailed checks on your paper and will send you a checklist detailing our editorial and formatting requirements in about a week. Please do not upload the final materials and make any revisions until you receive this additional information from us.

If you had not uploaded a Word file for the current version of the manuscript, we will need one before beginning the editing process; please email that to immunology@us.nature.com at your earliest convenience.

Thank you again for your interest in Nature Immunology. Please do not hesitate to contact me if you have any questions.

Sincerely,

Laurie

Laurie A. Dempsey, Ph.D.
Senior Editor
Nature Immunology
l.dempsey@us.nature.com
ORCID: 0000-0002-3304-796X

Reviewer #1 (Remarks to the Author):

In this revised manuscript Garner et al. provide a comprehensive single cell analysis of human MAIT cell diversity in blood and liver presented as a resource to the immunology community. They have used multiple single cell technologies to describe in detail how the MAIT cell population adapts to tissue localization (blood vs liver), a detailed map of the structure of the MAIT cell TCR repertoire within and between individuals, how MAIT cell responses are to some extent associated with TCR clonotypes, and the transcriptional response trajectories upon TCR vs innate cytokine stimulation with identification of potential underlying transcriptional programs. Overall, they have addressed the initial reviewer comments in a very good way and the paper is now significantly improved. This reviewer has a few remaining comments regarding the data interpretation as detailed below:

In the Results section on lines 378-392 the authors address CD56 as a possible subset-defining

marker. The interpretation of this data is still confusing. They have done new experiments repeating the findings by others that CD56+ MAIT cells have enhanced functionality to IL-12+IL-18 stim as compared to their CD56- counterparts. So CD56+ MAIT cells are clearly functionally different. At the same time NCAM1 transcript is very poorly detected in scRNAseq on the 10X platform for reasons that are unclear. Still the authors claim that NCAM1 does not define a transcriptionally distinct cluster. I think that given the limitation of the methodology used it is unnecessary to make this claim and suggest revising this interpretation.

In the Results section on lines 478-507 and fig 7, the authors see that activation potential within donors is associated with TRBV usage while this does not translate to individual clonotype population size. This is interesting and consistent with differences seen based on TCRVb usage before. However, then they interpret that this all means that the TCR clonotype does not determine the MAIT clone ability to expand: "Therefore, our data does not support the hypothesis that larger clones have expanded relative to other clones due to higher intrinsic functionality" and "...differences in clonotype functionality appear to be cell-intrinsic rather than directly linked to the affinity of TCR-ligand binding". These interpretations are not supported by the data given that they have not assessed proliferative potential of clones or TCR affinity of clonotypes. I suggest moderating the conclusions to fit better with the data.

Reviewer #2 (Remarks to the Author):

The authors have substantially revised their manuscript, including new data to address several of the concerns.

Major comments

- The authors have addressed all major comments, including by presenting large, entirely new datasets in their rebuttal.

Minor comments: A couple of very minor points remain.

Previous point 1. The authors make a persuasive argument and provide some additional analyses (not part of revised manuscript) to address this reviewer's concern. However, the authors should acknowledge the possibility of blood MAIT cell contamination of the liver in the manuscript.

Line 350: The authors should explicitly state that MAIT cells were isolated using CD161 and CD26 in the text.

In Extended data figure 6: What is the population of CD26 negative, CD161 intermediate cells that appears when PBMCs are stimulated with MR1-5-OP-RU and anti-CD28 mAbs? Is this consistent among all donors examined? As this is a novel approach, technical details should be carefully considered and described in the text accordingly.

Reviewer #4 (Remarks to the Author):

The manuscript by Garner et al utilises transcriptional profiling, TCR sequencing, surface-protein expression and ATAC-seq at single cell resolution to provided a detailed analysis of matched human

blood and liver MAIT cells, and to profile trajectories of blood MAIT cells in response to distinct methods of stimulation. The major findings from this work are that human MAIT cells, unlike their murine counterparts, are relatively transcriptionally homogeneous and that the little diversity that is present is largely driven by tissue location. TCR analysis allowed marriage of the transcriptional differences to clonotype, demonstrating that the transcriptional profile of MAIT cells can be influenced by their clonal origin. The authors also demonstrate distinct transcriptional outcomes as a result of cytokine versus TCR-mediated activation as well as the length of stimulation.

The manuscript is very well written and presented, the experimental approach and analysis are sound, and the datasets robust, especially given the use of human tissue samples. Indeed the additional data added by the authors in response to the reviewer comments have substantially increased the robustness of the manuscript. The interpretation of the data is also well reasoned. Overall the manuscript feels complete.

Similar to other reviewers note that while there is substantial novelty in this work, the insights aren't ground-breaking, and there is a lot of overlap in findings from previous papers, albeit with far greater depth and conclusive outcomes. However, I think that the robustness and depth provided here make this manuscript a valuable resource that will be of great interest to the field and will be highly utilised and cited.

Some minor comments worth addressing:

1. Page 4, line 133: Here, the described data includes analysis of surface proteins, however this technical approach has not yet been introduced in the manuscript. Please allude to Protein analysis at the start of the results section for clarification.
2. Page 6, line 197: Please include references to those previous studies.
3. Page 6, line 208-9: Technically, an aspect of this has been investigated (See Gheradin, ICB 2018, end of results section), although I think what the authors are specifically referring to here is TCR-b variable gene pairing, rather than any aspect of the beta-chain (in the case of the mentioned paper, CDR3b specifically)? Please clarify, as this is also then referred to in the discussion.

Author Rebuttal, first revision:

We thank the reviewers for the overall positive assessment of our revised manuscript. The latest version of the manuscript has been substantially shortened (word count and number of references) to comply with *Nature Immunology's* guidelines. We have endeavoured to keep the spirit and key scientific points of all previously requested changes.

Reviewer #1 (Remarks to the Author)

In this revised manuscript Garner et al. provide a comprehensive single cell analysis of human MAIT cell diversity in blood and liver presented as a resource to the immunology community.

They have used multiple single cell technologies to describe in detail how the MAIT cell population adapts to tissue localization (blood vs liver), a detailed map of the structure of the MAIT cell TCR repertoire within and between individuals, how MAIT cell responses are to some extent associated with TCR clonotypes, and the transcriptional response trajectories upon TCR vs innate cytokine stimulation with identification of potential underlying transcriptional programs. Overall, they have addressed the initial reviewer comments in a very good way and the paper is now significantly improved.

This reviewer has a few remaining comments regarding the data interpretation as detailed below:

In the Results section on lines 378-392 the authors address CD56 as a possible subset-defining marker. The interpretation of this data is still confusing. They have done new experiments repeating the findings by others that CD56+ MAIT cells have enhanced functionality to IL-12+IL-18 stim as compared to their CD56- counterparts. So CD56+ MAIT cells are clearly functionally different. At the same time NCAM1 transcript is very poorly detected in scRNAseq on the 10X platform for reasons that are unclear. Still the authors claim that NCAM1 does not define a transcriptionally distinct cluster. I think that given the limitation of the methodology used it is unnecessary to make this claim and suggest revising this interpretation.

We agree that our interpretation was confusing and thank the reviewer for highlighting this. We have modified the text (lines 305-318) and included additional plots showing CD56 gene and protein (measured by CITE-seq) expression in resting blood and liver MAIT cells (Extended Data Fig. 7m,n) – CD56+ cells were found throughout the UMAP and did not comprise a transcriptionally distinct cluster in either tissue at rest.

In the Results section on lines 478-507 and fig 7, the authors see that activation potential within donors is associated with TRBV usage while this does not translate to individual clonotype population size. This is interesting and consistent with differences seen based on TCRVb usage before. However, then they interpret that this all means that the TCR clonotype does not determine the MAIT clone ability to expand: “Therefore, our data does not support the hypothesis that larger clones have expanded relative to other clones due to higher intrinsic functionality” and “....differences in clonotype functionality appear to be cell-intrinsic rather than directly linked to the affinity of TCR-ligand binding”. These interpretations are not supported by the data given that they have not assessed proliferative potential of clones or TCR affinity of clonotypes. I suggest moderating the conclusions to fit better with the data.

We have modified the text to modulate our conclusions and have removed the words “expanded” and “affinity” since proliferation and TCR affinity were not directly measured.

Based on the lack of association between clonotype size and activation capacity (pseudotime position) and the identification of clonal differences in activation capacity on the TCR and cytokine trajectories, the new text (lines 408-411) reads as follows. “Therefore, larger clones are not intrinsically more functional than smaller clones. Given that variation between clonotypes was observed on the cytokine as well as the TCR trajectory, differences in clonotype functionality may not solely be associated with the strength of TCR-ligand binding.”

Reviewer #2 (Remarks to the Author)

The authors have substantially revised their manuscript, including new data to address several of the concerns.

Major comments

- The authors have addressed all major comments, including by presenting large, entirely new datasets in their rebuttal.

Minor comments: A couple of very minor points remain.

Previous point 1. The authors make a persuasive argument and provide some additional analyses (not part of revised manuscript) to address this reviewer’s concern. However, the authors should acknowledge the possibility of blood MAIT cell contamination of the liver in the manuscript.

We have added a sentence to acknowledge the possibility of low-frequency blood MAIT cell contamination in the liver (lines 507-509).

Line 350: The authors should explicitly state that MAIT cells were isolated using CD161 and CD26 in the text.

We have now indicated the sorting strategy directly in the text (lines 279-283).

In Extended data figure 6: What is the population of CD26 negative, CD161 intermediate cells that appears when PBMCs are stimulated with MR1-5-OP-RU and anti-CD28 mAbs? Is this consistent among all donors examined? As this is a novel approach, technical details should be carefully considered and described in the text accordingly.

The CD3⁺CD26⁻CD8^{low}CD161^{int} lymphocytes in Supplementary Fig. 2c (previously Extended Data Fig. 6c) are present in all donors and are likely to be NK cells. We have added a sentence to the figure legend to indicate this.

Reviewer #4 (Remarks to the Author)

The manuscript by Garner et al utilises transcriptional profiling, TCR sequencing, surface-protein expression and ATAC-seq at single cell resolution to provide a detailed analysis of matched human blood and liver MAIT cells, and to profile trajectories of blood MAIT cells in response to distinct methods of stimulation. The major findings from this work are that human MAIT cells, unlike their murine counterparts, are relatively transcriptionally homogeneous and that the little diversity that is present is largely driven by tissue location. TCR analysis allowed marriage of the transcriptional differences to clonotype, demonstrating that the transcriptional profile of MAIT cells can be influenced by their clonal origin. The authors also demonstrate distinct transcriptional outcomes as a result of cytokine versus TCR-mediated activation as well as the length of stimulation.

The manuscript is very well written and presented, the experimental approach and analysis are sound, and the datasets robust, especially given the use of human tissue samples. Indeed the additional data added by the authors in response to the reviewer comments have substantially increased the robustness of the manuscript. The interpretation of the data is also well reasoned. Overall the manuscript feels complete.

Similar to other reviewers note that while there is substantial novelty in this work, the insights aren't ground-breaking, and there is a lot of overlap in findings from previous papers, albeit with far greater depth and conclusive outcomes. However, I think that the robustness and depth provided here make this manuscript a valuable resource that will be of great interest to the field and will be highly utilised and cited.

Some minor comments worth addressing:

1. Page 4, line 133: Here, the described data includes analysis of surface proteins, however this technical approach has not yet been introduced in the manuscript. Please allude to Protein analysis at the start of the results section for clarification.

We thank the reviewer for this helpful suggestion. We have added a sentence at the beginning of the results to indicate that surface proteins were analysed alongside RNA in Exp 2 (lines 102-103).

2. Page 6, line 197: Please include references to those previous studies.

This phrase has now been removed from the results section due to word count limitations. The similarity of basic TCR repertoire characteristics with prior studies is mentioned and referenced in the discussion (line 530).

3. Page 6, line 208-9: Technically, an aspect of this has been investigated (See Gheradin, ICB

2018, end of results section), although I think what the authors are specifically referring to here is TCR-b variable gene pairing, rather than any aspect of the beta-chain (in the case of the mentioned paper, CDR3b specifically)? Please clarify, as this is also then referred to in the discussion.

In this paragraph (now lines 167-170), we investigate whether *TRAJ33*, *TRAJ12*, and *TRAJ20* MAIT TCRs pair with *TRBV6-1*, *TRBV6-4*, and *TRBV20-1* at different frequencies. This is distinct from Gherardin et al., 2018 who show that HEK293T cells transfected with three different MAIT TCRs that share a CDR3 β /TRBJ sequence but have distinct CDR3 α /TRAJ sequences exhibit differential binding to the MR1/5-OP-RU tetramer.

The relevant sentence has been removed from the discussion due to word count restrictions.

Final Decision Letter:

Dear Lucy,

I am delighted to accept your manuscript entitled "Single-cell analysis of human MAIT cell transcriptional, functional, and clonal diversity" for publication in an upcoming issue of Nature Immunology.

Over the next few weeks, your paper will be copyedited to ensure that it conforms to Nature Immunology style. Once your paper is typeset, you will receive an email with a link to choose the appropriate publishing options for your paper and our Author Services team will be in touch regarding any additional information that may be required.

Please note that Nature Immunology is a Transformative Journal (TJ). Authors may publish their research with us through the traditional subscription access route or make their paper immediately open access through payment of an article-processing charge (APC). Authors will not be

required to make a final decision about access to their article until it has been accepted. [Find out more about Transformative Journals](https://www.springernature.com/gp/open-research/transformative-journals).

Authors may need to take specific actions to achieve [compliance with funder and institutional open access mandates](https://www.springernature.com/gp/open-research/funding/policy-compliance-faqs). If your research is supported by a funder that requires immediate open access (e.g. according to [Plan S principles](https://www.springernature.com/gp/open-research/plan-s-compliance)) then you should select the gold OA route, and we will direct you to the compliant route where possible. For authors selecting the subscription publication route, the journal's standard licensing terms will need to be accepted, including [self-archiving policies](https://www.springernature.com/gp/open-research/policies/journal-policies). Those licensing terms will supersede any other terms that the author or any third party may assert apply to any version of the manuscript.

Your paper will be published online soon after we receive your corrections and will appear in print in the next available issue. Content is published online weekly on Mondays and Thursdays, and the embargo is set at 16:00 London time (GMT)/11:00 am US Eastern time (EST) on the day of publication. Now is the time to inform your Public Relations or Press Office about your paper, as they might be interested in promoting its publication. This will allow them time to prepare an accurate and satisfactory press release. Include your manuscript tracking number (NI-RS33452B) and the name of the journal, which they will need when they contact our office.

About one week before your paper is published online, we shall be distributing a press release to news organizations worldwide, which may very well include details of your work. We are happy for your institution or funding agency to prepare its own press release, but it must mention the embargo date and Nature Immunology. Our Press Office will contact you closer to the time of publication, but if you or your Press Office have any enquiries in the meantime, please contact press@nature.com.

Also, if you have any spectacular or outstanding figures or graphics associated with your manuscript - though not necessarily included with your submission - we'd be delighted to consider them as candidates for our cover. Simply send an electronic version (accompanied by a hard copy) to us with a possible cover caption enclosed.

If you have not already done so, we strongly recommend that you upload the step-by-step protocols used in this manuscript to the Protocol Exchange. Protocol Exchange is an open online resource that allows researchers to share their detailed experimental know-how. All uploaded protocols are made freely available, assigned DOIs for ease of citation and fully searchable through nature.com. Protocols can be linked to any publications in which they are used and will be linked to from your article. You can also establish a dedicated page to collect all your lab Protocols. By uploading your Protocols to Protocol Exchange, you are enabling researchers to more readily reproduce or adapt the methodology you use, as well as increasing the visibility of your protocols and papers. Upload your Protocols at www.nature.com/protocolexchange/. Further information can be found at www.nature.com/protocolexchange/about .

Please note that we encourage the authors to self-archive their manuscript (the accepted version before copy editing) in their institutional repository, and in their funders' archives, six months after publication. Nature Portfolio recognizes the efforts of funding bodies to increase access of the research they fund, and strongly encourages authors to participate in such efforts. For information about our editorial policy, including license agreement and author copyright, please visit www.nature.com/ni/about/ed_policies/index.html

Kind regards,

Laurie

Laurie A. Dempsey, Ph.D.
Senior Editor
Nature Immunology
l.dempsey@us.nature.com
ORCID: 0000-0002-3304-796X